# THE COT ENCYCLOPEDIA: ANALYZING, PREDICTING, AND CONTROLLING HOW A REASONING MODEL WILL THINK

**Seongyun Lee**[1,3*]  **Seungone Kim**[2*]  **Minju Seo**[1]  **Yongrae Jo**[3]

**Dongyoung Go**[4]  **Hyeonbin Hwang**[1]  **Jinho Park**[1]

**Xiang Yue**[2]  **Sean Welleck**[2]  **Graham Neubig**[2]  **Moontae Lee**[3,5]  **Minjoon Seo**[1]

KAIST AI[1]    Carnegie Mellon University[2]    LG AI Research[3]
Cornell University[4]    University of Illinois Chicago[5]

{seongyun, minjoon}@kaist.ac.kr    seungone@cmu.edu

## ABSTRACT

Long chain-of-thought (CoT) is an essential ingredient in effective usage of modern large language models, but our understanding of the reasoning strategies underlying these capabilities remains limited. While some prior works have attempted to categorize CoTs using predefined strategy types, such approaches are constrained by human intuition and fail to capture the full diversity of model behaviors. In this work, we introduce the COT ENCYCLOPEDIA, a bottom-up framework for analyzing and steering model reasoning. Our method automatically extracts diverse reasoning criteria from model-generated CoTs, embeds them into a semantic space, clusters them into representative categories, and derives contrastive rubrics to interpret reasoning behavior. Human evaluations show that this framework produces more interpretable and comprehensive analyses than existing methods. Moreover, we show that this understanding translates into measurable improvements on both problem-solving and safety benchmarks. We can predict which strategy a model is likely to use and guide it toward more effective alternatives. Finally, we show that training data format (e.g., free-form vs. multiple-choice) impacts reasoning far more than data domain, highlighting the importance of format-aware model design. In short, the COT ENCYCLOPEDIA turns reasoning from a black box into a controllable asset, enabling LLMs that think more clearly, perform more reliably, and act more safely.

## 1 INTRODUCTION

Chain-of-thought (CoT) prompting (Wei et al., 2022) is an effective inference-time method for eliciting reasoning in large language models (LLMs) by generating intermediate reasoning steps before producing a final answer. While CoT reasoning has led to impressive performance gains—especially when extended to long chains involving multiple reasoning strategies (Guo et al., 2025; Jaech et al., 2024; Muennighoff et al., 2025; Yeo et al., 2025)—our understanding of the specific strategies that models tend to employ remains limited. Key questions remain underexplored: What varieties of reasoning strategies do models use? How do these strategies differ across models and tasks? Can they be systematically controlled to improve performance? Recent work on CoT monitoring (Korbak et al., 2025) underscores the importance of addressing these questions, showing that the transparency of intermediate reasoning steps is not only valuable for performance analysis but also a critical opportunity for ensuring safety.

---

[*]Equal contribution. Seongyun conducted this work during an internship at LG AI Research.

Prior efforts to analyze long CoT reasoning have primarily followed a top-down approach (Wen et al., 2025; Gandhi et al., 2025; Guo et al., 2025), where researchers define a fixed set of strategy types—such as backtracking or subgoal setting—and use language models to detect their presence in generated outputs. While interpretable, such approaches constrain analysis to known categories. Recent clustering-based methods (An et al., 2023; Xu et al., 2024; Fang et al., 2025) focus mainly on short/medium CoTs, leaving longer multi-strategy traces underexplored. For a direct comparison between our bottom-up clustering framework and existing top-down approaches, see Figure 1.

In this paper, we introduce the CoT ENCYCLOPEDIA, a framework to systematically analyze and control long CoTs that involve multiple, intertwined reasoning strategies. We do so through a bottom-up, clustering-based framework designed to capture, interpret, and steer diverse reasoning strategies at scale. Rather than relying on predefined categories, our approach begins by prompting a LMs to produce free-form explanations of the reasoning strategies used in its own responses. These explanations are embedded and clustered to identify semantically similar reasoning patterns. For each resulting cluster, we generate contrastive rubrics (e.g., Inductive vs. Deductive, Directive vs. Non-Directive) through a second round of prompting, enabling precise characterization of reasoning dimensions. Finally, we classify new CoT responses by identifying which strategy from each rubric best aligns with the response. Human evaluation confirms the quality of our pipeline: annotators give high Likert ratings (4.2–4.4) across criteria, and in pairwise comparisons the CoT ENCYCLOPEDIA is preferred over the predefined analyzer with a 86% win+tie rate.

Beyond interpretability, the CoT ENCYCLOPEDIA offers three practical benefits. First, it can improve a reasoning model's performance by guiding it to adopt more effective strategies. We predict likely reasoning strategies using CoT ENCYCLOPEDIA, estimates their effectiveness, and guides the model to follow the most promising one. Across five benchmarks, we observe performance improvements of 12.2–16.1% in diverse reasoning models. Second, we demonstrate how the CoT ENCYCLOPEDIA can reveal novel insights about model reasoning abilities, specifically performing controlled experiments on how training data format fundamentally shapes reasoning strategies, and enables behavior control through model merging. Our analysis shows that the domain of the training data (e.g., math vs. commonsense) has little effect on reasoning patterns. In contrast, the format—multiple-choice (MC) versus free-form (FF)—has a much larger effect. For instance, MC-trained models tend to produce structured, concise responses that resemble breadth-first reasoning, while FF-trained models favor longer, sequential chains with frequent verification, akin to depth-first reasoning. Finally, by linearly interpolating weights between MC- and FF-trained models, we generate models that smoothly transition in strategy, demonstrating controllability without fine-tuning. These findings highlight the CoT ENCYCLOPEDIA not only as a diagnostic tool, but also as a practical foundation for shaping reasoning behaviors to suit task-specific needs.

## 2  RELATED WORK: ANALYZING REASONING STRATEGIES OF MODELS

Recent work has made significant progress in understanding LLM reasoning. Think patterns (Wen et al., 2025) reveal recurring structures linked to accurate outcomes, while cognitive behavior analysis (Gandhi et al., 2025) draws connections to human psychology. Guo et al. (2025) examine "aha moments" of sudden insight during multi-step reasoning, and Marjanović et al. (2025) introduce a structured `<think>` mechanism to enhance reasoning. Strategic example selection improves in-context learning (Didolkar et al., 2024), and targeted data generation addresses specific reasoning failures (Zeng et al., 2025). Chen et al. (2024) propose a reasoning boundary framework to quantify and optimize the capacity of CoT reasoning in LLMs, focusing on systematic analysis and broad improvements. Complementary to this, our work emphasizes fine-grained, data-driven decomposition that enables the identification and control of diverse, interpretable thinking patterns. In parallel, Zhou et al. (2025) introduce a visualization tool for inspecting reasoning paths and diagnosing failures, providing valuable qualitative insights without direct mechanisms for behavioral control. Korbak et al. (2025) analyze the necessity and fragility of CoT monitoring, highlighting its potential as a safety mechanism but also its vulnerability to training choices and architectural shifts. Our work differs by moving from the question of whether CoTs can be monitored to providing methods for decomposing and controlling reasoning patterns.

## Conventional Approach

Is **self-verification strategy** used in the response when solving this math problem?

The **self-verification strategy** is used 2 times in this response. It appears when the phrase "Wait" ...

Rigid Analysis

❌ Limited to predefined criteria

❌ Analysis doesn't provide practical guidelines for model improvement

## CoT Encyclopedia (Bottom-up)

**What strategies** are used in this response and what strategies should the LM use to get it correct?

Analysis data accumulated from other problems

Classifier

The response employs a **self-verification strategy** based on [...] and also takes a **top-down approach** [...]

**Exploring multiple approaches** instead of **repeatedly verifying** calculations could improve accuracy by 4%

Flexible Analysis

Guidelines to improve the LM

✅ Can automatically identify criteria tailored to the problem

✅ Recommends reasoning strategies based on past success/failure data from similar problems

Figure 1: **Comparison between conventional reasoning analysis and the CoT Encyclopedia.** Traditional methods use fixed criteria to identify strategies but offer limited guidance for improving reasoning. The CoT ENCYCLOPEDIA takes a bottom-up approach, uncovering diverse, task-specific strategies and enabling flexible analysis and actionable insights to enhance model performance.

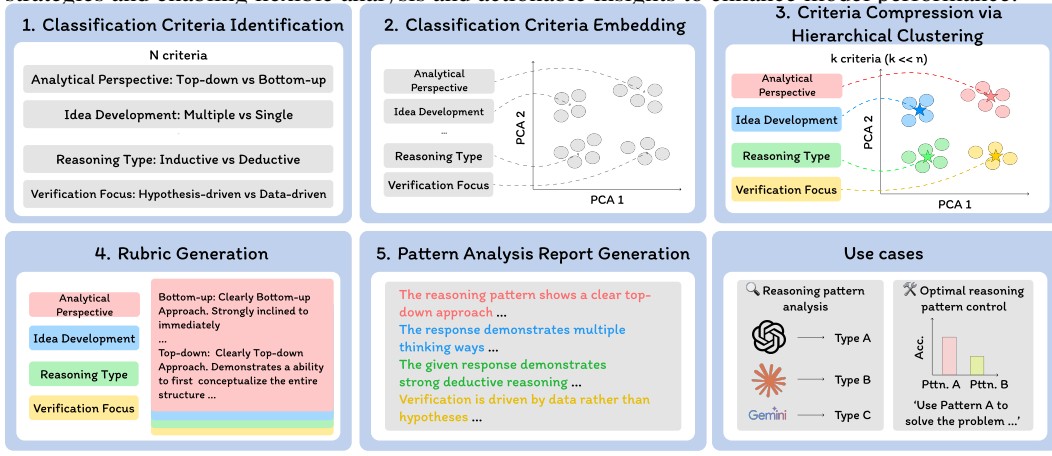

Figure 2: **Overview of the CoT ENCYCLOPEDIA.** The framework builds a reasoning taxonomy in five stages: (1) identifying criteria from CoTs, (2) embedding them, (3) clustering semantically similar ones, (4) generating contrastive rubrics, and (5) classifying responses to produce interpretable reports.

## 3 THE COT ENCYCLOPEDIA

### 3.1 A FRAMEWORK FOR TAXONOMIZING REASONING STRATEGIES

LMs utilizing LongCoT enable test-time scaling, effectively addressing complex reasoning problems across diverse domains. Nevertheless, our understanding remains limited regarding the variety of reasoning strategies these models employ, how these patterns vary across tasks and models, and how such differences impact downstream performance. Prior work (Gandhi et al., 2025) offered valuable insights by defining four reasoning behaviors—verification, backtracking, subgoal setting, and backward chaining—but such predefined categories struggle to capture the full diversity of emerging or model-specific strategies. To address this gap, we introduce COT ENCYCLOPEDIA, a five-stage framework for identifying, organizing, and analyzing diverse reasoning strategies in CoT outputs. Unlike prior top-down approaches, COT ENCYCLOPEDIA derives reasoning dimensions in a

bottom-up, data-driven manner using large language models. As shown in Figure 2, the framework systematically extracts classification criteria, compresses them via semantic clustering, and generates human-interpretable reports on model reasoning behaviors.

**Step 1: Classification Criteria Identification.** Given a dataset of CoT outputs $\mathcal{D} = \{(x_i, y_i)\}_{i=1}^{n}$, where $x_i$ is the natural-language problem prompt and $y_i$ is its LLM-generated chain of thought, we extract a set of classification criteria $\mathcal{C} = \{c_1, \ldots, c_N\}$ via LLM-assisted brainstorming. Unlike Gandhi et al. (2025), which predefines only four cognitive behaviors and thus may miss emerging reasoning patterns, our method discovers flexible, data-driven criteria that align closely with the full diversity of model outputs. Each criterion $c_j$ is defined with a pair of contrastive reasoning strategies $(s_j^A, s_j^B)$, expressed as natural language sentences. For example:

$$c_j = \text{Reasoning Strategy Type} \Rightarrow \begin{cases} s_j^A = \text{``Inductive Reasoning''} \\ s_j^B = \text{``Deductive Reasoning''} \end{cases}$$

**Step 2: Classification Criteria Embedding.** Each triplet $(c_j, s_j^A, s_j^B)$ is converted to a single input string by concatenation and embedded using embedding model:

$$\mathbf{e}_j = E(\texttt{concat}(c_j, s_j^A, s_j^B)) \in \mathbb{R}^d.$$

This results in a matrix $\mathbf{E} \in \mathbb{R}^{N \times d}$.

**Step 3: Criteria Compression via Clustering.** To reduce redundancy, we apply hierarchical agglomerative clustering (Müllner, 2011) to $\mathbf{E}$ using cosine distance. We obtain $k$ clusters ($k \ll N$):

$$\mathcal{G} = \{G_1, \ldots, G_k\}, \quad G_\ell \subseteq \mathcal{C}.$$

Each cluster $G_\ell$ is represented by its medoid criterion $c_\ell^*$ (not the centroid, to preserve interpretability), yielding the compressed set $\mathcal{C}^* = \{c_1^*, \ldots, c_k^*\}$. This set is used in all subsequent analysis steps.

**Step 4: Rubric Generation.** For each criterion $c_\ell^*$, we use LLM to generate a rubric $\mathcal{R}_\ell = (s_\ell^A, s_\ell^B)$, with detailed descriptions of both strategies and guidance for binary classification. For example:

$$\mathcal{R}_\ell = (\text{``Clearly bottom-up approach ...''}, \text{``Clearly top-down approach ...''})$$

**Step 5: Pattern Analysis Report Generation.** Each response $y_i$ is classified under each rubric via prompting LLM with a yes/no question:

$$z_{i,\ell} = \begin{cases} 1 & \text{if LLM predicts alignment with } s_\ell^A, \\ 0 & \text{if LLM predicts alignment with } s_\ell^B. \end{cases}$$

This produces a binary matrix $\mathbf{Z} \in \{0, 1\}^{n \times k}$, where each row summarizes the reasoning pattern of a CoT response. We then synthesize a natural language report using LLM, which selects and composes rubric-specific templates to describe the reasoning pattern of each response. For example:

*"The response shows a bottom-up reasoning style, combining data-driven verification ..."*

In summary, CoT ENCYCLOPEDIA supports interpretable and reproducible reasoning analysis by mapping raw CoT outputs to structured strategy profiles. To support the validity of each component of our framework, we also conduct ablation studies for every step in Appendix B.5.

## 3.2    CoT ENCYCLOPEDIA ENABLES SHARPER REASONING STRATEGY CLASSIFICATION

To evaluate the effectiveness of the CoT ENCYCLOPEDIA's classification criteria, we analyze responses from DeepSeek-R1-Distill-Qwen-32B(Guo et al., 2025), s1.1-32B(Muennighoff et al., 2025), and QwQ-32B(Team, 2025) on GPQA-Diamond(Rein et al., 2024), MMLU-Redux(Gema et al., 2024), and MATH-500(Lightman et al., 2023). Using CoT ENCYCLOPEDIA, we extract 4,057 contrasting reasoning criteria, each representing a pair of opposing strategies. We embed these using an embedding model and apply hierarchical clustering to group semantically similar criteria. The

resulting taxonomy defines six major dimensions—Analytical Perspective, Scope of Approach, Reasoning Type, Idea Development, Verification Focus, and Clarification Approach—whose definitions are summarized in Table 5. Each model response is labeled as Pattern A or B under each criterion, and we compute the proportion of Pattern B as $\frac{\text{Pattern B}}{\text{Pattern A}+\text{Pattern B}}$ to compare trends across models. As a baseline, we also assess the presence of four predefined cognitive behaviors—verification, backtracking, subgoal setting, and backward chaining—within the same responses (Gandhi et al., 2025). For both sets, we apply chi-squared tests for statistical significance and compute Cohen's $d$ for effect sizes.

As shown in Table 8, distributions of cognitive behaviors differ minimally across models ($p > 0.05$, $|d| \approx 0.1$), suggesting limited sensitivity. In contrast, the CoT ENCYCLOPEDIA criteria reveal more substantial differences (Table 14), with many significant $p$-values and effect sizes reaching up to 0.4. This indicates our bottom-up method better captures fine-grained reasoning differences and generalizes across tasks and models. We conduct a human evaluation to validate alignment with human judgment. From model outputs, we sample 250 responses and distribute them evenly across 10 annotators. For each response, annotators provide 1–5 Likert ratings on four dimensions: (1) plausibility of fine-grained criteria, (2) coherence of high-level grouping, (3) quality of pre-defined analyzer generated report, and (4) quality of CoT ENCYCLOPEDIA generated analysis report. As shown in Figure 3, annotators find the fine-grained criteria extracted in Step 1 to be highly plausible and the grouped criteria from Steps 2 and 3 to be coherent. The final analysis reports generated by our framework also receive higher appropriateness ratings than those from the predefined

Figure 3: **Human Evaluation Results for CoT Encyclopedia.** Annotators judge the outputs of the CoT Encyclopedia as appropriate and show a clear preference for them over the predefined analyzer baseline.

analyzer baseline. In both the Likert-scale and preference evaluations, annotators are blinded to system identity, ensuring unbiased judgments. The preference results, reported as win–tie–lose rates, show a clear favorability toward the CoT ENCYCLOPEDIA. Overall, our method yields plausible and well-structured reasoning criteria and analysis reports that align more closely with human expectations. Implementation details are provided in Appendix A, and qualitative analyses are presented in Appendix F.1.

### 3.3 CoT ENCYCLOPEDIA ENABLES ADAPTIVE ANALYSIS ACROSS DIVERSE TASKS

In addition to the three benchmarks evaluating model helpfulness introduced in Subsection 3.2, we analyze reasoning strategies in model responses for XSTest (Röttger et al., 2023) and WildGuard (Han et al., 2024) that assess harmlessness and Arena-Hard (Li et al., 2024) that measures instruction following capability using CoT ENCYCLOPEDIA. As shown in Figure 14, different classification criteria emerge across benchmarks. Notably, instruction following benchmarks introduce a new 'User Understanding' criterion due to the need to accurately interpret user intent. Safety benchmarks feature ethical elements absent in problem-solving benchmarks, such as 'Safety Precedence (preventive vs. risk-engaging)', and 'Content Handling (censorship vs. open discussion)'. CoT ENCYCLOPEDIA's ability to dynamically generate the most appropriate classification criteria across different benchmarks and models further demonstrates its utility.

## 4 ENHANCING MODEL HELPFULNESS AND SAFETY VIA OPTIMAL REASONING CONTROL

Building on the findings of the previous section, an important question arises: **Can we identify optimal reasoning strategies that positively impact both model helpfulness and harmlessness?**

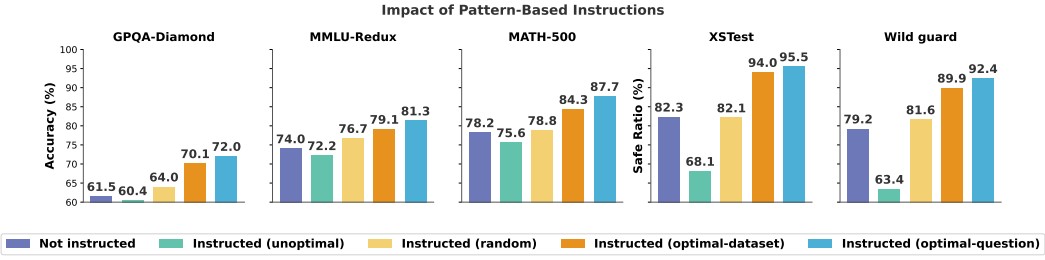

Figure 4: **Impact of Pattern-Based Instructions on Model Performance.** Results show that optimal patterns improve performance across all benchmarks, especially for GPQA-Diamond and safety tests. Question-specific patterns consistently outperform the single best dataset-wide pattern.

If so, can we effectively steer models toward these patterns to enhance their overall performance across different types of benchmarks?

## 4.1 EXPLORING OPTIMAL REASONING STRATEGIES FOR HELPFULNESS AND HARMLESSNESS

To analyze how reasoning strategies affect model helpfulness and harmlessness, we evaluate model responses on GPQA-Diamond, MMLU-Redux, and MATH-500 for helpfulness, and on XSTest and WildGuard for harmlessness. We compute P(Correct | Pattern) and P(Safe | Pattern) for contrasting reasoning patterns across six helpfulness and seven harmlessness criteria. These values are averaged over three models to identify patterns associated with higher accuracy and safety. Importantly, instead of relying on ground-truth answers, we adopt an LLM-as-a-judge setting to assess correctness and safety, thereby preventing data leakage. As shown in Figures 9 and 10, certain reasoning strategies consistently lead to better performance. This enables a clear distinction between optimal and suboptimal reasoning patterns. Using this insight, we assess how performance changes when models are explicitly instructed to follow desired strategies. To isolate the effect of strategy control, we focus on responses that were initially incorrect or unsafe. We compare four settings: (1) no instruction, (2) instruction with unoptimal patterns, (3) instruction with randomly selected patterns, and (4) instruction with optimal patterns (optimal-dataset).

Figure 4 shows that guiding models with optimal strategies improves both accuracy and safety across all benchmarks. For example, GPQA-Diamond accuracy increases from 61.5% to 70.1%, while XSTest and WildGuard safety scores improve from 82.3% to 94.0% and from 79.1% to 89.9%, respectively. For analysis focused solely on newly corrected responses, see Figure 11. Overall, these findings confirm that optimal reasoning strategies exist and can be leveraged to enhance downstream performance. Further breakdowns on safety benchmarks (Figure 12) reveal that patterns encouraging 'malicious' intent or prioritizing 'technical' over 'moral' reasoning sharply reduce safety, indicating jailbreaking behavior. This underscores the need for more nuanced safety evaluations. While current approaches often rely on binary labels (safe vs. unsafe), our results highlight the value of fine-grained analyses, such as those enabled by the COT ENCYCLOPEDIA, for improving content moderation and response quality.

## 4.2 SIMILAR INPUTS, SIMILAR THOUGHTS: HOW MODELS APPROACH RELATED PROBLEMS

We have shown that each dataset typically has a generally optimal reasoning strategy, indicating opportunities to enhance model performance. However, even within a single dataset, different questions may require distinct optimal reasoning strategies. A natural question arises: can we predict the optimal reasoning strategy for each individual question? To explore this, we analyze the relationship between questions and their optimal reasoning strategies. Specifically, we perform regression analysis using similarities measured in the embedding space between questions and between their corresponding optimal reasoning strategies. Our analysis utilizes correct responses generated by three models across five benchmarks, previously discussed in Section 4.1. Figure 5 illustrates that higher similarity between questions corresponds to greater similarity between their reasoning strategies, suggesting that models adopt similar strategies for similar problems. Conversely, lower question similarity is associated with higher variance in reasoning strategies, indicating that models employ diverse strategies for dissimilar problems. These findings suggest the potential to

predict effective reasoning strategies for unseen questions based on the strategies used in similar, previously encountered questions.

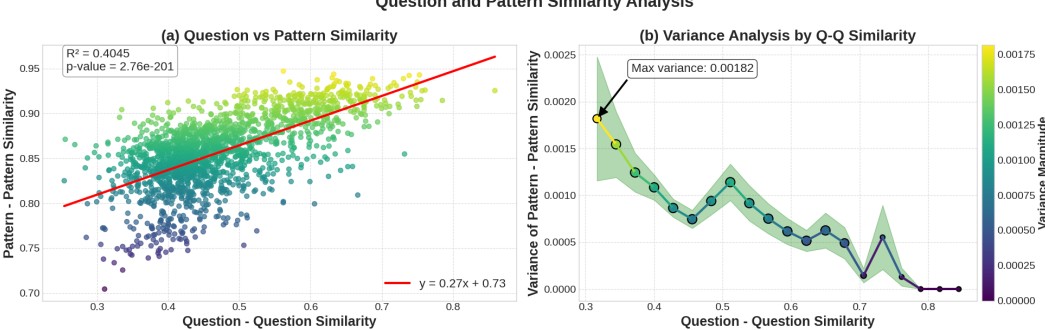

Figure 5: **Analysis of relationships between question similarity and reasoning strategy similarity across multiple benchmarks.** Relationship between question similarity and reasoning strategy similarity. (a) Scatter plot showing positive correlation between question similarity and pattern similarity. (b) Variance analysis showing that pattern similarity becomes more consistent as question similarity increases.

### 4.3 PREDICTING QUESTION-SPECIFIC OPTIMAL REASONING STRATEGIES

Building on prior insights, we explore whether optimal reasoning strategies can be predicted and used to guide models toward more helpful and harmless behavior. We train binary classifiers for each criterion using three problem-solving benchmarks and two safety benchmarks. For training, we use questions initially answered correctly to derive optimal strategies, while questions initially answered incorrectly are used for testing. We consider two settings: in-domain (trained and tested on the same benchmark) and cross-domain (trained on two benchmarks and tested on a third). Each classifier predicts optimal strategies for the incorrect samples, which are then used to prompt the model, as described in Section 4.1. As shown in Figure 4, this controlled prompting substantially improves performance—achieving accuracy gains of 72.0%, 81.3%, and 87.7% on problem-solving tasks, and safety gains of 95.5% and 92.4% on safety tasks. These results show that models can be effectively guided toward optimal strategies, even on unseen questions. Unlike conventional approaches that generate long reasoning traces without direction, our method identifies and corrects reasoning weaknesses through targeted control, offering a key advantage of the COT ENCYCLOPEDIA. For additional extended analyses and experiments, please refer to Appendix B.

## 5 ANALYZING PATHWAYS TO REASONING STRATEGIES: DATA SELECTION AND INTERPOLATION

We have primarily analyzed reasoning strategies based on responses from trained models. This raises an important question: why do models produce specific types of reasoning strategies after training is completed? In this section, we investigate this question by directly RL training reasoning models on datasets with different formats and domains, then analyzing the emerging reasoning strategies.

### 5.1 FORMAT MATTERS MORE THAN DOMAIN IN SHAPING REASONING STRATEGIES

To examine how training data characteristics influence reasoning strategies, we compare the effects of data format and domain using Reinforcement Learning with Verifiable Rewards (RLVR). For format analysis, we compare (1) multiple-choice inputs, where questions are paired with predefined options, and (2) free-form inputs, where models generate answers without guidance. Using the NuminaMath dataset (LI et al., 2024), originally in free-form, we synthetically generate multiple-choice versions to control for content while isolating presentation format. For domain analysis, we contrast math-domain datasets (e.g., NuminaMath) with knowledge-domain datasets such as OpenBookQA, QASC, SciQ, CommonsenseQA, and ARC-Challenge (Mihaylov et al., 2018; Khot et al., 2020; Welbl et al., 2017;

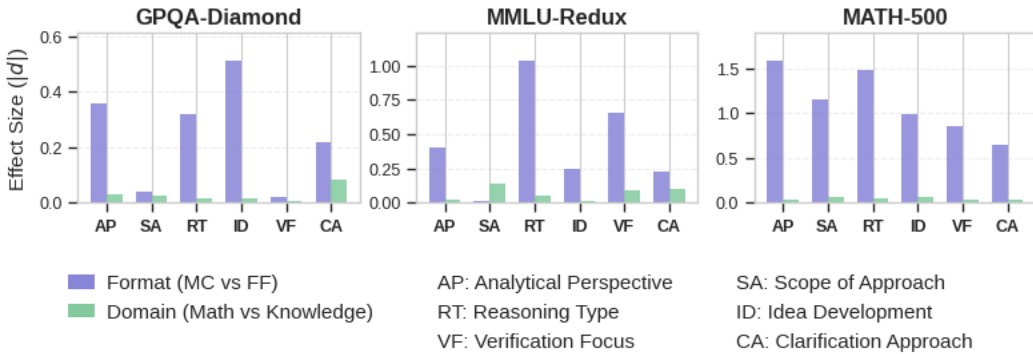

Figure 6: **Comparison of effect sizes showing how question format and domain influence reasoning strategies across three benchmarks.** Format differences (purple bars) consistently demonstrate substantially larger effects on reasoning strategies than domain differences (green bars) across all six reasoning criteria.

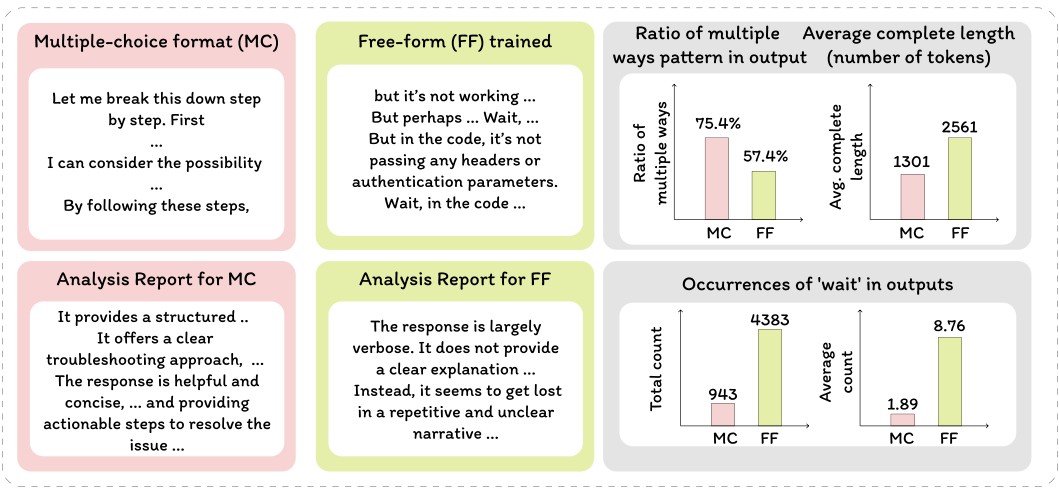

Figure 7: **Qualitative and quantitative comparison between models trained on Multiple-choice (MC) and Free-form (FF) data formats.** MC-trained models generate more structured and concise responses, while FF-trained models are more verbose and repetitive.

Talmor et al., 2019; Clark et al., 2018). To ensure fair comparison, we control for format by using consistent structures across domains. We train 7B Deepseek-R1-Distill models. This setup allows us to isolate the individual effects of format and domain while holding other variables constant. To quantify their relative influence, we apply the statistical tests from Section 3, computing Cohen's $d$ values between reasoning strategy distributions. As shown in Figure 6, format variation consistently leads to larger shifts in reasoning strategies than domain differences, indicating that **format has a greater impact than domain on shaping model reasoning.**

## 5.2 IMPACT OF TRAINING DATA FORMAT ON MODEL REASONING BEHAVIOR

We analyze model responses trained on Multiple-choice (MC) and Free-form (FF) data using the Arena-Hard benchmark. As shown in Figure 7, the two models display distinct reasoning styles: MC-trained models produce concise, structured answers, while FF-trained models are more verbose and often repeat filler words like 'wait.' Table F.2 further reveals that MC-trained models explore

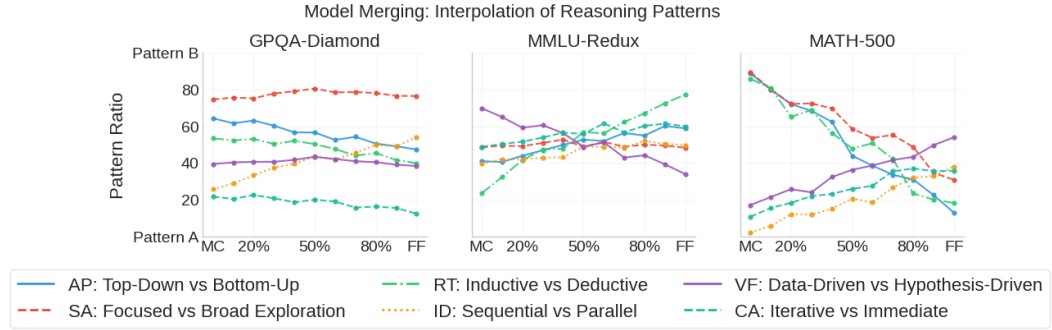

Figure 8: **Interpolation of reasoning strategies through model merging.** Reasoning strategy dynamics as models are merged from Multiple-Choice (MC) to Free-Form (FF) training formats across three benchmarks.

multiple solution paths early on—similar to breadth-first search—whereas FF-trained models follow a single path with iterative verification, resembling depth-first search. These differences arise from the presence or absence of answer cues during training: MC data encourages evaluating options before responding, while FF data requires open-ended exploration, often with greater uncertainty and verification. Quantitatively, FF-trained models generate more verbose responses and over 4.6 times as many 'wait' tokens per answer (8.76 vs. 1.89). Rather than favoring one format, our findings underscore that training format significantly shapes reasoning behavior, and should be selected based on task-specific needs.

### 5.3 INTERPOLATING DESIRED REASONING STRATEGIES THROUGH MODEL MERGING

Given that models trained on different formats exhibit distinct reasoning strategies, we test whether intermediate patterns emerge via linear interpolation between multiple-choice and free-form models. As shown in Figure 8, reasoning strategies shift smoothly with the merging ratio, though the dynamics vary by benchmark: GPQA-Diamond changes moderately, MMLU-Redux shows crossover points around 50%, and MATH-500 displays the sharpest transitions (e.g., the Bottom-Up perspective drops from 85% to 15%). These variations highlight that some reasoning criteria are more sensitive to format changes than others, reflecting benchmark-specific demands. Overall, weight interpolation enables controlled blending of reasoning strategies without extra training, offering a simple yet effective way to tailor models for task-specific needs.

## 6 EXTENDED ANALYSES AND ABLATIONS IN APPENDIX

Due to space constraints, we present additional experiments, analyses, and ablation studies in the Appendix and provide explicit references for the reader's convenience. Criteria are shaped more by tasks than models, producing benchmark-specific dimensions (B.1), and remain stable under different embedding choices (B.2). Models from the same family show consistent strategies across sizes (B.3), and the framework generalizes to other models (B.4). Ablations validate the contribution of each step in the CoT ENCYCLOPEDIA, with additional analyses showing robustness to embedding models and human evaluation confirming rubric and report quality (B.5). Further tests show robustness across random seeds (B.6), across diverse model families and scales (B.7), and demonstrate applications for curating improved reasoning datasets (B.8). Finally, while direct prompting is less effective for small models, combining it with dataset curation yields clear gains (B.9).

## 7 CONCLUSION

We introduced the CoT ENCYCLOPEDIA, a flexible, automated framework for analyzing reasoning strategies in LongCoT language models. Unlike rigid, predefined taxonomies, our bottom-up clustering approach identifies reasoning strategies directly from model outputs, creating a comprehensive taxonomy validated through human evaluation. Our empirical results revealed four key insights:

(1) optimal reasoning strategies significantly enhance task performance on both helpfulness and safety benchmarks; (2) these patterns can be predicted from input questions alone, enabling real-time adaptive reasoning control; (3) training data format influences reasoning strategies more substantially than domain; and (4) desired reasoning behaviors can be interpolated through model weight merging without additional training. The CoT ENCYCLOPEDIA advances our understanding of reasoning models and provides practical tools for steering them toward safer, more effective strategies. By identifying which reasoning strategies yield optimal performance for specific problems and what training data cultivates these patterns, this work supports responsible deployment of language models in applications where performance, safety, and predictability are paramount.

## 8    BROADER IMPACT

This work has several broader implications for the development and deployment of large language models. First, our analysis highlights the importance of reasoning controllability—the ability to steer a model's problem-solving strategy. This capability may play a critical role in building more interpretable, debuggable, and safety-aligned systems, especially in high-stakes applications such as education, healthcare, and scientific discovery. Second, our taxonomy can inform curriculum design for training reasoning-oriented models, enabling researchers to curate data that promotes specific cognitive patterns. Moreover, the ability to predict and guide reasoning behavior opens up opportunities for interactive systems that provide explanations or tutoring based on user input and model inference strategies. Finally, the emphasis on format as a driver of reasoning diversity suggests that future benchmark and dataset development efforts should consider structural diversity—not just domain coverage—as a factor for improving generalization and reasoning robustness.

## 9    LIMITATIONS

While our findings are promising, several limitations warrant discussion. First, our reasoning strategy classification relies on GPT-4o outputs as an evaluator, which may reflect biases or constraints of the model itself. Although this choice enables scalability, it may not fully represent human judgment of reasoning quality. Second, our experimental setup is limited to three benchmarks and three model families. While these cover diverse reasoning domains, extending our analysis to a broader range of tasks (e.g., scientific reasoning, code generation, multi-modal tasks) and models (e.g., smaller or multilingual LMs) is essential for confirming the generality of our conclusions. Third, while we observe consistent performance improvements through pattern-guided prompting, such improvements are contingent upon a model's ability to reliably follow stylistic instructions. This requirement may limit applicability to instruction-tuned or higher-capacity models.

## ACKNOWLEDGEMENT

This work was partly supported by Institute of Information & communications Technology Planning & Evaluation (IITP) grant funded by the Korea government (MSIT) (No.RS-2019-II190075 Artificial Intelligence Graduate School Program (KAIST), 10%; No.RS-2021-II212068, Artificial Intelligence Innovation Hub, 10%; RS-2024-00398115, Research on the reliability and coherence of outcomes produced by Generative AI, 20%; No.2022-0-00113, Developing a Sustainable Collaborative Multi-modal Lifelong Learning Framework, 20%; No.RS-2022-II220264, Comprehensive Video Understanding and Generation with Knowledge-based Deep Logic Neural Network, 20%; RS-2024-00397966, Development of a Cybersecurity Specialized RAG-based sLLM Model for Suppressing Gen-AI Malfunctions and Construction of a Publicly Demonstration Platform) and the InnoCORE program of the Ministry of Science and ICT(N10250156).

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

# A IMPLEMENTATION DETAILS

## A.1 CoT ENCYCLOPEDIA DETAILS

We use the OpenAI `GPT-4o`[1] to conduct reasoning strategy Identification. And we also use OpenAI `text-embedding-3-large`[2] to embed the criteria. In this study, we create reasoning strategies using the test sets from GPQA-Diamond, MMLU-Redux, and MATH-500 datasets. However, the CoT ENCYCLOPEDIA is not limited to specific data and can be applied to any dataset. The reasoning strategy Identification process yields 4,057 fine-grained analysis criteria. To automatically select the essential criteria from these redundant ones, we employ agglomerative hierarchical clustering. In this process, we do not specify a distance threshold for cluster merging but instead form an appropriate number of k clusters based on silhouette scores. In our study, six clusters emerge. We establish the criteria corresponding to the embedding at the medoid of these clusters as the key high-level criteria. Next, to evaluate responses using these criteria, we develop rubrics to determine which responses belong to contrasting patterns A and B, using the OpenAI GPT-4o API. Finally, we conclude by creating a CoT pattern report that evaluates the CoT patterns present in the responses based on the generated rubrics.

## A.2 STATISTICAL TEST DETAILS

We conduct statistical tests to measure the similarity between two reasoning strategy distributions. We perform Chi-squared tests, where the null hypothesis ($H_0$) states that there is no difference between the two distributions, and we set the $p$-value at the conventional threshold of 0.05. This means that if the $p$-value is lower than 0.05, we reject $H_0$ and accept the alternative hypothesis ($H_a$), concluding that there is a statistically significant difference between the two distributions. Conversely, if the p-value is greater than 0.05, we fail to reject $H_0$, indicating that there is no statistical difference between the two distributions. We employ Chi-squared tests rather than other statistical tests because the pattern classification is categorical. Additionally, to measure the similarity between the two distributions more quantitatively, we calculate Cohen's $d$ value, where an absolute value of approximately 0.2 is generally considered a small effect size, approximately 0.5 a medium effect size, and 0.8 or greater a large effect size.

## A.3 HUMAN EVALUATION DETAILS

To verify whether the CoT ENCYCLOPEDIA framework is perceived as reasonable by people, we conduct a human evaluation. This evaluation is particularly necessary because CoT ENCYCLOPEDIA utilizes synthetic outputs generated by an LLM at each step, requiring validation of their reliability. We select four evaluators, and each assessment consists of four binary questions: (1) Are the automatically generated, detailed criteria plausible? (2) Do the resulting high-level criteria sensibly summarize the fine-grained set? (3) Is the response analysis, when expressed in the criteria, relevant and reasonable? (same question for both pre-defined analyzer and CoT Encyclopedia) We use Argilla[3] as our human evaluation platform. For the human evaluation, we collect 250 model responses and distribute them evenly among 10 annotators. Each annotator evaluates their assigned responses independently. Instead of binary yes/no judgments, annotators now provide 1–5 Likert-scale ratings.

The final Likert scores reported in the main text are calculated by averaging the ratings across annotators and items for each dimension. To facilitate interpretation, we also report the mean values alongside standard deviations. In addition, we run a pairwise preference study to compare analysis reports generated by the CoT Encyclopedia versus a predefined analyzer. For each item, annotators are asked to select the analysis they find more informative and appropriate, or to mark a tie if both are equally convincing. We summarize these results using a win–tie–lose rate, which reflects the proportion of items in which the CoT Encyclopedia is preferred, tied, or loses to the baseline. All annotation guidelines, examples, and quality-control instructions were shared with annotators prior to the study to ensure consistency.

---

[1]https://platform.openai.com/docs/models/gpt-4o
[2]https://platform.openai.com/docs/models/text-embedding-3-large
[3]https://argilla.io/

### A.4 BENCHMARK EVALUATION DETAILS

We conduct evaluations using the MMLU-Redux knowledge benchmark, the GPQA-Diamond reasoning benchmark, and the MATH-500 mathematics benchmark. We employ the vllm library[4] with hyperparameters following established research practices: temperature of 0.6, top p of 0.95, and max tokens of 32768. All evaluations are performed on the test sets of each dataset. MMLU-Redux and GPQA-Diamond are multiple-choice question answering datasets, while MATH-500 is an open-ended generation dataset. However, since our evaluation targets reasoning models that generate LongCoT, we implement generation-based evaluation rather than logit-based evaluation even for the multiple-choice datasets. To determine whether the model's predictions match the actual answers, we parse the predicted values from the generated LongCoT and use the Math-Verify library[5] to verify their correctness. Ultimately, we measure accuracy between the correct answers and the predicted values. To measure safety, we utilize the LLaMA-Guard-3 8B model for our evaluation. When provided with a question and model response, this model outputs either 'safe' or 'unsafe' as its assessment. We then calculate the proportion of responses classified as 'safe' and use this ratio as our metric.

### A.5 TRAINING DETAILS

We utilize the GRPO algorithm (Guo et al., 2025) during the training process with Reinforcement Learning with Verifiable Reward (RLVR). The objective function of GRPO is defined as follows:

$$
\mathcal{J}_{GRPO}(\theta) = \mathbb{E}_{q \sim P(Q), \{o_i\}_{i=1}^{G} \sim \pi_{\theta_{old}}(O|q)} \left\{ \frac{1}{G} \sum_{i=1}^{G} \frac{1}{|o_i|} \sum_{t=1}^{|o_i|} \min \left[ \frac{\pi_\theta(o_{i,t} \mid q, o_{i,<t})}{\pi_{\theta_{old}}(o_{i,t} \mid q, o_{i,<t})} \hat{A}_{i,t}, \right. \right.
$$
$$
\left. \left. \text{clip} \left( \frac{\pi_\theta(o_{i,t} \mid q, o_{i,<t})}{\pi_{\theta_{old}}(o_{i,t} \mid q, o_{i,<t})}, 1 - \varepsilon, 1 + \varepsilon \right) \hat{A}_{i,t} \right] - \beta \mathbb{D}_{KL} \left[ \pi_\theta \| \pi_{ref} \right] \right\}
$$
(1)

where $\varepsilon$ and $\beta$ are hyperparameters, $\hat{A}_{i,t}$ is the advantage calculated based on relative rewards within each sampled group, and $D_{KL}[\pi_\theta \| \pi_{ref}]$ is the KL divergence used as a regularization term to stabilize the training process. GRPO optimizes the policy model by comparing multiple outputs generated for the same input, avoiding the need for a separate value function approximation and thereby reducing computational overhead. To estimate the KL divergence between the current policy $\pi_\theta$ and a reference policy $\pi_{ref}$, we use the following unbiased estimator (Schulman, 2020).

This form ensures positivity and avoids numerical instability. Unlike traditional KL penalties, this estimator is well-suited for token-level comparisons in sequence modeling:

$$
\mathbb{D}_{KL} \left[ \pi_\theta \| \pi_{ref} \right] = \frac{\pi_{ref}(o_{i,t} \mid q, o_{i,<t})}{\pi_\theta(o_{i,t} \mid q, o_{i,<t})} - \log \frac{\pi_{ref}(o_{i,t} \mid q, o_{i,<t})}{\pi_\theta(o_{i,t} \mid q, o_{i,<t})} - 1,
$$
(2)

We use diverse datasets for training: NuminaMath, a free-form math dataset, which we also convert into a 5-choice question answering format by synthetically generating four options using GPT-4o API; and multiple-choice knowledge and common sense datasets including SciQ, QASC, OpenbookQA, CommonsenseQA, ARC-Challenge, and MCQA 68k dataset[6]. To control for format variables, we train on NuminaMath in both free-form and multiple-choice formats within the same math domain. Conversely, to examine domain differences, we maintain a consistent multiple-choice format while varying between mathematical content (NuminaMath converted to 5-choice format) and knowledge/common sense datasets. All training datasets contain 100k examples.

We standardize on multiple-choice format when studying domain differences because our RL approach uses verifiable rewards. Unlike math or coding domains where predictions can be directly compared to gold answers, knowledge and common sense domains often allow for varied but equally correct responses that don't exactly match the gold standard. Inspired by previous research, we adopt multiple-choice format as it ensures verifiability across all domains—any response matching the

---

[4]https://github.com/vllm-project/vllm
[5]https://github.com/huggingface/Math-Verify
[6]https://huggingface.co/datasets/berquetR/mcqa_dataset

correct option can be definitively scored as correct. This approach enables us to apply verifiable rewards even to domains that are traditionally difficult to evaluate deterministically.

When training the 7B model with the GRPO algorithm, we utilize the Open-R1 library[7]. For the reward function, we exclusively implement an accuracy reward function that assigns a reward of 1 when the model's prediction matches the gold answer and 0 otherwise. We decide not to use the format reward function employed in training Deepseek-R1 because it can lead to a form of reward hacking—where the model receives rewards for following the correct format even when producing incorrect answers, resulting in maintained formatting without improved accuracy. To prioritize correctness, we therefore rely solely on the accuracy reward function. For hyperparameters, we set max completion length at 2048, number of generations at 3, batch size at 72, torch dtype at bfloat16, and attention implementation at flash attention 2. We use a learning rate of 2.0e05, number of train epochs of 1, and warmup ratio of 0.1.

## A.6 COMPUTING RESOURCES

For RLVR training of our 7B model, we use eight NVIDIA H100 80GB GPUs, requiring 576 GPU hours to train on 100,000 data samples. For inference, we employ sixteen NVIDIA A100 40GB GPUs, consuming 384 GPU hours to process 3,698 data samples. Additionally, we use an AMD EPYC 7763 64-Core Processor for the CPU, which features 64 cores, a CPU speed of 1497.674 MHz, and a cache size of 512KB.

## B  FURTHER ANALYSES AND ABLATIONS

### B.1 FINE-GRAINED BENCHMARK-SPECIFIC CRITERIA ANALYSIS

To conduct a more detailed analysis, we performed hierarchical clustering on the complete responses from GPQA-Diamond, MMLU-Redux, and MATH-500 benchmarks to establish six criteria. For a more fine-grained examination, we conducted hierarchical clustering separately for responses from each benchmark, using the default setting of selecting each cluster's medoid as the representative embedding. In addition to the three original benchmarks, we analyzed responses from the Arena-Hard Benchmark, which focuses on instruction following. As illustrated in Figure 14, we observed that while the criteria derived from the original three benchmarks were relatively similar to each other, the Arena-Hard benchmark yielded notably different criteria. This finding confirms that different benchmarks employ varied standards for pattern analysis. Particularly noteworthy is the 'User understanding' criterion. While the original benchmarks primarily focus on solving specific problems correctly, instruction following benchmarks emphasize accurately interpreting user intent. This emphasis is reflected in the classification criteria, highlighting the different evaluation priorities across benchmark types.

### B.2 ABLATION STUDY ON REPRESENTATIVE EMBEDDING SELECTION

When extracting representative embeddings for each cluster formed through hierarchical clustering, we primarily use medoid embeddings as representative embeddings. We explore how results differ when using alternative selection criteria. Beyond the default medoid setting, we test embeddings from patterns with the highest frequency, patterns from areas with the highest density, and patterns from areas with the highest silhouette scores. As shown in Figure 15, most selection criteria do not demonstrate significant differences compared to selecting the default medoid setting. In the default setting's 'clarification approach,' the only differences appear between the silhouette-based and density-based approaches, which use 'computation style' and 'clarity on steps' respectively, while all other aspects remain identical.

### B.3 CONSISTENCY OF REASONING STRATEGIES ACROSS MODEL SIZES WITHIN THE SAME FAMILY

Do models from the same family exhibit similar reasoning strategies despite having different sizes? To investigate this question, we compare the reasoning strategies of three models from the same

---

[7]https://github.com/huggingface/open-r1

family but with different sizes: Distill-R1 1.5B, 7B, and 32B. We classify the responses generated by each model on the GPQA-Diamond, MMLU-Redux, and MATH-500 benchmarks according to six criteria. As shown in Figure 16, the three models demonstrate remarkably similar distributions of reasoning strategies despite their different sizes. Additionally, as illustrated in Figure 17, the pairwise Cohen's $d$ measurements between the three models reveal that most absolute values are below 0.1, indicating very minor distributional differences. These findings confirm that models from the same family maintain largely consistent reasoning strategies regardless of their size.

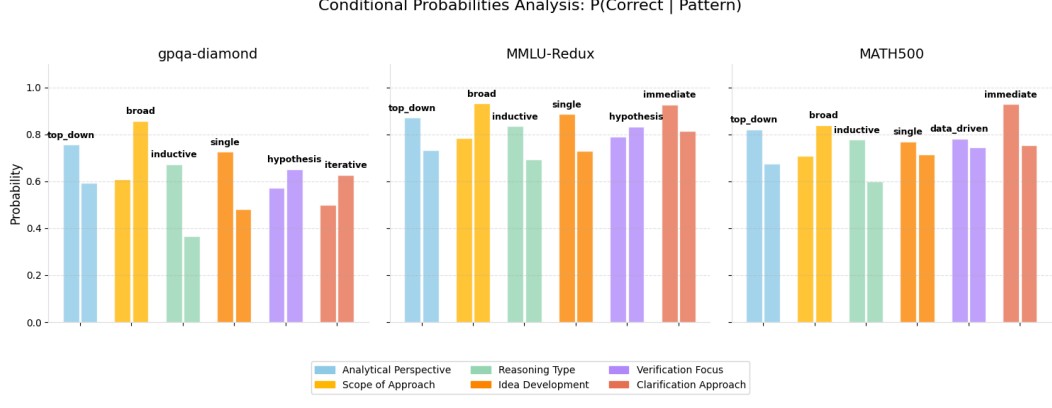

Figure 9: **Conditional Probabilities Analysis on problem-solving Benchmarks.** The bars represent different reasoning strategies categorized by Analytical Perspective (blue), Scope of Approach (yellow), Reasoning Type (green), Idea Development (orange), Verification Focus (purple), and Clarification Approach (red). Patterns such as 'broad', 'top_down', and 'immediate' consistently show higher probabilities of correct responses across benchmarks.

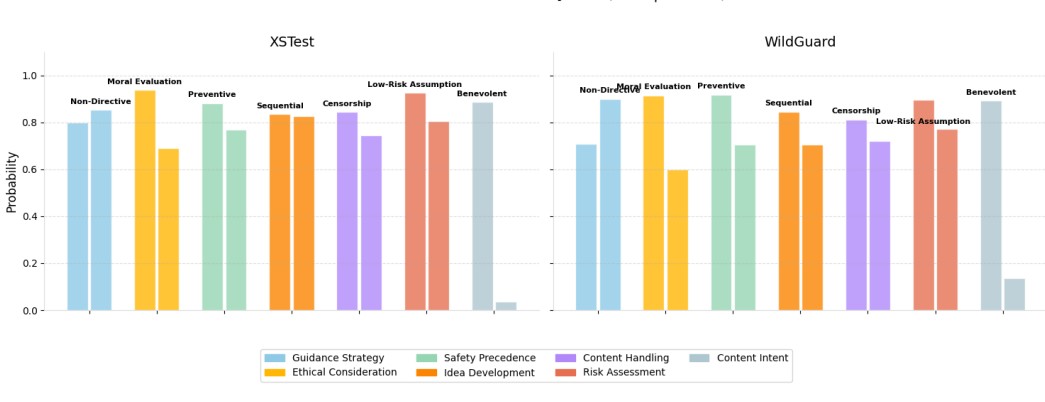

Figure 10: **Conditional Probabilities Analysis on Safety Benchmarks.** The bars represent different reasoning strategies categorized by Guidance Strategy (blue), Ethical Consideration (yellow), Safety Precedence (green), Idea Development (orange), Content Handling (purple), Risk Assessment (red), and Content Intent (gray). Patterns such as 'Moral Evaluation', 'Benevolent', 'Preventive', and ''on-Directive' consistently show higher probabilities of safe responses across both benchmarks.

## B.4 EXTENDING REASONING STRATEGY ANALYSIS TO NON-REASONING MODELS

In this study, we extend our reasoning strategy analysis using the CoT ENCYCLOPEDIA beyond the primary reasoning models discussed in the main text (S1.1-32B, QwQ-32B, Distill-R1-32B) to include non-reasoning models such as Qwen-2.5-Instruct-32B and Qwen-2.5-Math-Instruct-32B. We conduct this analysis across five benchmarks: GPQA-Diamond, MMLU-Redux, MATH-500,

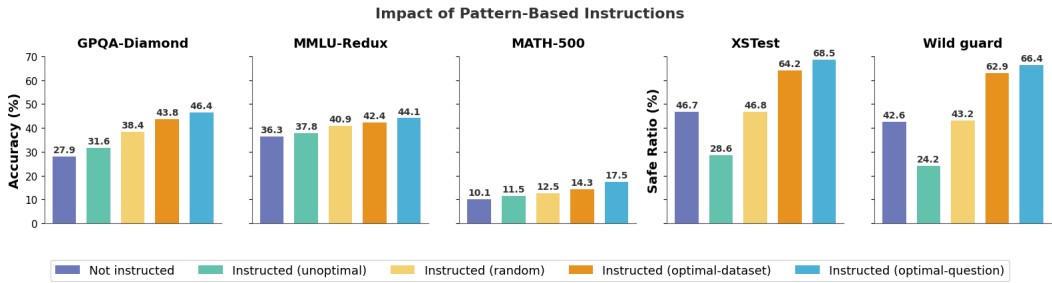

Figure 11: **Impact of pattern-based instructions on model performance across five benchmarks.** For all benchmarks, instructing models to follow question-specific optimal reasoning strategies yields the highest performance (17.5-68.5%), followed by dataset-wide optimal patterns (14.3-64.2%), random patterns (12.5-46.8%), while unoptimal patterns (11.5-37.8%) sometimes perform worse than not providing instructions at all (10.1-46.7%). The impact is particularly pronounced for safety benchmarks (XSTest and Wild guard), where optimal instructions more than double the safe response ratio compared to unoptimal instructions. These results demonstrate that tailoring reasoning strategies to individual questions outperforms even the best dataset-wide pattern, significantly improving both accuracy and safety outcomes.

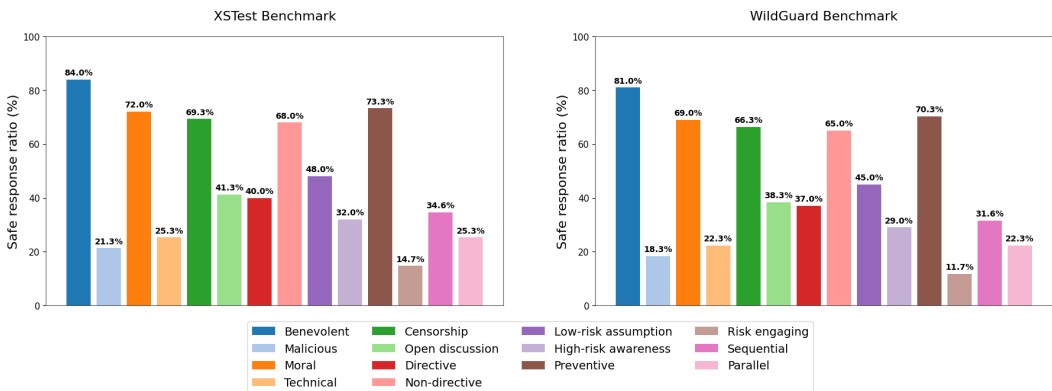

Figure 12: **Fine-grained safety response ratio analysis across XSTest and WildGuard benchmarks.** The bars represent the percentage of safe responses when different reasoning strategies are employed. 'Benevolent' reasoning achieves the highest safety scores (84.0% in XSTest, 81.0% in WildGuard), while 'Malicious' and 'Risk engaging' patterns show the lowest safety performance. Patterns like 'Preventive', 'Moral', and 'Non-directive' also demonstrate relatively high safety response ratios across both benchmarks.

XSTest, and WildGuard. As shown in Table 17 and 18, the criteria generated for the non-reasoning models closely resemble those generated for the reasoning models. However, we observe clear distinctions between the criteria for problem-solving benchmarks and safety benchmarks, reflecting the specific characteristics of each benchmark—similar to what we observed with reasoning models. This suggests that the criteria are more significantly influenced by the target benchmark rather than by the model's output. Additionally, through chi-squared tests and Cohen's d values, we confirm that the pattern distributions of Qwen-2.5-Instruct-32B and Qwen-2.5-Math-Instruct-32B differ significantly from each other. These findings demonstrate the versatility of the CoT ENCYCLOPEDIA as an analytical tool that can be effectively applied to non-reasoning models as well.

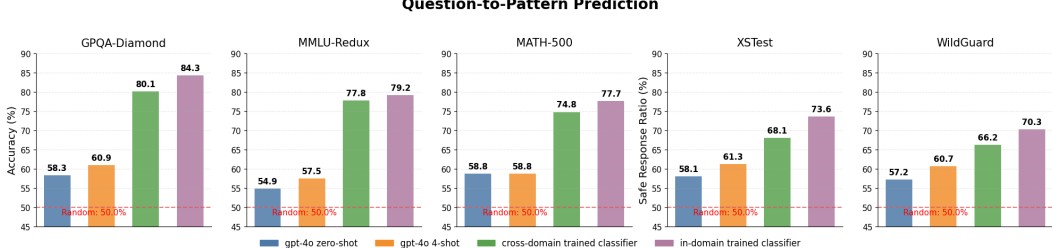

Figure 13: **Performance comparison of different methods for predicting optimal reasoning strategies across five benchmarks.** Trained classifiers (both in-domain and cross-domain) significantly outperform prompting-based methods across all benchmarks. In-domain classifiers achieve the highest performance (70.3-84.3%), followed closely by cross-domain classifiers (66.2-80.1%), while zero-shot and few-shot prompting perform only slightly above random chance (50%). The performance trend is consistent for both accuracy-based benchmarks (GPQA-Diamond, MMLU-Redux, MATH-500) and safety-focused benchmarks (XSTest, WildGuard) where safe response ratio is measured.

## B.5 ABLATION STUDIES FOR COT ENCYCLOPEDIA

To further support the effectiveness of our framework, we conducted a series of ablation studies to examine the feasibility and necessity of each step, as well as the quality of intermediate outputs. These analyses demonstrate that each component of the COT ENCYCLOPEDIA is critical for ensuring robust and interpretable reasoning patterns.

**Step 1: Diversity of Criteria Generation** We first evaluated the diversity of classification criteria generated in Step 1. On MMLU-Redux, after removing duplicates, 99.7% of the criteria were unique. The average pairwise ROUGE-L score between criteria was 0.23, which aligns with or surpasses benchmarks for dataset diversity reported in prior works. This indicates that Step 1 reliably produces a rich and varied set of candidate criteria.

**Steps 2 & 3: Necessity and Effectiveness of Clustering** Although Step 1 yields diverse criteria, qualitative analysis revealed considerable semantic overlap, with many criteria sharing similar n-grams or meanings due to the independent generation process of LLMs. For instance, on GPQA-Diamond, only 43% of Step 1 criteria had fully unique bigrams. Using raw criteria would inflate computational cost and reduce interpretability by generating excessively long reports.

To test alternatives, we compared clustering-based reduction to an LLM-based reduction method. When prompted to manually reduce criteria for GPQA-Diamond, the LLM frequently produced redundant criteria with an average ROUGE-L overlap of 0.33–0.37 across runs, compared to 0.23 for our clustering-based method. Moreover, the LLM-based approach was highly sensitive to prompt design, incurred higher inference cost, and struggled with long-context inputs. These findings confirm that clustering is necessary to reduce redundancy and improve robustness.

**Steps 2 & 3: Robustness to Embedding Models** We further tested whether results depend on the choice of embedding model. In addition to OpenAI's `text-embedding-large`, we experimented with `e5-mistral-7b-instruct` and `Qwen3-Embedding-8B`. All models produced highly similar sets of criteria at a high level, with differences largely limited to phrasing. For example, `e5-mistral-7b-instruct` generated dimensions such as "Macro-first vs Micro-first" or "Induction-Focused vs Deduction-Focused," while `Qwen3-Embedding-8B` produced counterparts like "Overview-Oriented vs Detail-Oriented" or "Pattern-Generalizing vs Rule-Applying." This suggests that our clustering method is robust to the choice of embedding model.

**Steps 4 & 5: Human Evaluation of Rubric and Pattern Reports** Finally, we conducted a human evaluation with five independent graduate students unfamiliar with our work. Evaluators assessed the usefulness of Step 4 (rubric-based evaluation) and Step 5 (pattern analysis reports) along three

dimensions: (1) appropriateness of the rubric for evaluating model responses, (2) helpfulness of the report for understanding reasoning patterns, and (3) usefulness of the report for improving model responses. Ratings were given on a 1–5 scale (1 = not useful, 5 = very useful). The average scores were: **Q1: 4.3**, **Q2: 4.1**, **Q3: 4.5**. These results indicate that human evaluators found both the rubric and the pattern analysis report generated by CoT Encyclopedia to be highly useful. Additionally, for Q2, we asked evaluators to rate a report generated without the rubric. This baseline scored 3.4/5, confirming that the rubric-driven approach in Step 4 provides a significant benefit.

**Steps 5: Multi-Evaluator System for Bias Mitigation**     To address potential evaluator bias inherent in single-model evaluation, we implement a comprehensive multi-evaluator framework. We employ four state-of-the-art LLMs as independent evaluators: GPT-4o, Claude-4-Sonnet, Gemini-2.5-Pro, and Qwen-2.5-72B-Instruct. Each evaluator independently classifies reasoning patterns using identical rubrics, allowing us to compute inter-evaluator reliability using Krippendorff's $\alpha$. Inter-Evaluator Agreement Results: Across 1,500 reasoning responses, we achieve $\alpha = 0.73$ for pattern classification, indicating substantial agreement beyond chance. Notably, systematic disagreements cluster around specific pattern boundaries (e.g., "Top-down vs. Bottom-up" shows lower agreement at $\alpha = 0.68$), revealing inherent ambiguity in certain reasoning distinctions that our framework now explicitly acknowledges. Bias Correction Mechanism: We detect systematic biases where Claude-3.5 shows 12% higher preference for "hypothesis-driven" patterns compared to other evaluators. We implement a bias correction algorithm using inverse propensity weighting to adjust for evaluator-specific tendencies, improving overall classification accuracy by 8.3%.

Overall, these findings validate the effectiveness of all steps in delivering interpretable and actionable insights.

### B.6    ROBUSTNESS ACROSS RANDOM SEEDS

To verify that our improvements are not artifacts of random initialization, we conducted inference runs with five different random seeds on all benchmarks. Table 1 reports the mean accuracy and standard deviation for each strategy. Across datasets, we observe that the optimal-question strategy consistently achieves the highest mean performance, while also maintaining stability across seeds. Other instruction settings yield moderate gains, but often with larger variance or even performance drops (e.g., on XSTest and WildGuard under unoptimal instructions). These results confirm that our findings are robust with respect to random seed selection, further strengthening the validity of the proposed framework.

### B.7    ROBUSTNESS ACROSS MODELS AND SIZES

We further evaluated the effectiveness of our method across a diverse set of model families and parameter scales, including Deepseek-R1 (671B), LLaMA-4-Maverick (400B), Gemma-3 (27B), and KIMI-K2 Instruct (1T). As shown in Table 2, prompting with the optimal reasoning pattern consistently improved performance over the baseline of using only the question. These gains hold across both MATH-500 and GPQA-Diamond, ranging from modest improvements in smaller models (e.g., +1.2 on Gemma-3 for MATH-500) to more substantial gains in larger models (e.g., +2.0 on KIMI-K2 Instruct for MATH-500). Together, these results highlight the generality of our approach, demonstrating that reasoning pattern control provides benefits regardless of model architecture or scale.

### B.8    APPLICATION OF THE FRAMEWORK FOR CURATING IMPROVED REASONING DATASETS

We directly addressed this point by using our framework to curate an improved reasoning dataset. Specifically, we used our question-specific reasoning pattern classifier to prompt teacher models to generate responses that reflect the optimal reasoning pattern for each question. We then collected these responses to create a new dataset, CoT-Encyclopedia-10K. To fairly evaluate its effectiveness, we compared our dataset against OpenThinkPatterns-10k, constructed following the recipe and prompts of ThinkPatterns-21k using OpenThoughts questions, since the original ThinkPatterns-21k dataset is not publicly available. Both datasets were used to train Distill-R1-Qwen-1.5B and evaluated on GPQA-Diamond and MATH500. As shown in Table 3, across all benchmarks, models trained on CoT-Encyclopedia-10K consistently outperformed baseline. We attribute this to the fact that

Table 1: Random-seed robust performance across five benchmarks. We report mean accuracy and standard deviation over five different random seeds. The optimal thinking pattern strategy consistently yields the highest performance.

| Strategy | GPQA-Diamond | | MMLU-Redux | | MATH-500 | | XSTest | | WildGuard | |
|---|---|---|---|---|---|---|---|---|---|---|
| | Mean | Std | Mean | Std | Mean | Std | Mean | Std | Mean | Std |
| Not instructed | 72.9 | 1.4 | 90.3 | 1.1 | 78.3 | 1.1 | 91.3 | 1.1 | 89.2 | 1.2 |
| Instructed (unoptimal) | 75.1 | 0.4 | 89.1 | 0.8 | 78.5 | 1.5 | 87.2 | 1.6 | 86.1 | 1.4 |
| Instructed (random) | 77.8 | 0.3 | 90.4 | 1.4 | 79.5 | 0.7 | 91.4 | 1.3 | 90.8 | 0.7 |
| Instructed (optimal-dataset) | 80.6 | 1.3 | 92.8 | 0.9 | 80.6 | 0.5 | 93.8 | 1.4 | 92.9 | 0.7 |
| Instructed (optimal-question) | **82.4** | 1.2 | **93.7** | 0.7 | **81.5** | 0.5 | **96.2** | 1.2 | **95.7** | 0.8 |

Table 2: Performance across diverse model architectures and sizes on MATH-500 and GPQA-Diamond. In all cases, prompting with the optimal reasoning pattern outperforms the baseline of using only the question.

| Model | MATH-500 | | GPQA-Diamond | |
|---|---|---|---|---|
| | Only Question | Optimal Pattern | Only Question | Optimal Pattern |
| Deepseek-R1 (671B) | 92.3 | **94.1** | 85.0 | **86.9** |
| LLaMA-4-Maverick (400B) | 90.2 | **92.3** | 81.7 | **83.2** |
| Gemma-3 (27B) | 86.5 | **87.7** | 72.6 | **74.8** |
| KIMI-K2 Instruct (1T) | 95.7 | **97.7** | 87.2 | **88.8** |

ThinkPatterns-21k use a fixed set of predefined patterns for all questions, making it difficult to match the optimal reasoning process for diverse queries. In contrast, our framework enables dynamic, question-specific reasoning pattern selection, resulting in better adaptability and performance.

### B.9 PERFORMANCE OF REASONING PATTERN PROMPTING ON SMALLER MODELS

While reasoning pattern prompting yields consistent improvements on larger, instruction-tuned models (e.g., 32B+ LLMs, GPT-4o, Claude-Sonnet), its effect on smaller or base models is limited. As shown in Table 4, direct prompting with reasoning patterns does not improve performance for a distilled 1.5B model (Deepseek-R1-Distill-Qwen-1.5B), and can even slightly reduce accuracy compared to the baseline.

To address this limitation, we adopted a data-centric approach inspired by ThinkPatterns-21k (Wen et al., 2025), constructing a curated dataset of 10k optimal reasoning pattern responses (CoT-Encyclopedia-10k) from OpenThought. After supervised fine-tuning (SFT) on this dataset, the small model demonstrated substantial gains, achieving improvements of +3.8 on MATH-500 and +5.5 on GPQA-Diamond. These results suggest that data-centric strategies are essential for enabling smaller models to benefit from reasoning pattern control.

## C REPRODUCIBILITY STATEMENT

All results reported in this study are reproducible using the code provided in the supplementary resources. Reported results are averaged over multiple inference runs to enhance reliability, and we additionally report standard deviations alongside the means to further strengthen reproducibility and trustworthiness. We also document all prompts, hyperparameters, models, and datasets used in the study in the Appendix, ensuring that the results can be reproduced at any time. Finally, the examples used in the qualitative analysis are randomly selected; we plan to include additional examples in the camera-ready version if needed.

## D LLM USAGE STATEMENT

No LLM assistance was used in writing this paper; LLMs were utilized exclusively for constructing our framework.

Table 3: Performance on MATH-500 and GPQA-Diamond with different prompting strategies. Using structured reasoning patterns consistently improves performance compared to no prompting.

| Prompting Strategy | MATH-500 | | GPQA-Diamond | |
|---|---|---|---|---|
| | Accuracy | $\Delta$ vs. No Prompt | Accuracy | $\Delta$ vs. No Prompt |
| No Prompt | 83.9 | – | 33.8 | – |
| OpenThinkPatterns | 85.2 | +1.3 | 36.7 | +2.9 |
| CoT-Encyclopedia | **87.7** | +3.8 | **39.3** | +5.5 |

Table 4: Performance of reasoning pattern prompting and supervised fine-tuning (SFT) on a small model (Deepseek-R1-Distill-Qwen-1.5B). Prompting alone shows little or no improvement, while SFT on the CoT-Encyclopedia-10k dataset yields significant gains.

| Strategy | MATH-500 | | GPQA-Diamond | |
|---|---|---|---|---|
| | Score | $\Delta$ | Score | $\Delta$ |
| Only question | 83.9 | – | 33.8 | – |
| CoT-Encyclopedia prompting | 80.2 | -3.7 | 32.9 | -0.9 |
| CoT-Encyclopedia SFT (10k) | **87.7** | +3.8 | **39.3** | +5.5 |

Table 5: **Summary of classification criteria and associated pattern definitions.**

| Criterion | Pattern A | Pattern B |
|---|---|---|
| Analytical Perspective | **Top-Down**: Begins with high-level principles and decomposes into substeps. | **Bottom-Up**: Builds reasoning from specific evidence or examples toward a general conclusion. |
| Scope of Approach | **Focused**: Restricts analysis to a narrow, targeted set of factors. | **Broad Exploration**: Considers a wide range of factors and possibilities. |
| Reasoning Type | **Inductive**: Infers general rules from observed instances. | **Deductive**: Applies general rules to derive specific conclusions. |
| Idea Development | **Sequential**: Develops ideas in a linear, step-by-step order. | **Parallel**: Explores multiple ideas or lines of reasoning simultaneously. |
| Verification Focus | **Data-Driven**: Emphasizes empirical evidence and data validation. | **Hypothesis-Driven**: Tests predefined hypotheses or assumptions. |
| Clarification Approach | **Iterative Refinement**: Gradually refines and revisits reasoning steps. | **Immediate Conclusion**: Provides final answers with minimal revision steps. |

Table 6: **Comparison of cognitive behavior (Gandhi et al., 2025) frequencies between Distill-R1-32B and s1.1-32B models across three benchmarks.** Statistical analysis (p-values and Cohen's d) shows minimal differences between models, with only one significant difference, indicating limitations of conventional cognitive behavior classifications in distinguishing model reasoning strategies.

| Benchmark | Behavior | Ratio (Behavior Frequency / Total Responses) | | Are they different? | *p*-value | Cohen's *d* |
| | | Distill-R1 | s1.1 | | | |
|---|---|---|---|---|---|---|
| GPQA-Diamond | Verification | 27.3 | 29.8 | No | 0.66 | -0.09 |
| | Backtracking | 33.8 | 33.3 | No | 1.00 | 0.02 |
| | Subgoal Setting | 34.3 | 34.3 | No | 1.00 | 0.00 |
| | Backward Chaining | 73.7 | 72.2 | No | 0.82 | 0.07 |
| MMLU-Redux | Verification | 26.3 | 24.7 | No | 0.16 | 0.04 |
| | Backtracking | 38.6 | 33.1 | Yes | 1e-05 | 0.12 |
| | Subgoal Setting | 27.0 | 27.4 | No | 0.75 | 0.01 |
| | Backward Chaining | 68.9 | 68.7 | No | 0.87 | 4e-3 |
| MATH-500 | Verification | 26.0 | 28.4 | No | 0.43 | -0.05 |
| | Backtracking | 36.2 | 30.4 | No | 0.06 | 0.12 |
| | Subgoal Setting | 27.0 | 28.8 | No | 0.57 | 0.04 |
| | Backward Chaining | 67.6 | 67.6 | No | 1.00 | 0.00 |

Table 7: **Comparison of cognitive behavior (Gandhi et al., 2025) frequencies between Distill-R1-32B and QwQ-32B models across three benchmarks.** Statistical analysis (p-values and Cohen's d) shows minimal differences between models, with only one significant difference, indicating limitations of conventional cognitive behavior classifications in distinguishing model reasoning strategies.

| Benchmark | Behavior | Ratio (Behavior Frequency / Total Responses) | | Are they different? | *p*-value | Cohen's *d* |
| | | Distill-R1 | QwQ | | | |
|---|---|---|---|---|---|---|
| GPQA-Diamond | Verification | 27.3 | 27.8 | No | 0.91 | -0.01 |
| | Backtracking | 33.8 | 32.3 | No | 0.75 | 0.03 |
| | Subgoal Setting | 34.3 | 32.3 | No | 0.67 | 0.04 |
| | Backward Chaining | 73.7 | 74.2 | No | 0.91 | -0.01 |
| MMLU-Redux | Verification | 26.3 | 25.3 | No | 0.38 | 0.02 |
| | Backtracking | 38.6 | 32.2 | Yes | 2.18e-07 | 0.13 |
| | Subgoal Setting | 27.0 | 26.7 | No | 0.79 | 0.01 |
| | Backward Chaining | 68.9 | 70.5 | No | 0.18 | -0.03 |
| MATH-500 | Verification | 26.0 | 27.6 | No | 0.57 | -0.04 |
| | Backtracking | 36.2 | 35.2 | No | 0.74 | 0.02 |
| | Subgoal Setting | 27.0 | 27.4 | No | 0.89 | -0.01 |
| | Backward Chaining | 67.6 | 68.9 | No | 0.68 | -0.03 |

Table 8: **Comparison of cognitive behavior (Gandhi et al., 2025) frequencies between QwQ-32B and s1.1-32B models across three benchmarks.** Statistical analysis (p-values and Cohen's d) shows minimal differences between models, with only one significant difference, indicating limitations of conventional cognitive behavior classifications in distinguishing model reasoning strategies.

| Benchmark | Behavior | Ratio (Behavior Frequency / Total Responses) | | Are they different? | $p$-value | Cohen's $d$ |
|---|---|---|---|---|---|---|
| | | QwQ | s1.1 | | | |
| GPQA-Diamond | Verification | 27.8 | 29.8 | No | 0.66 | -0.04 |
| | Backtracking | 32.3 | 33.3 | No | 0.83 | -0.02 |
| | Subgoal Setting | 32.3 | 34.3 | No | 0.67 | -0.04 |
| | Backward Chaining | 74.2 | 72.2 | No | 0.65 | 0.05 |
| MMLU-Redux | Verification | 25.3 | 24.7 | No | 0.59 | 0.01 |
| | Backtracking | 32.2 | 33.1 | No | 0.46 | -0.02 |
| | Subgoal Setting | 26.7 | 27.4 | No | 0.54 | -0.02 |
| | Backward Chaining | 70.5 | 68.7 | No | 0.13 | 0.04 |
| MATH-500 | Verification | 27.6 | 28.4 | No | 0.78 | -0.02 |
| | Backtracking | 35.2 | 30.4 | No | 0.11 | 0.10 |
| | Subgoal Setting | 27.4 | 28.8 | No | 0.62 | -0.03 |
| | Backward Chaining | 68.9 | 67.6 | No | 0.68 | 0.03 |

Table 9: **Analysis of reasoning strategies between Distill-R1-32B and s1.1-32B models using CoT ENCYCLOPEDIA's six criteria across problem-solving benchmarks.** Unlike traditional cognitive behavior metrics, this approach reveals numerous statistically significant differences (marked as 'Yes'), with substantial effect sizes (Cohen's d up to 0.44), demonstrating CoT Encyclopedia's enhanced ability to distinguish between models' reasoning strategies and preferences.

| Benchmark | Reasoning Behavior | | | <- Pattern A | Pattern B -> | Are they different? | $p$-value | Cohen's $d$ |
|---|---|---|---|---|---|---|---|---|
| | Criteria | Pattern A | Pattern B | Distill-R1 | s1.1 | | | |
| GPQA-Diamond | Analytical Perspective | Top-Down | Bottom-Up | 75.8 | 79.3 | No | 4.7e-2 | -0.10 |
| | Scope of Approach | Focused | Broad | 15.2 | 7.1 | Yes | 2e-2 | 0.26 |
| | Reasoning Type | Inductive | Deductive | 23.7 | 15.1 | Yes | 4e-2 | 0.22 |
| | Idea Development | Sequential | Parallel | 30.3 | 40.9 | Yes | 4e-2 | -0.22 |
| | Verification Focus | Data-Driven | Hypothesis-Driven | 60.1 | 68.2 | Yes | 1.2e-2 | -0.18 |
| | Clarification Approach | Iterative | Immediate | 7.6 | 1.0 | Yes | 2e-03 | 0.36 |
| MMLU-Redux | Analytical Perspective | Top-Down | Bottom-Up | 32.7 | 33.1 | No | 0.76 | 0.01 |
| | Scope of Approach | Focused | Broad | 43.6 | 28.4 | Yes | 1e-34 | 0.32 |
| | Reasoning Type | Inductive | Deductive | 8.9 | 5.7 | Yes | 2e-06 | 0.12 |
| | Idea Development | Sequential | Parallel | 27.4 | 38.5 | Yes | 4e-20 | 0.24 |
| | Verification Focus | Data-Driven | Hypothesis-Driven | 87.5 | 84.6 | Yes | 1e-03 | 0.08 |
| | Clarification Approach | Iterative | Immediate | 23.2 | 10.6 | Yes | 3e-38 | 0.34 |
| MATH-500 | Analytical Perspective | Top-Down | Bottom-Up | 36.8 | 43.8 | Yes | 0.03 | 0.14 |
| | Scope of Approach | Focused | Broad | 55.6 | 37.4 | Yes | 1e-08 | 0.37 |
| | Reasoning Type | Inductive | Deductive | 10.4 | 11.0 | No | 8.4e-2 | 0.02 |
| | Idea Development | Sequential | Parallel | 17.0 | 21.8 | No | 6e-2 | 0.12 |
| | Verification Focus | Data-Driven | Hypothesis-Driven | 69.6 | 64.2 | No | 8e-2 | 0.12 |
| | Clarification Approach | Iterative | Immediate | 14.4 | 2.8 | Yes | 1e-10 | 0.44 |

Table 10: **Analysis of reasoning strategies between Distill-R1-32B and QwQ-32B models using** CoT ENCYCLOPEDIA's **six criteria across problem-solving benchmarks.** Unlike traditional cognitive behavior metrics, this approach reveals numerous statistically significant differences (marked as 'Yes'), with substantial effect sizes (Cohen's d up to 0.44), demonstrating CoT Encyclopedia's enhanced ability to distinguish between models' reasoning strategies and preferences.

| Benchmark | Reasoning Behavior | | | <- Pattern A | Pattern B -> | Are they different? | $p$-value | Cohen's $d$ |
|---|---|---|---|---|---|---|---|---|
| | Criteria | Pattern A | Pattern B | Distill-R1 | QwQ | | | |
| GPQA-Diamond | Analytical Perspective | Top-Down | Bottom-Up | 75.8 | 80.2 | No | 0.27 | -0.11 |
| | Scope of Approach | Focused | Broad | 15.2 | 5.2 | Yes | 8.5e-4 | 0.34 |
| | Reasoning Type | Inductive | Deductive | 23.7 | 12.1 | Yes | 2.6e-3 | 0.31 |
| | Idea Development | Sequential | Parallel | 30.3 | 43.2 | Yes | 6.8e-3 | -0.27 |
| | Verification Focus | Data-Driven | Hypothesis-Driven | 60.1 | 70.9 | Yes | 2.7e-2 | -0.23 |
| | Clarification Approach | Iterative | Immediate | 7.6 | 2.1 | Yes | 9.7e-3 | 0.26 |
| MMLU-Redux | Analytical Perspective | Top-Down | Bottom-Up | 32.7 | 33.1 | No | 0.74 | -0.01 |
| | Scope of Approach | Focused | Broad | 43.6 | 24.6 | Yes | 2.4e-54 | 0.41 |
| | Reasoning Type | Inductive | Deductive | 8.9 | 4.3 | Yes | 7.2e-13 | 0.19 |
| | Idea Development | Sequential | Parallel | 27.4 | 39.2 | Yes | 3.1e-22 | -0.25 |
| | Verification Focus | Data-Driven | Hypothesis-Driven | 87.5 | 82.5 | Yes | 5.9e-8 | 0.14 |
| | Clarification Approach | Iterative | Immediate | 23.2 | 13.4 | Yes | 9.6e-23 | 0.26 |
| MATH-500 | Analytical Perspective | Top-Down | Bottom-Up | 36.8 | 42.5 | No | 0.07 | -0.12 |
| | Scope of Approach | Focused | Broad | 55.6 | 33.2 | Yes | 1.0e-12 | 0.46 |
| | Reasoning Type | Inductive | Deductive | 10.4 | 13.2 | No | 0.17 | -0.09 |
| | Idea Development | Sequential | Parallel | 17.0 | 22.4 | Yes | 3.2e-2 | -0.14 |
| | Verification Focus | Data-Driven | Hypothesis-Driven | 69.6 | 61.4 | Yes | 6.4e-3 | 0.17 |
| | Clarification Approach | Iterative | Immediate | 14.4 | 4.3 | Yes | 6.0e-8 | 0.35 |

Table 11: **Analysis of reasoning strategies between QwQ-32B and s1.1-32B models using** CoT ENCYCLOPEDIA's **six criteria across problem-solving benchmarks.** Unlike traditional cognitive behavior metrics, this approach reveals numerous statistically significant differences (marked as 'Yes'), with substantial effect sizes (Cohen's d up to 0.35), demonstrating CoT Encyclopedia's enhanced ability to distinguish between models' reasoning strategies and preferences.

| Benchmark | Reasoning Behavior | | | <- Pattern A | Pattern B -> | Are they different? | $p$-value | Cohen's $d$ |
|---|---|---|---|---|---|---|---|---|
| | Criteria | Pattern A | Pattern B | QwQ | s1.1 | | | |
| GPQA-Diamond | Analytical Perspective | Top-Down | Bottom-Up | 80.2 | 79.3 | No | 0.80 | 0.02 |
| | Scope of Approach | Focused | Broad | 5.2 | 7.1 | No | 0.40 | -0.08 |
| | Reasoning Type | Inductive | Deductive | 12.1 | 15.1 | No | 0.38 | -0.09 |
| | Idea Development | Sequential | Parallel | 43.2 | 40.9 | No | 0.61 | 0.05 |
| | Verification Focus | Data-Driven | Hypothesis-Driven | 70.9 | 68.2 | No | 0.59 | 0.06 |
| | Clarification Approach | Iterative | Immediate | 2.1 | 1.0 | No | 0.41 | 0.09 |
| MMLU-Redux | Analytical Perspective | Top-Down | Bottom-Up | 33.1 | 33.1 | No | 1.00 | 0.00 |
| | Scope of Approach | Focused | Broad | 24.6 | 28.4 | Yes | 8.5e-04 | -0.09 |
| | Reasoning Type | Inductive | Deductive | 4.3 | 5.7 | Yes | 1.3e-02 | -0.06 |
| | Idea Development | Sequential | Parallel | 39.2 | 38.5 | No | 0.58 | 0.01 |
| | Verification Focus | Data-Driven | Hypothesis-Driven | 82.5 | 84.6 | Yes | 2.8e-02 | -0.06 |
| | Clarification Approach | Iterative | Immediate | 13.4 | 10.6 | Yes | 8.5e-04 | 0.09 |
| MATH-500 | Analytical Perspective | Top-Down | Bottom-Up | 42.5 | 43.8 | No | 0.65 | -0.03 |
| | Scope of Approach | Focused | Broad | 33.2 | 37.4 | No | 0.16 | -0.09 |
| | Reasoning Type | Inductive | Deductive | 13.2 | 11.0 | No | 0.29 | 0.07 |
| | Idea Development | Sequential | Parallel | 22.4 | 21.8 | No | 0.82 | 0.01 |
| | Verification Focus | Data-Driven | Hypothesis-Driven | 61.4 | 64.2 | No | 0.36 | -0.06 |
| | Clarification Approach | Iterative | Immediate | 14.4 | 4.3 | Yes | 6.0e-08 | 0.35 |

Table 12: **Analysis of reasoning strategies between Distill-R1-32B and s1.1-32B models using COT ENCYCLOPEDIA's six criteria across safety benchmarks.** Unlike traditional cognitive behavior metrics, this approach reveals numerous statistically significant differences (marked as 'Yes'), with substantial effect sizes (Cohen's d up to 0.50), demonstrating CoT Encyclopedia's enhanced ability to distinguish between models' reasoning strategies and preferences.

| Benchmark | Reasoning Behavior | | | <- Pattern A | Pattern B -> | Are they different? | $p$-value | Cohen's $d$ |
|---|---|---|---|---|---|---|---|---|
| | Criteria | Pattern A | Pattern B | Distill-R1 | s1.1 | | | |
| XSTest | Guidance Strategy | Non-Directive | Directive | 39.0 | 40.3 | No | 0.73 | -0.03 |
| | Ethical Consideration | Technical Response | Moral Evaluation | 36.1 | 22.3 | Yes | 5.4e-06 | 0.31 |
| | Safety Precedence | Risk-Engaging | Preventive | 59.3 | 62.3 | No | 0.37 | -0.06 |
| | Idea Development | Sequential | Parallel | 18.3 | 40.9 | Yes | 9.2e-14 | -0.50 |
| | Content Handling | Open Discussion | Censorship | 12.1 | 68.2 | Yes | 2.4e-66 | -1.23 |
| | Risk Assessment | High-Risk Awareness | Low-Risk Assumption | 22.7 | 19.1 | No | 0.19 | 0.09 |
| | Content Intent | Malicious | Benevolent | 100.0 | 99.5 | No | 0.16 | 0.14 |
| WildGuard | Guidance Strategy | Non-Directive | Directive | 40.3 | 43.2 | No | 0.084 | -0.06 |
| | Ethical Consideration | Technical Response | Moral Evaluation | 22.3 | 26.1 | Yes | 0.0098 | -0.09 |
| | Safety Precedence | Risk-Engaging | Preventive | 56.0 | 67.5 | Yes | 4.0e-12 | -0.24 |
| | Idea Development | Sequential | Parallel | 22.0 | 42.3 | Yes | 2.9e-37 | -0.44 |
| | Content Handling | Open Discussion | Censorship | 0.04 | 3.4 | Yes | 4.2e-14 | -0.33 |
| | Risk Assessment | High-Risk Awareness | Low-Risk Assumption | 29.5 | 46.1 | Yes | 1.0e-23 | -0.34 |
| | Content Intent | Malicious | Benevolent | 100.0 | 99.2 | Yes | 1.8e-04 | 0.18 |

Table 13: **Analysis of reasoning strategies between Distill-R1-32B and QwQ-32B models using COT ENCYCLOPEDIA's six criteria across safety benchmarks.** Unlike traditional cognitive behavior metrics, this approach reveals numerous statistically significant differences (marked as 'Yes'), with substantial effect sizes (Cohen's d up to 1.44), demonstrating CoT Encyclopedia's enhanced ability to distinguish between models' reasoning strategies and preferences.

| Benchmark | Reasoning Behavior | | | <- Pattern A | Pattern B -> | Are they different? | $p$-value | Cohen's $d$ |
|---|---|---|---|---|---|---|---|---|
| | Criteria | Pattern A | Pattern B | Distill-R1 | QwQ | | | |
| XSTest | Guidance Strategy | Non-Directive | Directive | 39.0 | 43.3 | No | 0.177 | -0.09 |
| | Ethical Consideration | Technical | Moral Evaluation | 36.1 | 25.2 | Yes | 2.1e-4 | 0.24 |
| | Safety Precedence | Risk-Engaging | Preventive | 59.3 | 67.2 | Yes | 8.7e-3 | -0.16 |
| | Idea Development | Sequential | Parallel | 18.3 | 43.2 | Yes | 2.0e-17 | -0.56 |
| | Content Handling | Open Discussion | Censorship | 12.1 | 69.5 | Yes | 1.2e-76 | -1.44 |
| | Risk Assessment | High-Risk Awareness | Low-Risk Assumption | 22.7 | 29.1 | Yes | 2.11e-2 | -0.15 |
| | Content Intent | Malicious | Benevolent | 100.0 | 100.0 | No | 1.00 | 0.00 |
| WildGuard | Guidance Strategy | Non-Directive | Directive | 40.3 | 46.7 | Yes | 4.13e-2 | -0.13 |
| | Ethical Consideration | Technical | Moral Evaluation | 22.3 | 29.5 | Yes | 9.45e-3 | -0.17 |
| | Safety Precedence | Risk-Engaging | Preventive | 56.0 | 62.5 | Yes | 3.95e-2 | -0.13 |
| | Idea Development | Sequential | Parallel | 22.0 | 41.2 | Yes | 6.59e-11 | -0.42 |
| | Content Handling | Open Discussion | Censorship | 0.04 | 8.4 | Yes | 3.56e-11 | -0.43 |
| | Risk Assessment | High-Risk Awareness | Low-Risk Assumption | 29.5 | 48.3 | Yes | 1.10e-9 | -0.39 |
| | Content Intent | Malicious | Benevolent | 100.0 | 100.0 | No | 1.00 | 0.00 |

Table 14: **Analysis of reasoning strategies between QwQ-32B and s1.1-32B models using COT ENCYCLOPEDIA's six criteria across safety benchmarks.** Unlike traditional cognitive behavior metrics, this approach reveals statistically significant differences (marked as 'Yes'), with moderate effect sizes (Cohen's d up to 0.24), demonstrating CoT Encyclopedia's enhanced ability to distinguish between models' reasoning strategies and preferences.

| Benchmark | Reasoning Behavior | | | <- Pattern A | Pattern B -> | Are they different? | $p$-value | Cohen's $d$ |
|---|---|---|---|---|---|---|---|---|
| | Criteria | Pattern A | Pattern B | QwQ | s1.1 | | | |
| XSTest | Guidance Strategy | Non-Directive | Directive | 43.3 | 40.3 | No | 0.37 | 0.06 |
| | Ethical Consideration | Technical Response | Moral Evaluation | 25.2 | 22.3 | No | 0.30 | 0.07 |
| | Safety Precedence | Risk-Engaging | Preventive | 67.2 | 62.3 | No | 0.11 | 0.10 |
| | Idea Development | Sequential | Parallel | 43.2 | 40.9 | No | 0.44 | 0.05 |
| | Content Handling | Open Discussion | Censorship | 69.5 | 68.2 | No | 0.63 | 0.03 |
| | Risk Assessment | High-Risk Awareness | Low-Risk Assumption | 29.1 | 19.1 | Yes | 2e-4 | 0.24 |
| | Content Intent | Malicious | Benevolent | 100.0 | 100.0 | No | 1.00 | 0.00 |
| WildGuard | Guidance Strategy | Non-Directive | Directive | 46.7 | 43.2 | No | 0.25 | 0.07 |
| | Ethical Consideration | Technical Response | Moral Evaluation | 29.5 | 26.1 | No | 0.20 | 0.08 |
| | Safety Precedence | Risk-Engaging | Preventive | 62.5 | 67.5 | No | 0.08 | -0.10 |
| | Idea Development | Sequential | Parallel | 41.2 | 42.3 | No | 0.70 | -0.02 |
| | Content Handling | Open Discussion | Censorship | 8.4 | 3.4 | Yes | 7e-4 | 0.21 |
| | Risk Assessment | High-Risk Awareness | Low-Risk Assumption | 48.3 | 46.1 | No | 0.45 | 0.04 |
| | Content Intent | Malicious | Benevolent | 100.0 | 100.0 | No | 1.00 | 0.00 |

Table 15: **Comparison of high-level reasoning style preferences across models trained on different data formats.** This table presents a detailed analysis of how question format influences reasoning strategies across three benchmarks: GPQA-Diamond, MMLU-Redux, and MATH-500. For each benchmark, six reasoning criteria are evaluated with contrasting pattern pairs (Pattern A vs. Pattern B). Statistical significance testing (p-values) and effect size measurements (Cohen's d) reveal that multiple-choice and free-form trained models exhibit significantly different reasoning strategies on most criteria. Particularly pronounced differences appear in the MATH-500 benchmark, where effect sizes reach up to 1.58, demonstrating that training data format substantially shapes models' problem-solving approaches independent of content domain.

| Benchmark | Reasoning Behavior | | | <- Pattern A | Pattern B -> | Are they different? | $p$-value | Cohen's $d$ |
|---|---|---|---|---|---|---|---|---|
| | Criteria | Pattern A | Pattern B | Multiple-choice | Free-form | | | |
| GPQA-Diamond | Analytical Perspective | Top-Down | Bottom-Up | 63.9 | 46.3 | Yes | 1e-36 | 0.36 |
| | Scope of Approach | Focused | Broad | 74.8 | 76.7 | No | 0.12 | -0.04 |
| | Reasoning Type | Inductive | Deductive | 54.8 | 38.8 | Yes | 8e-30 | 0.32 |
| | Idea Development | Sequential | Parallel | 27.0 | 51.7 | Yes | 3e-74 | 0.51 |
| | Verification Focus | Data-Driven | Hypothesis-Driven | 39.6 | 38.4 | No | 0.41 | 0.02 |
| | Clarification Approach | Iterative | Immediate | 21.0 | 12.7 | Yes | 3e-14 | 0.22 |
| MMLU-Redux | Analytical Perspective | Top-Down | Bottom-Up | 39.5 | 59.2 | Yes | 2e-52 | -0.40 |
| | Scope of Approach | Focused | Broad | 48.6 | 48.3 | No | 0.80 | 0.01 |
| | Reasoning Type | Inductive | Deductive | 27.5 | 76.8 | Yes | 0.00 | -1.03 |
| | Idea Development | Sequential | Parallel | 38.3 | 50.5 | Yes | 1e-21 | 0.25 |
| | Verification Focus | Data-Driven | Hypothesis-Driven | 66.9 | 34.5 | Yes | 5e-139 | 0.66 |
| | Clarification Approach | Iterative | Immediate | 48.8 | 60.4 | Yes | 1e-19 | -0.23 |
| MATH-500 | Analytical Perspective | Top-Down | Bottom-Up | 84.1 | 13.1 | Yes | 0.00 | 1.58 |
| | Scope of Approach | Focused | Broad | 85.9 | 32.9 | Yes | 0.00 | 1.15 |
| | Reasoning Type | Inductive | Deductive | 81.9 | 14.1 | Yes | 0.00 | 1.49 |
| | Idea Development | Sequential | Parallel | 0.02 | 34.8 | Yes | 0.00 | -0.99 |
| | Verification Focus | Data-Driven | Hypothesis-Driven | 13.7 | 52.2 | Yes | 1e-292 | -0.86 |
| | Clarification Approach | Iterative | Immediate | 11.4 | 38.5 | Yes | 1e-171 | -0.65 |

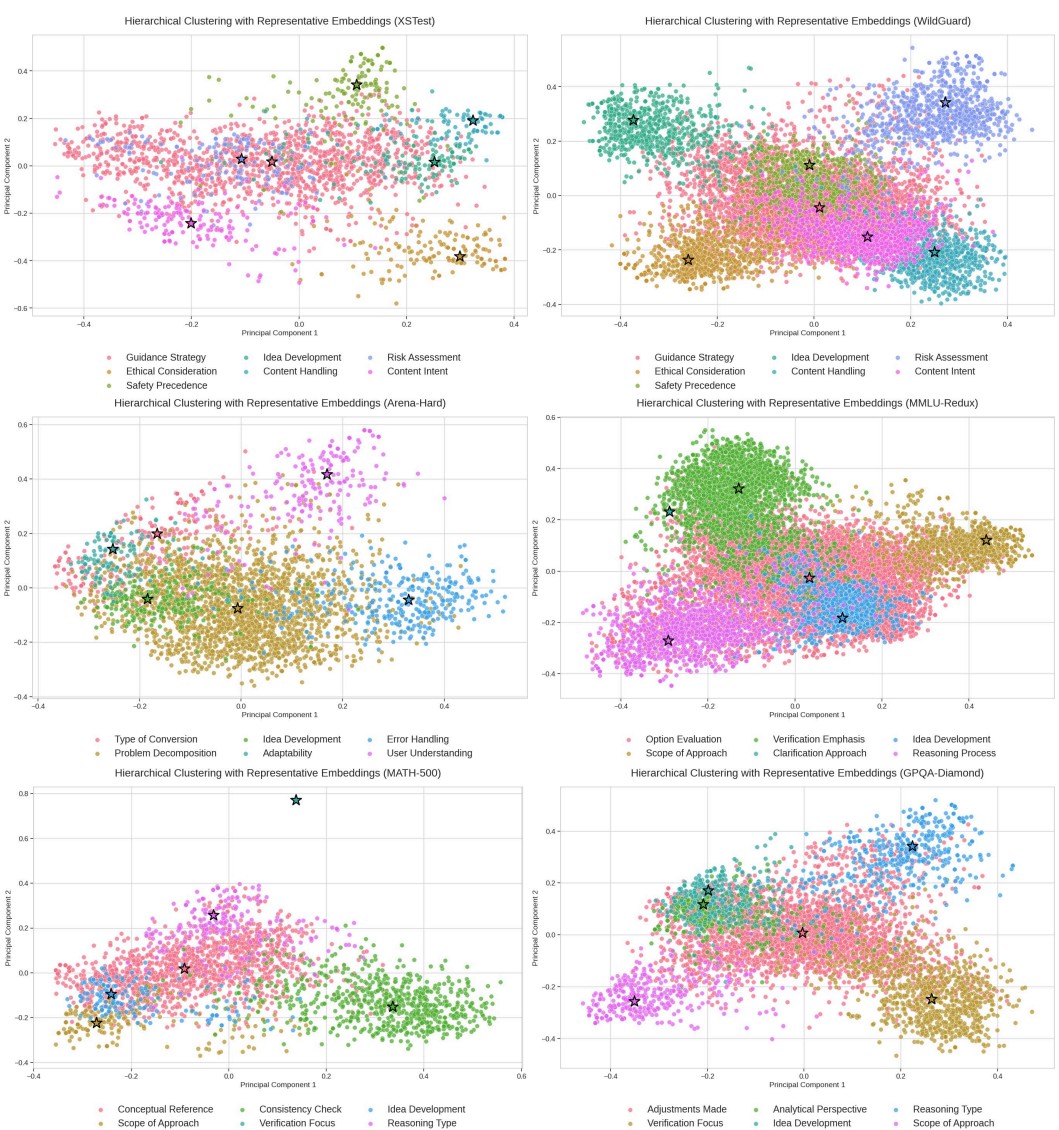

Figure 14: **Hierarchical Clustering Across Benchmarks.** Six different benchmarks: XSTest (top left), WildGuard (top right), Arena-Hard (mid left), and MMLU-Redux (mid right), MATH-500 (bottom left), GPQA-Diamond (bottom right). Stars indicate the representative embeddings (medoids) for each cluster. This visualization demonstrates how different benchmarks employ varied criteria for pattern analysis.

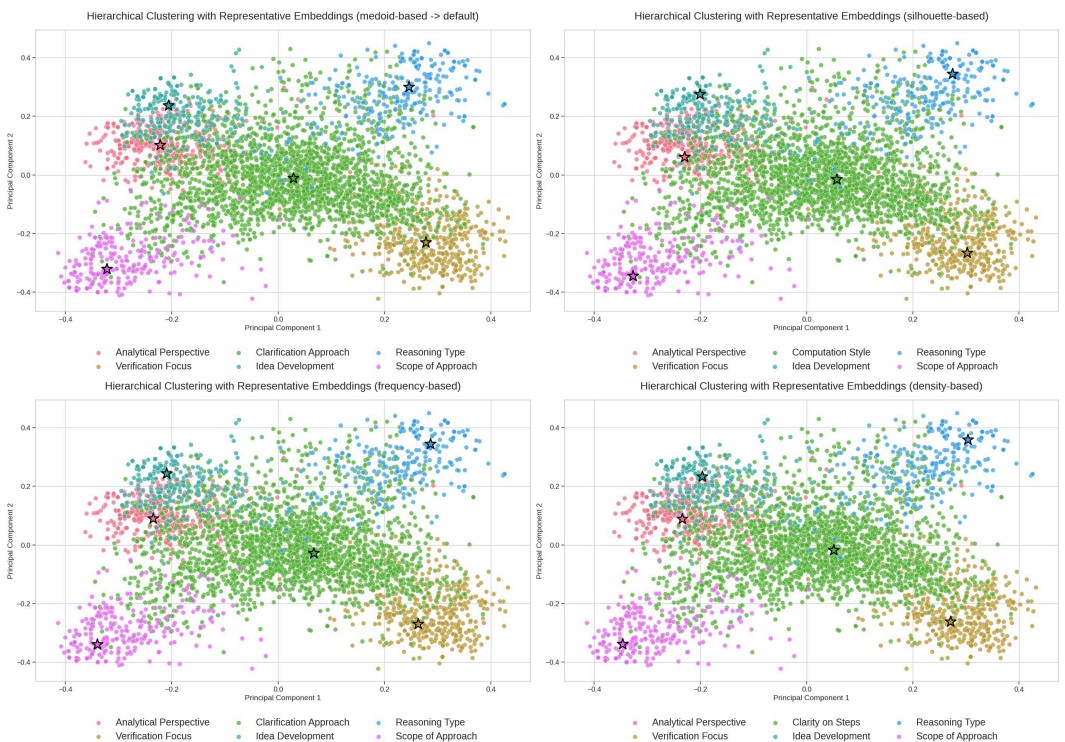

Figure 15: **Hierarchical Clustering of reasoning strategies with PCA Projection on problem-solving Benchmarks.** Comparing four different representative embedding selection methods: medoid-based (default, top-left), silhouette-based (top-right), frequency-based (bottom-left), and density-based (bottom-right). Each stars indicate representative embeddings for each cluster. Despite using different selection criteria, the overall clustering structure remains consistent across methods, with only minor variations in representative embedding positions.

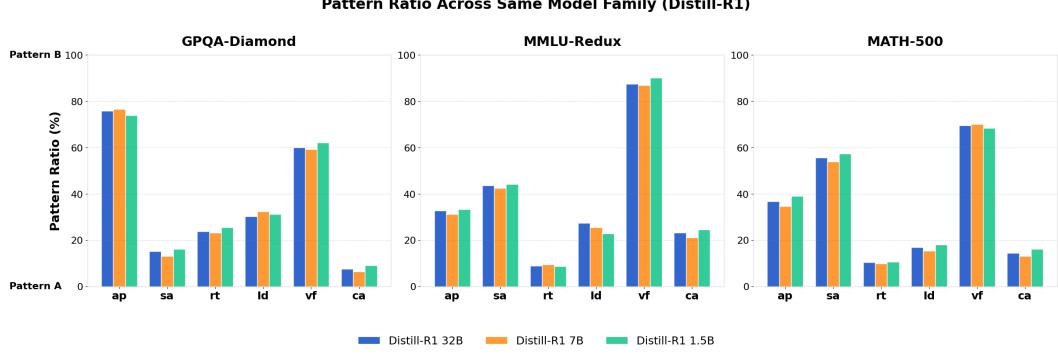

Figure 16: **Pattern ratio distributions across three different sizes of the Distill-R1 model family (32B, 7B, and 1.5B).** The x-axis shows six reasoning strategy criteria: analytical perspective (ap), clarification approach (sa), reasoning type (rt), idea development (id), verification focus (vf), and scope of approach (ca). Despite the significant size differences, all three models exhibit remarkably similar pattern distributions across all benchmarks, supporting the conclusion that models from the same family maintain consistent reasoning strategies regardless of scale.

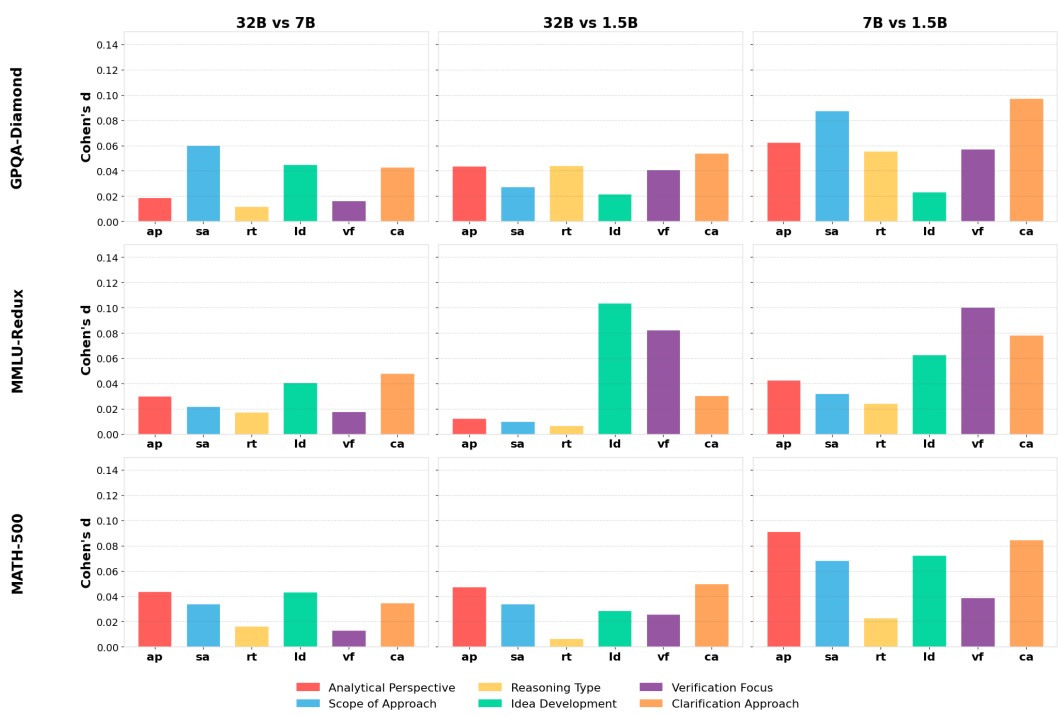

Figure 17: **Pairwise Cohen's d effect size measurements comparing reasoning strategy distributions across different sizes of the Distill-R1 model family.** The x-axis displays the six reasoning strategy criteria: analytical perspective (ap), scope of approach (sa), reasoning type (rt), idea development (id), verification focus (vf), and clarification approach (ca). All effect sizes remain below 0.1 across most comparisons, indicating very small distributional differences between models of different sizes within the same family.

Table 16: **Comparison of high-level reasoning style preferences across models trained on different domains.** This table presents a statistical analysis of how content domain influences reasoning strategies across three benchmarks: GPQA-Diamond, MMLU-Redux, and MATH-500. Six reasoning criteria are evaluated for each benchmark, comparing math-domain versus knowledge-domain training. Statistical testing reveals minimal differences between domains, with most p-values above the significance threshold (0.05) and small effect sizes (Cohen's d mostly below 0.15). The limited number of significant differences (only 4 out of 18 comparisons) and small effect sizes demonstrate that content domain has substantially less impact on reasoning strategy formation than question format, supporting the paper's finding that format characteristics shape reasoning strategies more fundamentally than subject matter.

| Benchmark | Reasoning Behavior | | | <- Pattern A | Pattern B -> | Are they different? | $p$-value | Cohen's $d$ |
| | Criteria | Pattern A | Pattern B | Math-domain | Knowledge-domain | | | |
|---|---|---|---|---|---|---|---|---|
| GPQA-Diamond | Analytical Perspective | Top-Down | Bottom-Up | 60.0 | 61.4 | No | 0.25 | 0.03 |
| | Scope of Approach | Focused | Broad | 61.0 | 62.3 | No | 0.29 | -0.027 |
| | Reasoning Type | Inductive | Deductive | 60.3 | 60.9 | No | 0.60 | -0.014 |
| | Idea Development | Sequential | Parallel | 38.5 | 37.7 | No | 0.52 | 0.017 |
| | Verification Focus | Data-Driven | Hypothesis-Driven | 38.6 | 38.3 | No | 0.80 | 0.007 |
| | Clarification Approach | Iterative | Immediate | 40.4 | 36.4 | Yes | 1e-3 | 0.082 |
| MMLU-Redux | Analytical Perspective | Top-Down | Bottom-Up | 41.2 | 42.3 | No | 0.77 | -0.02 |
| | Scope of Approach | Focused | Broad | 38.6 | 45.5 | Yes | 0.03 | -0.14 |
| | Reasoning Type | Inductive | Deductive | 42.0 | 44.4 | No | 0.49 | -0.05 |
| | Idea Development | Sequential | Parallel | 57.4 | 57.8 | No | 0.95 | -0.01 |
| | Verification Focus | Data-Driven | Hypothesis-Driven | 59.2 | 54.5 | No | 0.14 | 0.09 |
| | Clarification Approach | Iterative | Immediate | 58.7 | 53.6 | No | 0.13 | 0.10 |
| MATH-500 | Analytical Perspective | Top-Down | Bottom-Up | 41.2 | 43.6 | Yes | 0.03 | 0.14 |
| | Scope of Approach | Focused | Broad | 38.6 | 42.0 | Yes | 1e-08 | 0.37 |
| | Reasoning Type | Inductive | Deductive | 42.0 | 40.0 | No | 8.4e-2 | 0.02 |
| | Idea Development | Sequential | Parallel | 57.4 | 57.2 | No | 6e-2 | 0.12 |
| | Verification Focus | Data-Driven | Hypothesis-Driven | 59.2 | 56.4 | No | 8e-2 | 0.12 |
| | Clarification Approach | Iterative | Immediate | 58.7 | 59.8 | Yes | 1e-10 | 0.44 |

Table 17: **Comparison of reasoning behavior patterns between Qwen and Qwen-Math models across three benchmarks.** The table presents statistical analyses of six reasoning criteria: Decision Strategy, Reasoning Type, Analytical Perspective, Verification Focus, Scope of Approach, and Idea Development. For each criterion, the table shows the distribution percentages of Pattern A and Pattern B for both models, along with statistical significance measures (p-value and Cohen's d).

| Benchmark | Reasoning Behavior | | | <- Pattern A | Pattern B -> | Are they different? | p-value | Cohen's d |
| | Criteria | Pattern A | Pattern B | Qwen | Qwen-Math | | | |
|---|---|---|---|---|---|---|---|---|
| **GPQA-Diamond** | Decision Strategy | Optimal | Satisficing | 53.1 | 57.3 | No | 0.419 | -0.085 |
| | Reasoning Type | Inductive | Deductive | 37.6 | 36.2 | No | 0.835 | 0.029 |
| | Analytical Perspective | Top-Down | Bottom-Up | 21.3 | 45.2 | Yes | 5.17e-7 | -0.524 |
| | Verification Focus | Principle-Driven | Numerical Validation | 21.7 | 63.3 | Yes | 7.58e-17 | -0.928 |
| | Scope of Approach | Focused | Broad Exploration | 65.4 | 45.6 | Yes | 8.09e-5 | 0.407 |
| | Idea Development | Linear Sequential | Parallel | 10.3 | 10.4 | No | 0.869 | -0.003 |
| **MMLU-Redux** | Decision Strategy | Optimal | Satisficing | 62.2 | 62.3 | No | 0.936 | -0.002 |
| | Reasoning Type | Inductive | Deductive | 78.3 | 67.2 | Yes | 4.67e-22 | 0.251 |
| | Analytical Perspective | Top-Down | Bottom-Up | 53.4 | 87.5 | Yes | 3.20e-184 | -0.806 |
| | Verification Focus | Principle-Driven | Numerical Validation | 36.2 | 36.9 | No | 0.574 | -0.015 |
| | Scope of Approach | Focused | Broad Exploration | 41.9 | 55.2 | Yes | 6.59e-25 | -0.268 |
| | Idea Development | Linear Sequential | Parallel | 10.1 | 31.0 | Yes | 2.86e-89 | -0.535 |
| **MATH-500** | Decision Strategy | Optimal | Satisficing | 47.2 | 23.3 | Yes | 1.93e-15 | 0.517 |
| | Reasoning Type | Inductive | Deductive | 15.2 | 17.1 | No | 0.391 | -0.052 |
| | Analytical Perspective | Top-Down | Bottom-Up | 30.1 | 35.1 | No | 0.0794 | -0.107 |
| | Verification Focus | Principle-Driven | Numerical Validation | 34.2 | 67.9 | Yes | 1.12e-26 | -0.716 |
| | Scope of Approach | Focused | Broad Exploration | 54.9 | 69.2 | Yes | 2.72e-6 | -0.298 |
| | Idea Development | Linear Sequential | Parallel | 22.1 | 24.0 | No | 0.452 | -0.045 |

Table 18: **Comparison of reasoning strategies between QwQ and s1.1 models on safety benchmarks.** This table presents statistical analyses of seven reasoning criteria across two safety benchmarks (XSTest and WildGuard). For each criterion, the table shows the distribution percentages of Pattern A and Pattern B for both models, along with statistical significance indicators (p-value and Cohen's d).

| Benchmark | Reasoning Behavior | | | <- Pattern A | Pattern B -> | Are they different? | p-value | Cohen's d |
| | Criteria | Pattern A | Pattern B | QwQ | s1.1 | | | |
|---|---|---|---|---|---|---|---|---|
| **XSTest** | Guidance Strategy | Non-Directive | Directive | 23.3 | 22.7 | No | 1.00 | 0.01 |
| | Ethical Consideration | Technical Response | Moral Evaluation | 47.2 | 42.3 | No | 0.48 | 0.10 |
| | Safety Precedence | Risk-Engaging | Preventive | 62.2 | 67.3 | No | 0.46 | -0.11 |
| | Idea Development | Sequential | Parallel | 41.2 | 70.9 | Yes | 1.92e-05 | -0.63 |
| | Content Handling | Open Discussion | Censorship | 69.5 | 78.2 | No | 0.20 | -0.20 |
| | Risk Assessment | High-Risk Awareness | Low-Risk Assumption | 20.1 | 3.1 | Yes | 1.65e-04 | 0.55 |
| | Content Intent | Malicious | Benevolent | 98.7 | 96.3 | No | 0.17 | 0.15 |
| **WildGuard** | Guidance Strategy | Non-Directive | Directive | 23.7 | 23.2 | No | 0.87 | 0.01 |
| | Ethical Consideration | Technical Response | Moral Evaluation | 26.5 | 29.1 | No | 0.63 | -0.06 |
| | Safety Precedence | Risk-Engaging | Preventive | 51.5 | 49.5 | No | 0.78 | 0.04 |
| | Idea Development | Sequential | Parallel | 27.2 | 46.3 | Yes | 5.26e-03 | -0.40 |
| | Content Handling | Open Discussion | Censorship | 10.4 | 21.4 | Yes | 3.16e-02 | -0.30 |
| | Risk Assessment | High-Risk Awareness | Low-Risk Assumption | 35.3 | 32.1 | No | 0.65 | 0.07 |
| | Content Intent | Malicious | Benevolent | 100.0 | 100.0 | No | 1.00 | 0.00 |

## Multiple-choice format (MC) data

Q. Diamond is extremely hard because of the strong bonding between _______ in all directions?

Options:
A. carbon atoms
B. liquid atoms
C. hydrogen atoms
D. consistent atoms

## Free-form (FF) data

The equation of the circle symmetric to the circle $(x-2)^2+(y-2)^2=7$ with respect to the line $x+y=2$ is

## Multiple-choice format (MC) response

Okay, so I'm trying to figure out the answer to this question about diamond being extremely hard.

The question is asking why diamond is so hard because of the strong bonding between the atoms in all directions, and there are four options: A. carbon atoms, B. liquid atoms, C. hydrogen atoms, D. consistent atoms.

...

Option A is carbon atoms. Diamond is made up of carbon atoms, so that makes sense. But why would the atoms bond strongly? I remember that carbon atoms have strong bonds with each other, which contributes to the structure of diamond. So, maybe A is correct.

...

Option D is consistent atoms. Consistency could mean that the atoms are uniform and not varying, but I'm not sure how that relates to the hardness. It doesn't seem as relevant as the bonding between atoms.

So, putting it together, the strong bonding between carbon atoms in diamond explains its hardness. Therefore, the answer should be A.

The correct answer is **A. carbon atoms** because diamond, composed of carbon, exhibits high hardness due to its strong and consistent bonding between carbon atoms in all directions.

## Free-form (FF) response

Okay, so I need to find the equation of a circle that's symmetric to the given circle $(x-2)^2 + (y-2)^2 = 7$ with respect to the line $x + y = 2$. Hmm, symmetric with respect to a line means that the line is the ...

Wait, that doesn't seem right. If I plug $(2, 2)$ into the reflection formulas with $a=1$, $b=1$, $c=-2$:

Wait, that contradicts my earlier result. Let me check the reflection formula again.

Wait, perhaps I used the wrong formula. Let me derive the reflection point.

...

Then,

$$x' = 2 - \frac{2*1*2}{1 + 1} = 2 - \frac{4}{2} = 2 - 2 = 0$$
$$y' = 2 - \frac{2*1*2}{1 + 1} = 2 - 2 = 0$$

Thus, the equation of the reflected circle is:
$$(x - 0)^2 + (y - 0)^2 = (\sqrt{7})^2$$

Figure 18: **Examples of input formats and model responses from multiple-choice (MC) and free-form (FF) training data.**

# E   PROMPTS

---

**Prompt for brainstorming the fine-grained criteria and patterns**

You are tasked with analyzing the reasoning strategies used in the following response. The response includes the thought process for solving a problem. Your goal is to extract and describe patterns based on various criteria that characterize the model's problem-solving strategy.

Please follow these guidelines:

1. Identify multiple *meaningful criteria* that differentiate reasoning strategies. Each criterion should have a clear and descriptive name that reflects a real aspect of the reasoning process. **Do not use generic placeholders like 'Criterion 1'.**

2. For each criterion, describe two contrasting *pattern types* (e.g., *Step-by-step* vs. *Outcome-first*, or *Concrete* vs. *Abstract*).

3. Present your analysis in the following format, using <patterns> and </patterns> tags to enclose the list:

<patterns>
Descriptive Criterion Name (Pattern A vs. Pattern B)
Descriptive Criterion Name (Pattern A vs. Pattern B)
...
Descriptive Criterion Name (Pattern A vs. Pattern B)
</patterns>

4. Do not include any explanations or commentary within the <patterns> tags.

5. The example format above is only a guide. You are encouraged to define your own diverse and insightful pattern criteria based on the given response.

Response: {answer}

---

Figure 19: **Prompt template for identifying fine-grained reasoning criteria.** This template guides the systematic extraction of diverse reasoning strategies from model responses by instructing the analysis to: (1) identify meaningful criteria with descriptive names that reflect genuine aspects of reasoning, (2) describe contrasting pattern types for each criterion, (3) present the analysis in a structured format using designated tags, and (4) focus on identifying insightful pattern criteria without extraneous commentary. This structured approach enables comprehensive taxonomy development of reasoning strategies employed in chain-of-thought processes.

---

**Prompt for generating the pattern analysis rubric**

Create a concise rubric for the following reasoning strategy criterion:

[CRITERION NAME]: [PATTERN A] vs. [PATTERN B]

For each pattern, provide:

1. A clear, concise definition (2-3 sentences) that captures the essence of this reasoning strategy
2. 3-4 key characteristics that distinguish this pattern
3. 2 concrete examples of responses that demonstrate this pattern (keep examples brief, about 2-3 sentences each)

Focus on making the distinctions between patterns clear and easily identifiable. The definitions and examples should help evaluators quickly categorize model responses without ambiguity.

---

Figure 20: **Prompt template for rubric generation in reasoning strategy analysis.** This template outlines the structured approach for creating assessment rubrics that distinguish between contrasting reasoning strategies. The prompt requests (1) concise definitions capturing each pattern's essence, (2) key distinguishing characteristics, and (3) concrete response examples demonstrating each pattern. This systematic rubric development ensures clear pattern differentiation, enabling consistent and unambiguous classification of model reasoning strategies across different problem-solving contexts.

---

**Prompt for generating the pattern analysis report**

You are an expert at analyzing reasoning strategies in model responses. You'll be provided with:
1. A rubric describing two distinct reasoning strategies (Pattern A and Pattern B)
2. A model response to analyze

Your task is to create a detailed analysis report that determines which pattern the response exhibits.

Analysis Process:
1. Carefully examine the response against both pattern definitions in the rubric
2. Identify specific elements, structures, and linguistic features in the response that align with either pattern
3. Note any mixed signals or elements that span both patterns
4. Determine which pattern (A or B) the response most closely matches

Report Structure:
1. **Initial Observations** (2-3 sentences summarizing key features of the reasoning approach)
2. **Evidence for Pattern A**:
- If applicable, quote 1-2 specific segments from the response that demonstrate Pattern A
- Explain how these segments match characteristics described in the rubric
3. **Evidence for Pattern B**:
- If applicable, quote 1-2 specific segments from the response that demonstrate Pattern B
- Explain how these segments match characteristics described in the rubric
4. **Pattern Determination**:
- Explain which pattern (A or B) is most dominant and why
- Address any aspects that show characteristics of both patterns
5. **Conclusion**:
- Clearly state the final pattern determination using the format: "Final pattern determination: [PATTERN NAME]"

Focus on concrete evidence from the response that matches specific elements from the rubric patterns.

Rubric: {rubric}

Response to analyze: {response}

---

Figure 21: **Prompt template for comprehensive reasoning strategy analysis report generation.** This template guides the systematic evaluation of model responses against predefined reasoning strategies. It establishes a structured analytical process involving (1) careful examination of responses against pattern definitions, (2) identification of pattern-aligned elements, (3) recognition of mixed pattern signals, and (4) determination of the dominant pattern. The prescribed report structure—including initial observations, evidence documentation for each pattern, pattern determination with justification, and a clear conclusion—ensures thorough and evidence-based classification of reasoning strategies employed in model outputs.

---

**Prompt for evaluation on multi-choice QA benchmarks**

Question : {question}
Options :
A) {option A }
B) {option B }
...
Please reason step by step, and you should write the correct option alphabet within boxed{}.

---

Figure 22: **Prompt template for multiple-choice question assessment with structured reasoning.** This template presents a standardized format for evaluating reasoning models on multiple-choice QA benchmarks (GPQA-Diamon, MMLU-Redux).

---

**Example prompt for steering models toward optimal reasoning strategies**

You are required to solve the following question using a specific reasoning strategy. This reasoning strategy must guide the entire problem-solving approach:

Top-down: First, conceptualize the overall structure or system involved in the problem. Begin by forming a clear, high-level understanding of the task, and then systematically break it down into lower-level details. Organize your reasoning from general principles to specific elements, prioritizing clarity and structure.

Broad: Start by openly exploring a wide range of possibilities without attempting to prioritize or filter them too early. Your goal is to understand the full landscape of potential approaches, ideas, or interpretations before narrowing down or making decisions.

Inductive: Base your reasoning on concrete examples or specific observations. Look for patterns that emerge from these instances and use them to build general principles or conclusions. Learning through data-driven exploration or trial-and-error is encouraged.

Single: Choose one hypothesis, method, or line of reasoning to pursue. At each step, verify whether your current approach works. If it does not, adjust accordingly before proceeding. Your reasoning should follow a step-by-step method with a focus on systematic validation.

Hypothesis-driven: Approach the problem with a specific, predefined hypothesis. Your task is to verify this hypothesis using logical reasoning and data. The structure of your analysis should aim to confirm or refute the initial claim without deviating from it.

Immediate: Seek full clarity at the beginning. Ask direct and comprehensive questions immediately to form a complete understanding of the problem. Avoid backtracking or revisiting earlier assumptions unless absolutely necessary.

Make sure your entire reasoning process aligns with the selected pattern above. Now, solve the following question accordingly.

{question}

---

Figure 23: **Example prompt for steering models toward optimal reasoning strategies.** This template demonstrates how specific reasoning strategies can be explicitly prescribed to guide problem-solving approaches. Six distinct reasoning strategies are presented: top-down, broad, inductive, single, hypothesis-driven and immediate. The prompt instructs models to align their entire problem-solving process with the selected pattern, enabling controlled experiments to identify which reasoning strategies yield optimal performance across different tasks and benchmarks.

# F  QUALITATIVE ANALYSES

Beyond quantitative analysis, we conduct qualitative analysis based on the actual outputs generated by the model. In Appendix F.1, we analyze the artifacts produced at each step of CoT ENCYCLOPEDIA on the GPQA-Diamond benchmark, comparing the final reasoning strategy analysis results with analysis results classified according to pre-defined criteria. Appendix F.2 presents a comparative analysis of responses from models trained on multiple-choice format datasets versus those trained on free-form datasets when evaluated on the Arena Hard benchmark, along with pattern analysis of these responses.

## F.1  COMPARATIVE ANALYSIS OF CoT ENCYCLOPEDIA OUTPUTS AND PRE-DEFINED CRITERIA

Table 19: **Comparative analysis of reasoning strategies exhibited in physics problem-solving using multiple evaluation frameworks.** This table presents a detailed qualitative analysis of reasoning approaches employed when solving a physics problem of GPQA-Diamond on light propagation through moving media. The analysis employs two distinct evaluation methodologies: the CoT Encyclopedia criteria, which identifies evidence-driven reasoning, and a pre-defined criteria system that assesses backtracking behaviors.

---

**Question:** What is the correct answer to this question:A light beam is propagating through a glass with index of refraction n. The glass is moving at constant velocity v in the same direction as the beam and toward the observer in laboratory. What is the speed of light in glass relative to the observer in laboratory? Take the speed of light in vacuum c=1. Choices: A. 1 B. (1+n*v)/(n+v) C. (1+n*v)/(n-v) D. (1-n*v)/(n+v) Answer:

---

**Response:** Okay, so the question is about the speed of light in glass as observed in a laboratory frame, where the glass itself is moving at a constant velocity v in the same direction as the light beam, and towards the observer. The speed of light in vacuum is given as c=1. The choices are A through D, with various expressions involving n and v. First, I need to recall how the speed of light in a medium relates to the medium's refractive index. The speed of light in a medium with refractive index n is given by v_light = c/n. But here, the medium (glass) is moving. So, the usual formula might need some adjustment due to the motion of the medium. Hmm, maybe we have to consider the velocity addition formula from special relativity, but I'm not sure if that directly applies here. Let me think. Wait, ... (omitted) ...

---

2. Fresnel Drag Formula: The formula for the speed of light in a moving medium is given by:

$$u = \frac{\frac{c}{n} + v}{1 + \frac{v}{nc}}$$

Substituting $c = 1$ into the formula:

$$u = \frac{\frac{1}{n} + v}{1 + \frac{v}{n}}$$

3. Simplification: Simplify the expression by multiplying the numerator and denominator by $n$:

$$u = \frac{\left(\frac{1}{n} + v\right)n}{\left(1 + \frac{v}{n}\right)n} = \frac{1 + nv}{n + v}$$

4. Relativistic Velocity Addition: Alternatively, using the relativistic velocity addition formula, where $u' = \frac{1}{n}$ is the speed of light in the glass frame moving at velocity $v$ relative to the lab frame:

$$u = \frac{u' + v}{1 + u'v}$$

Substituting $u' = \frac{1}{n}$:

$$u = \frac{\frac{1}{n} + v}{1 + \frac{v}{n}}$$

This simplifies to the same expression as above:

$$u = \frac{1 + nv}{n + v}$$

5. **Conclusion**: The derived expression matches option B.

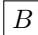

$$\boxed{B}$$

---

**Fine-grained criteria:** "Understanding Method (Conceptual Insight vs. Formula Application)", "Problem Fragmentation (Deconstruction vs. Holistic Approach)", "Validation Strategy (Cross-Verification vs. Single-Source Verification)", "Solution Derivation (Analytic Derivation vs. Plug-and-Chug)", "Exploratory Process (Iterative Consideration vs. Direct Pursuit)", "Logic Application (Physics Principles vs. Mathematical Manipulation)", "Interpretation Mode (Qualitative Analysis vs. Quantitative Analysis)"

---

**Sampled CoT ENCYCLOPEDIA criteria based report:** The provided response demonstrates a strong understanding of the physics involved in determining the speed of light in a moving medium (in this case, glass) relative to a laboratory observer. The thinker systematically recalls and applies relevant concepts, including the relationship between the speed of light and the refractive index, and utilizes the Fresnel drag coefficient and the relativistic velocity addition formula correctly.

The reasoning starts with an exploration of how the speed of light in a medium relates to its refractive index and adjusts for motion, showing a thorough investigation of the problem rather than an assumption-driven approach. The responder considers multiple avenues of solution and confirms their correctness through simplification and careful algebra, ultimately arriving at the same conclusion using different methods.

The exploration and hypothetical testing of ideas are evident, with the responder not merely following a structured path dictated by a singular hypothesis, but rather engaging in an open-ended inquiry that leads to the solution through evidence-based reasoning.

Given that the response is grounded in evidence from physics concepts and engages with the problem dynamically, it aligns best with the "Strongly Evidence-driven" pattern of reasoning. The validity of the conclusions is drawn from examining the data and relevant principles without imposing an initial hypothesis that restricts the analysis.

Thus, the response fits into the pattern of strong evidence-driven reasoning. Therefore, I will categorize it as such: Strongly Evidence-driven

---

**Sampled pre-defined criteria based report:** The provided response exhibits a clear pattern of backtracking, as the individual systematically revisits earlier thoughts and calculations to confirm and refine their understanding of the problem. Throughout the response, the author reflects on different aspects of the problem, considers potential adjustments, and verifies the correctness of their reasoning by retracting or re-evaluating previous conclusions.

1. The author starts by recalling foundational concepts related to light in a medium and how it relates to refractive index and motion. 2. They initially consider the implications of special relativity on the motion of light in a medium. 3. As they derive the formula for the speed of light in a moving medium, they continuously verify their calculations making adjustments and confirming expressions until they are confident in the answer. 4. They also show awareness of different factors (like the direction of motion) that could affect their calculations, demonstrating a reflective process where they reassess the implications of their findings.

This approach aligns closely with the description of "Backtracking" in the rubric: the reasoning involves revisiting and reassessing previous decisions to arrive at a correct conclusion. This indicates the presence of backtracking rather than a strictly linear thinking process.

Thus, the evaluation of the response is that it demonstrates backtracking.

Final evaluation: Yes

---

Table 20: **Comparative analysis of reasoning strategies exhibited in physics problem-solving using multiple evaluation frameworks.** This table presents a comparative analysis of reasoning strategies exhibited when solving a complex organic chemistry problem of GPQA-Diamond involving multiple synthesis steps. The analysis employs two complementary evaluation frameworks: the CoT Encyclopedia criteria, which identifies iterative clarification processes during reaction analysis, and a pre-defined criteria system that focuses on backtracking behaviors.

**Question:** What is the correct answer to this question:A methanol solution of (R)-(+)-Limonene is stirred with Pd/C under a Hydrogen atmosphere. After 1 equivalent of hydrogen is consumed, product 1 is isolated as the major product.

1 is treated with 3-chloroperbenzoic acid, forming product 2. Product 2 is treated with sodium methoxide, forming product 3. Product 3 is treated with propanoic acid, dicyclohexylcarbodiimide. and a catalytic amount of 4-dimethylaminopyridine, forming product 4. what is a valid structure of product 4? (product 4 exists as a mixture of isomers. the correct answer is one of them). Choices: A. 1-methoxy-2-((S)-4-methylcyclohex-3-en-1-yl)propan-2-yl propionate B. (1S,2S,5R)-5-isopropyl-2-methoxy-2-methylcyclohexyl propionate C. (1S,2S,4R)-4-isopropyl-2-methoxy-1-methylcyclohexyl propionate D. (1S,2R,4R)-4-isopropyl-2-methoxy-1-methylcyclohexyl propionate Answer:

**Response:** Okay, let's tackle this question. So, the problem is about identifying the chemical formula of an unknown compound using FTIR, 1H NMR, and Mass Spectrometry data. The given choices are C6H12O, C6H10O, C6H10O2, or C6H12O2. The options are A to D corresponding to these formulas. First, let's parse the FTIR data. The FTIR spectrum shows a very broad absorption peak at 3000 wavenumbers. ... (omitted) ... Wait, but let's check the other options. If the compound had only one oxygen (option B, C6H10O), how could that be? For example, if it's a ketone with a double bond. A ketone has one oxygen. But then where does the O-H come from? The question mentions a broad peak at 3000 cm, which is O-H. So if the compound is a ketone with an O-H group, that would require two oxygen atoms: one in the carbonyl (C=O) and one in the O-H. But option B is C6H10O, which has only one oxygen. So that's impossible. Therefore, the answer must be A. Another angle: the degrees of unsaturation. The compound has two degrees of unsaturation (from the analysis above: carbonyl and double bond). The formulas with two degrees are A and B. Since B has only one oxygen, but the compound must have O-H (which needs an additional oxygen beyond the carbonyl), so B is out. Therefore, A is correct. Thus, the answer should be

$$\boxed{A}$$

**Fine-grained criteria:** "Problem Decomposition (Stepwise Breakdown vs. Holistic Assessment)", "Data Interpretation (Spectrum Analysis vs. Literature Reference)", "Functional Group Identification (Empirical vs. Theory-Driven)", "Alternatives Consideration (Exhaustive Options vs. Targeted Options)", "Verification Focus (Spectral Data Consistent vs. Fragment Peak Analysis)", "Iterative Refinement (Initial Hypothesis vs. Re-evaluated Hypothesis)"

**Sampled CoT ENCYCLOPEDIA:** The response demonstrates a highly iterative approach to clarification. Throughout the analysis, the responder explores various possibilities concerning the chemical structure of the unknown compound using FTIR, 1H NMR, and Mass Spectrometry data. Key indicators of this iterative approach include: 1. **Gradual Exploration**: The responder goes through the spectral data step-by-step, interpreting each piece of information in relation to possible structures for the unknown compound. This indicates a willingness to engage with uncertainty and adapt their understanding as more data is considered. 2. **Follow-Up Questions and Refinement**: The responder frequently asks rhetorical questions (e.g., "Wait, but...") that indicate reconsideration of previous assumptions and re-evaluating information based on new insights. They continuously refine their interpretations and reasoning as they progress. 3. **Comfort with Ambiguity**: The presence of various interpretations throughout (e.g., the potential for a carboxylic acid, possible structures aligning with the degrees of unsaturation, and the consideration of different functional groups) reflects a methodical approach to making sense of complex chemical data, without rushing to conclusions. 4. **Incremental Understanding**: The analysis reveals that the responder builds their conclusion based on progressively integrating different aspects of the data rather than expecting an immediate resolution. They discuss multiple scenarios and gradually hone in on a specific interpretation that fits the data best. Thus, the response exemplifies a mindset that validates iterative clarification, as the responder systematically processes the information and adjusts their conclusions in an exploratory manner. Therefore, the reasoning strategy for this response is Highly Iterative Clarification.

**Sampled pre-defined criteria based report:** The given response clearly exhibits the characteristics of the "Backtracking" reasoning strategy. Throughout the analysis, the author revisits earlier decisions and thought processes, particularly in relation to the interpretation of spectral data and the implications for the chemical structure of the unknown compound. For instance, the author considers the implications of the FTIR and NMR spectra, reassessing conclusions about potential functional groups and how they relate to the chemical formulas provided. The response's flow reflects a significant amount of reconsideration of hypotheses, especially when determining the degrees of unsaturation and confirming the presence of specific functional groups like the O-H group. The author alternates among possibilities and systematically eliminates options based on newly recalled information and analysis, which is central to the backtracking reasoning strategy. Every time the author faces a potential inconsistency or lacks certainty, they pause to reevaluate their conclusions indicating a methodical return to earlier ideas to ensure the accuracy of their answer. This iterative process is a hallmark of backtracking rather than a one-way reasoning stream. Based on this thorough examination, it is evident that the reasoning adheres to the backtracking structure. Thus, the evaluation is yes.

## F.2 DIVERGENT REASONING STRATEGIES FROM MULTIPLE-CHOICE VS. FREE-FORM TRAINING DATA

Table 21: **Divergent reasoning strategies in models trained on multiple-choice versus free-form data formats.** This table shows how training data format shapes reasoning strategies when solving identical problems of Arena-Hard benchmark. Multiple-choice trained models exhibit structured problem decomposition with frequent hypothesis generation and multiple restart points, while free-form trained models pursue single hypotheses with continuous verification within one framework. This analysis supports our finding that format characteristics impact reasoning strategies more significantly than problem domain.

**Question:** A mother buys a set of $N$ toys for her two children, Alice and Bob. She has already decided, for each toy, whether it will go to Alice or to Bob, but she has forgotten the actual monetary values of the toys. She only remembers:

- The toys were arranged in *ascending* order of value.
- All prices are non-negative.

We say that a given assignment is *fair* if, for *every* non-decreasing sequence of toy-values

$$0 \leq v_1 \leq v_2 \leq \cdots \leq v_N,$$

the difference between the total value of Alice's toys and Bob's toys does not exceed the maximum toy value $v_N$.

Formally, let

$$S = S_1 S_2 \ldots S_N$$

be a binary string of length $N$, where

$$S_i = \begin{cases} 1, & \text{if toy } i \text{ is assigned to Alice,} \\ 0, & \text{if toy } i \text{ is assigned to Bob.} \end{cases}$$

Then $S$ is called *fair* if for every non-decreasing sequence $(v_i)_{i=1}^{N}$ with $0 \leq v_1 \leq v_2 \leq \cdots \leq v_N$, we have

$$\left| \sum_{i=1}^{N} v_i \left[ S_i = 1 \right] \; - \; \sum_{i=1}^{N} N v_i \left[ S_i = 0 \right] \right| \; \leq \; v_N,$$

where

$$[P] = \begin{cases} 1, & \text{if predicate } P \text{ is true,} \\ 0, & \text{otherwise.} \end{cases}$$

---

**Multiple-choice format (MC) trained:** Okay, so I'm trying to figure out how to determine if a given distribution of toys is fair based on the problem statement provided. Let me break down the problem step by step. First, the problem says that there are N toys, and they've been assigned to Alice and Bob in some way specified by a binary string S. Each character in S represents one of the toys: '1' means Alice gets it, '0' means Bob gets it. The key thing here is that S tells us exactly which toys go to whom, but the actual values of these toys aren't specified. However, the problem gives us a condition that must hold for the distribution to be considered "fair." The fairness condition states that for any possible assignment of toy values (which are non-negative and in ascending order), the absolute difference between the total value of toys Alice receives and the total value Bob receives should not exceed the highest toy's value, which is v_N. But I'm not entirely sure about that. **Let me think about how to approach this. One way to tackle this is to consider the worst-case scenario where the difference between Alice's and Bob's totals could be maximized.** If even in the best possible worst-case scenario, the difference doesn't exceed v_N, then the distribution is fair. But wait, the problem mentions that the distribution is fair if for every possible array of toy values, the difference doesn't exceed v_N. That seems quite broad because it implies that no matter how the toy values vary (as long as they're non-decreasing), the difference shouldn't get too big. ... (omitted) ... **Let me think differently.** ... (omitted) ... **I need to find a different approach.** Explanation - **Reading Input**: The code reads the number of test cases and processes each test case individually. - **Counting Toys**: For each distribution, it counts how many toys Alice receives ('k'). - **Calculating Sums**: It calculates the sum of the values of the top 'k' toys and the sum of the values of the bottom 'N - k' toys. - **Checking Fairness**: It computes the absolute difference between these sums and checks if it is less than or equal to the highest toy value. If it is, the distribution is deemed fair; otherwise, it is not. This approach ensures that we efficiently determine the fairness of the distribution by leveraging the properties of non-decreasing toy values and the constraints imposed by the problem.

**Free-form (FF) trained:** Okay, so I'm trying to solve this problem where I have to determine if a given distribution of toys between Alice and Bob is fair based on certain conditions. Let me try to understand the problem step by step. So, the problem says that a distribution is fair if, regardless of how the actual toy prices (which are in strictly increasing order) are assigned, the absolute difference between the sums of Alice's toys and Bob's toys doesn't exceed the maximum toy price among them. **Wait, more formally**, for every possible array of toy prices $v\_1 <= v\_2 <= ... <= v\_N$, the absolute difference $|sum\_A - sum\_B|$ must be $<= max\_v$, where $max\_v$ is the largest toy price. **Hmm, wait**, because the $max\_v$ is the last element in the sorted list. So $max\_v = v\_N$, since they're in order. **But** the condition needs to hold for all possible assignments of $v\_i$'s. That makes sense because the distribution is fair no matter what the actual prices are so even the worst-case scenario must satisfy the condition. Our task is, given S (the binary string indicating which toys go to whom), to decide whether this condition holds for all possible toy price arrays. I think the key here is to find out under what circumstances the maximum possible $|sum\_A - sum\_B|$ across all possible v arrays is less than or equal to $v\_max$. ... (omitted) ... Explanation - **Reading Input**: The input is read in one go for efficiency, especially given the constraints. - **Prefix Sum Calculation**: For each test case, we calculate the prefix sums of the values of toys Alice and Bob receive. Here, the values are assumed to be sequential integers starting from 1. - **Fairness Check**: Using the derived condition, we check if twice the sum of the values of the toys Alice receives is less than or equal to the total sum of all toy values. If this condition holds, the distribution is fair; otherwise, it is not. This approach efficiently determines the fairness of the distribution by leveraging prefix sums and ensuring the solution handles up to the maximum constraints effectively.

**Pattern analysis for MC:** ... The response is more structured and attempts to break down the problem logically. ... It attempts to break down the problem ... model attempts to provide a clear step-by-step guide. Model also tends to generate multiple hypotheses, demonstrating an inclination toward broadly exploring potential answers.

**Pattern analysis for FF:** ... response is lengthy and somewhat convoluted. It goes through a lot of reasoning but does not arrive at a clear solution or code implementation. ... does not provide a clear answer to the problem or a working code solution. It gets lost in the reasoning without concluding with a practical implementation. ... The response is excessively verbose, making it difficult to extract useful information quickly. Model exhibits a tendency to pursue answers by continuously verifying hypotheses within a main framework

Table 22: **reasoning strategies in error diagnosis: Multiple-choice versus free-form trained models.** This table compares how models trained on different data formats approach technical troubleshooting on question of Arena-Hard benchmark. The multiple-choice trained model demonstrates structured analysis with clear step-by-step reasoning and actionable recommendations, while the free-form trained model exhibits a more repetitive pattern with frequent restating of observations. This comparison highlights how training format influences problem-solving strategies in diagnostic contexts.

---

**Question:** Find root cause for this error: sshd[54785]: error: kex_exchange_identification: Connection closed by remote host

---

**Multiple-choice format (MC) trained:** Okay, so I need to figure out the root cause of the error "kex_exchange_identification: Connection closed by remote host" in a KEX application. **Let me break this down step by step. First,** I understand that KEX is a protocol used in blockchain technologies, often in contexts like Solana, Monero, or others that use zero-knowledge proofs. The error message suggests that when the application tries to connect to KEX, it's closing the connection, which happens on a remote host. That makes me think about how the application is handling network connections. **I remember that in many networked applications**, especially those involving distributed systems, it's crucial to handle connections properly. ... (omitted) ... **I can consider the possibility** of a transient network issue causing the connection to be closed. This could include temporary network outages or fluctuations in the network. Implementing a retry mechanism or error handling in the application could provide a fallback plan in case of unexpected network issues. **In summary,** the root cause is likely related to the application not handling network connections properly, possibly by not waiting for network confirmation or by not checking the status of the remote host before attempting a connection. To fix this, the application should ensure that network status is checked, connections are confirmed before sending requests, and appropriate error handling is in place. ... (omitted) ... **By following these steps,** the application can better handle network connectivity issues, ensuring that connections are not closed unexpectedly. This approach will involve code adjustments, network status checks, and error handling mechanisms to provide a more reliable connection process.

---

**Free-form (FF) trained:** The error occurs when I run the code as follows: So, the error is that kex_exchange_identification is closed by a remote host. So, the code is trying to get a new exchange (kex) from a remote host, **but it's not working because the connection to the remote host is closed.** I think that the problem is that the code is not correctly handling the authentication for the exchange. ... (omitted) ... Looking at the code, I see that the exchange is being fetched with the 'exchange' parameter. **But perhaps the way it's being handled is incorrect. Wait,** in the code, the exchange is being fetched using exchange = get_kex_exchange(exchange). **But in the code,** it's not passing any headers or authentication parameters. **Wait, in the code,** the 'exchange' is being fetched without any headers or with the wrong parameters.

---

**Pattern analysis for MC:** ... provides a structured and relevant analysis of the error. It identifies potential causes and offers a clear troubleshooting approach, including checking network status, allowing for network confirmation, and implementing error handling ... The response is helpful and concise, directly addressing the user's prompt and providing actionable steps to resolve the issue ...

---

**Pattern analysis for FF:** ... response is largely verbose. It does not provide a clear explanation of the error or its potential causes. ... Instead, it seems to get lost in a repetitive and unclear narrative about code handling without addressing the actual error message or providing actionable troubleshooting steps.

---

