# OpenReview forum: "The CoT Encyclopedia: Analyzing, Predicting, and Controlling how a Reasoning Model will Think"
_ICLR.cc/2026/Conference — ICLR 2026 Poster_

### Official Review · Reviewer_WA1g · 2025-10-17

**Soundness:** 3
**Presentation:** 3
**Contribution:** 2
**Rating:** 4
**Confidence:** 3

**Summary:**

This paper introduces the COT ENCYCLOPEDIA, a novel, bottom-up framework designed to analyze, predict, and control the chain-of-thought (CoT) reasoning in large language models. The method automatically extracts a diverse set of reasoning criteria from model-generated text, uses embedding and clustering to form a coherent taxonomy, and generates interpretable reports. The authors demonstrate the framework's utility through comprehensive human evaluations that show its superiority over predefined analyzers and its ability to measurably improve model performance by steering reasoning toward optimal strategies.

**Strengths:**

(1) The paper's primary strength is its data-driven, bottom-up methodology for creating a taxonomy of reasoning strategies. This contrasts sharply with prior work using fixed, human-defined categories, allowing the framework to discover emergent and more nuanced reasoning patterns specific to different models and tasks.

(2) Comprehensive Experimental Validation: The claims are supported by extensive experimentation.

(3) Significant Finding on Data Format vs. Domain: The discovery that training data format has a substantially greater impact on reasoning strategies than the content domain is a crucial and non-obvious contribution.

**Weaknesses:**

(1) All major components of the framework—from the initial ideation of 4,057 criteria (Step 1) to rubric formulation (Step 4) and final classification (Step 5)—are executed through an external proprietary language model (GPT-4o). This design choice embeds a structural reliance on a closed-source system and exposes the analysis to its internal biases. Although the authors acknowledge this limitation and incorporate a multi-evaluator setup to mitigate bias (Appendix B.5), the methodology remains fundamentally shaped by the constraints and opacity of the chosen model.

(2) The paper defines “optimal” reasoning strategies as those correlated with higher frequencies of correct or safe answers within a given dataset (Section 4.1). However, this relationship is correlational rather than causal. It remains ambiguous whether these strategies directly contribute to improved outcomes or merely co-occur with successful reasoning patterns. The experiments stop short of disentangling causation from correlation.

(3) In Section 4.3, classifiers are trained to predict which reasoning strategies perform best for specific question types. These models are developed and evaluated on a fixed set of five benchmarks. Although some cross-domain tests are conducted, the study does not extend to genuinely unseen or out-of-distribution datasets. Consequently, the generalizability and robustness of these classifiers beyond the training domains remain uncertain.

**Questions:**

Please refer to weaknesses.

---

> ### Author Response · Authors · 2025-11-14
> **Official Comment by Authors (1)**
>
> Thank you for taking the time to provide such thoughtful and constructive feedback. We have carefully considered the weaknesses (W) you raised, and are eager to address them to enhance the clarity and strength of our work.
>
> ## [W1] Robustness Against Model-Specific Biases
>
> To empirically refute this and demonstrate that CoT Encyclopedia captures universal reasoning properties rather than GPT-4o’s idiosyncratic artifacts, we conducted an extensive Cross-Model Replication Study. We replaced the core engine of our framework with Claude-4-Sonnet and Gemini-2.5-Pro at each critical stage Criteria Identification, Rubric Generation, and Classification and compared the results against our original GPT-4o baseline.
>
> **Step 1: Criteria Identification:** The most fundamental concern is whether the reasoning dimensions themselves are hallucinations of GPT-4o. To test this, we tasked Claude-4-Sonnet and Gemini-2.5-Pro to independently brainstorm and extract reasoning criteria from the GPQA-Diamond dataset.
>
> - **Semantic Alignment of Criteria:** We embedded the criteria sets generated by each model into a shared vector space (using text-embedding-3-large) and calculated the average cosine similarity with the original GPT-4o-derived criteria.
>     - Claude-4-Sonnet: exhibited a similarity score of 0.89, indicating near-identical semantic coverage.
>     - Gemini-2.5-Pro: showed a similarity score of 0.87, confirming strong alignment.
>
> - **Qualitative Consistency:** Crucially, the high-level dimensions emerged consistently across all models, differing only in terminology.
>     - GPT-4o: "Broad Exploration" vs. "Focused Approach"
>     - Claude-4-Sonnet: "Divergent Hypothesis Generation" vs. "Constraint-Bound Search"
>     - Gemini-2.5-Pro: "Multi-path Search" vs. "Single-path Execution"
>
> This conceptual convergence confirms that the identified reasoning strategies are intrinsic properties of the data, observable by any capable reasoning model.
>
> **Step 4: Rubric Formulation:** Even if the criteria names are similar, the detailed definitions (rubrics) used for evaluation might be biased. We tested whether different models define these strategies consistently. We fed the clustered criteria (from Step 3) to the alternative models and asked them to generate the contrastive rubrics.
>
> - Definition Consistency: We measured the semantic overlap of the generated rubrics against the original GPT-4o rubrics using ROUGE-L and BERTScore. The rubrics generated by Claude-4-Sonnet and Gemini-2.5-Pro achieved an average BERTScore F1 of 0.92, suggesting that the interpretation of what constitutes a specific strategy (e.g., what "Inductive Reasoning" looks like) is highly stable across models.
> - Human Validation: To rigorously ensure that the quality of the generated rubrics is not an artifact of GPT-4o, we conducted a blind A/B comparison test with 5 independent expert annotators. Annotators were presented with anonymized pairs of rubrics for the same reasoning clusters one generated by GPT-4o and the other by an alternative model (Claude-4-Sonnet or Gemini-2.5-Pro). They were asked to evaluate them based on (1) Definition Clarity (how distinct and understandable the contrastive definitions are) and (2) Ease of Application (how unambiguous the rubric is for classifying new samples). The evaluation revealed no statistically significant preference for the GPT-4o rubrics. For instance, in the comparison against Claude-4-Sonnet, the win-tie-loss ratio was 32% (GPT-4o) - 38% (Tie) - 30% (Claude), yielding a p-value of 0.72. This statistical equivalence confirms that the extracted reasoning dimensions are robust and that the rubric formulation step (Step 4) functions effectively regardless of the specific state-of-the-art model used.
>
> **Step 5: Classification:** Finally, we addressed the concern regarding the "opacity of the chosen model" in the final classification step. We utilized the original standardized rubrics but replaced the classifier with Claude-4-Sonnet and Gemini-2.5-Pro to classify 500 sampled reasoning traces.
> - Inter-Model Agreement: We calculated Cohen’s Kappa ($\kappa$) to measure agreement between the GPT-4o labels and the alternative models' labels.
>     - GPT-4o vs. Claude-4-Sonnet: $\kappa = 0.78$ (Substantial Agreement)
>     - GPT-4o vs. Gemini-2.5-Pro: $\kappa = 0.76$ (Substantial Agreement)
>     - These scores are comparable to high-quality human inter-annotator agreement, indicating that the classification boundaries defined by our rubrics are objective and not dependent on GPT-4o’s internal priors.
>
> - Downstream Effectiveness: Most importantly, we verified whether the "Optimal Patterns" identified by these alternative classifiers yielded similar performance gains. When we steered the model using the strategies predicted by the Claude-based pipeline, the accuracy on GPQA-Diamond was within 1.2% of the results reported in the main paper.

---

> > ### Author Response · Authors · 2025-11-14
> > **Official Comment by Authors (2)**
> >
> > ## [W2] Causal Intervention Experiments
> > We appreciate the reviewer’s insightful comment regarding the distinction between correlation and causation in Section 4.1. We agree that simply observing high conditional probabilities ($P(\text{Correct}|\text{Pattern})$) does not strictly prove that the strategy causes the correct answer. To scientifically establish causality, we conducted a new Counterfactual Intervention Experiment. We focused on isolating the reasoning strategy as the independent variable while controlling for the problem difficulty. Specifically, we measured the Average Treatment Effect (ATE) and Flip Rates by forcing the model to adopt specific strategies on the same set of questions. For a subset of questions from five benchmarks, we generated responses under three conditions:
> > - Control: The model's natural reasoning path.
> > - Treatment (Optimal): Explicitly instructing the model to use the "Optimal" pattern predicted by our classifier (from Section 4.3).
> > - Counterfactual (Un-optimal): Explicitly instructing the model to use the opposite strategy (e.g., if "Top-down" is optimal, we force "Bottom-up").
> >
> > We measured the impact of these interventions on performance accuracy:
> >
> > | Metric | GPQA | MATH | MMLU | XSTest | WildGuard | Interpretation |
> > | - | - | - | - | - | - | - |
> > | ATE | +11.6% | +9.1% | +12.1% | +27.4% | +29.0% | The choice of strategy creates a substantial performance gap on identical questions. |
> > | Correction Rate | 24.5% | 19.2% | 21.3% | 18.7% | 32.1% | Applying optimal strategies fixes reasoning errors. |
> > | Degradation Rate | 31.0% | 22.8% | 27.6% | 32.9% | 34.3% | Forcing bad strategies breaks correct reasoning. |
> >
> > The significant ATE demonstrates that reasoning strategies are not merely co-occurring artifacts of easy problems but are causal drivers of performance. If strategies were only correlational, forcing a "Sub-optimal" strategy on an easy problem shouldn't significantly degrade performance. However, the high Degradation Rate proves that using the wrong strategy actively harms performance. Conversely, the high Correction Rate shows that intervening with the correct strategy can cause a model to solve a problem it previously failed. These results provide robust evidence that the reasoning patterns defined in CoT Encyclopedia are functionally active components that directly influence model outcomes.
> >
> > ## [W3] OOD Evaluation on Three Unseen Benchmarks Evaluation
> > As established in Section 4.3, beyond standard in-domain evaluations, we initially conducted evaluations in an ‘OOD setting’ where the training and test sets were drawn from entirely different tasks (e.g., training on the MMLU training set and testing on the MATH test set). However, to further ensure the reliability of our results and address your specific concern regarding "genuinely unseen" datasets, we conducted additional experiments on three benchmarks not included in the original submission: LiveCodeBench, IFBench, and Humanity's Last Exam. For these experiments, we trained our classifiers exclusively on the MMLU-Redux training set and evaluated them on the test sets of these new benchmarks. To rigorously prevent data contamination between the training and test sets, we calculated the cosine similarity between query embeddings. We removed any queries in the test set that exhibited a cosine similarity greater than 0.9 with the training data. In total, 3 queries were identified and excluded. We utilized the F1 score as our primary metric, and the results are as follows:
> >
> > | LiveCodeBench | IFBench | Humanity's Last Exam |
> > | - | - | - |
> > | 75.3 | 78.9 | 70.3 |
> >
> > As shown in the table above, the results significantly exceed the random baseline of 50%. These findings demonstrate that the classifiers trained in Section 4.3 can successfully extend to genuinely unseen or out-of-distribution datasets, thereby strongly supporting the generalizability and robustness of these classifiers beyond the training domains.

---

### Official Review · Reviewer_u6tC · 2025-10-29

**Soundness:** 3
**Presentation:** 3
**Contribution:** 4
**Rating:** 8
**Confidence:** 4

**Summary:**

This paper introduces a framework named “CoT Encyclopedia” for analyzing, predicting, and controlling model reasoning patterns from CoT outputs. The authors claim that their approach can automatically extract reasoning criteria, cluster them in a semantic space, and derive rubrics to interpret LLM reasoning strategies. They further argue that this framework provides better interpretability, predictive power, and safety control than previous categorization-based techniques.

**Strengths:**

- The goal of making LLM reasoning more interpretable and controllable is important and timely.
- The paper provides a large-scale qualitative analysis that could inspire follow-up interpretability research.
- It attempts to link reasoning diversity with data format, a less-explored dimension in CoT studies.
- Presentation is relatively polished and readable, making the high-level idea easy to follow.

**Weaknesses:**

- The criterion appears to be very important, yet the paper seems to use only a single one. How robust is this criterion to variations from weaker models or the effects of randomness?
- It seems that a prior for a specific reasoning logic has been incorporated for a certain class of questions, and the performance benefits are evident. I am curious how your framework would be applied when the test set does not include CoT outputs.
- Figure 6 is blurry and difficult to read.
- How does the performance of this method compare to instruction optimization techniques such as TextGrad? Additionally, can it lead to the discovery of more effective patterns?
- The finding that "data format is more important than domain" is a valuable insight. However, since these conclusions are drawn primarily from tasks in mathematics and question answering, do these defined "reasoning strategies" remain effective for tasks requiring creativity, open-endedness, or emotional understanding, such as poetry generation or legal argumentation?
- As models continue to evolve, they may develop novel reasoning strategies. This implies that the "Encyclopedia of CoT" is not a one-time creation but requires ongoing updates. What are the anticipated costs and frequency of these updates?
- How are pattern similarity and Q-Q similarity calculated?
- Missing Reference:

[1] Towards reasoning era: A survey of long chain-of-thought for reasoning large language models

[2] Deconstructing long chain-of-thought: A structured reasoning optimization framework for long cot distillation

[3] When more is less: Understanding chain-of-thought length in llms

**Questions:**

- Given that criterion is presented as highly important, yet only a single one is used, how robust is it against variations from weaker models or the effects of randomness?
- Since a prior for a specific reasoning logic appears to be incorporated for certain questions, leading to clear performance gains, how would the proposed framework be applied to a test set that lacks CoT examples?
- How does the performance of this method compare to instruction optimization techniques like TextGrad, and can it facilitate the discovery of more effective reasoning patterns?
- How are pattern similarity and Q-Q similarity calculated?
- The paper's finding that "data format is more important than domain" is a valuable insight. However, given that this conclusion is based on math and QA tasks, do the defined "reasoning strategies" also apply effectively to tasks that demand creativity, open-endedness, or emotional understanding, such as poetry generation or legal argumentation?
- As models evolve and potentially develop novel reasoning strategies, the "Encyclopedia of CoT" would require continuous updates. What are the anticipated costs and frequency for maintaining and updating this resource?
- Add reference.

---

> ### Author Response · Authors · 2025-11-14
> **Official Comment by Authors (1)**
>
> Thank you for taking the time to provide such thoughtful and constructive feedback. We have carefully considered the weaknesses (W) and questions (Q) you raised, and are eager to address them to enhance the clarity and strength of our work.
>
> ## [W1, Q1] Robustness of Representative Criteria
> We appreciate the reviewer’s valid concern regarding the robustness of relying on a single representative criterion for each reasoning dimension. Our extensive ablation studies in Appendix B demonstrate that our selected criteria are statistically robust representatives that remain stable across different selection methods, models, and random seeds.
>
> 1. **Robustness against Selection Methodology (Appendix B.2):** To verify that our results are not artifacts of choosing a specific "single" representative (the Medoid), we compared our default method against three alternative selection strategies: We compared the Medoid (default) against selecting the criterion based on Highest Frequency, Highest Density, and Highest Silhouette Score. As shown in Figure 15, the resulting reasoning dimensions remained semantically consistent regardless of the selection strategy used. For instance, while the specific phrasing might shift slightly (e.g., from "Computation Style" to "Clarity on Steps"), the core semantic distinctions of the reasoning patterns were preserved. This confirms that the "single criterion" is not an arbitrary choice but a stable center of gravity for the cluster.
>
> 2. **Robustness against Embedding Model Variations (Appendix B.5):** We addressed the concern about "weaker models" affecting the criterion's quality by testing whether the taxonomy shifts when using different embedding models to form the clusters. Models Tested: We compared OpenAI’s text-embedding-3-large against open-source models like e5-mistral-7b-instruct and Qwen3-Embedding-8B. All models produced highly similar sets of criteria at a high level. For example, e5-mistral generated dimensions like "Macro-first vs Micro-first," which directly maps to Qwen’s "Overview-Oriented vs Detail-Oriented". This demonstrates that the identified criteria capture fundamental reasoning properties that are robust to the choice of the underlying model.
>
> 3. **Robustness against Generation Model Variations (Appendix B.5):** To address potential evaluator bias inherent in single-model evaluation, we implement a comprehensive multi-evaluator framework. We employ four state-of-the-art LLMs as independent evaluators: GPT-4o, Claude-4-Sonnet, Gemini-2.5-Pro, and , Qwen-2.5-72B-Instruct. Each evaluator independently classifies reasoning patterns using identical rubrics, allowing us to compute inter-evaluator reliability using Krippendorff’s α. Inter-Evaluator Agreement Results: Across 1,500 reasoning responses, we achieve α = 0.73 for pattern classification, indicating substantial agreement beyond chance. Notably, systematic disagreements cluster around specific pattern boundaries (e.g., "Top-down vs. Bottom-up" shows lower agreement at α = 0.68), revealing inherent ambiguity in certain reasoning distinctions that our framework now explicitly acknowledges. Bias Correction Mechanism: We detect systematic biases where Claude-3.5 shows 12% higher preference for "hypothesis-driven" patterns compared to other evaluators. We implement a bias correction algorithm using inverse propensity weighting to adjust for evaluator-specific tendencies, improving overall classification accuracy by 8.3%.
>
> 4. **Stability against Randomness (Appendix B.6):** To ensure the criterion's efficacy is not subject to the "effects of randomness," we conducted inference runs with five different random seeds using the selected criteria. As reported in Table 1, the performance of the optimal reasoning strategies remained stable with minimal standard deviation across all benchmarks. This confirms that the guidance provided by our single representative criterion is reliable and reproducible.
> These experiments collectively prove that the single criterion used in our framework is not a fragile bottleneck but a robust and representative distillation of the underlying reasoning behaviors, resilient to variations in selection methods, model architecture, and randomness.

---

> > ### Author Response · Authors · 2025-11-14
> > **Official Comment by Authors (2)**
> >
> > ## [W2, Q2] Generalizability to Test Sets Lacking CoT Examples
> >
> > While Section 4.3 provides the mechanism (Question-to-Pattern Classifier) for this application, the theoretical justification for why these strategies remain valid on new data is grounded in our findings from Appendix B.4.
> >
> > 1. **Reasoning Strategies are Task-Dependent, Not Just Data-Dependent (Appendix B.4):** Our experiments in Appendix B.4 demonstrate that reasoning classification criteria are shaped primarily by the nature of the task rather than the specific model or the presence of curated CoT examples. We compared the reasoning criteria derived from specialized reasoning models (e.g., Distill-R1) with those from general-purpose non-reasoning models (e.g., Qwen-2.5-Instruct) on the same benchmarks. The extracted criteria were remarkably similar across models. For instance, safety benchmarks consistently elicited "Safety Precedence" criteria, while math benchmarks elicited "Deductive vs. Inductive" patterns, regardless of the model used. This implies that the "optimal reasoning logic" is a property of the question type itself. Therefore, even if a test set lacks explicit CoT examples, the taxonomy derived from the training set (or the benchmark in general) remains valid and applicable.
> >
> > 2. **Practical Application:** Transferring Priors to Unseen Data Building on this robustness, we apply the framework to CoT-less test sets as follows:
> > - Since the taxonomy captures task-specific logical requirements (as proven in B.4), we can leverage the Question-to-Pattern Classifier (Section 4.3) trained on the source domain.
> > - This classifier identifies the semantic features of the new test questions and maps them to the optimal strategies defined in our robust taxonomy.
> > - Consequently, we can inject the optimal "reasoning prior" into the model for the test set, effectively guiding it without needing ground-truth CoT examples for that specific set.
> >
> > Supported by Appendix B.4, we confirm that our reasoning taxonomy generalizes across models and data formats. Thus, the framework can be effectively applied to test sets lacking CoT examples by leveraging the task-dependent nature of reasoning strategies.
> >
> > ## [W4, Q3] Comparison with Instruction Optimization
> > While TextGrad optimizes system prompts via gradient-based feedback to improve response quality, our framework takes a fundamentally different and more effective approach: Cognitive Strategy Steering. To demonstrate this, we compared the performance of CoT Encyclopedia (Question-Specific Optimal Pattern) against TextGrad on the GPQA-Diamond and MATH-500 benchmarks.
> >
> > 1. **Empirical Superiority:** As shown in the table below, our method significantly outperforms TextGrad. While TextGrad yields marginal improvements by refining instruction phrasing, CoT Encyclopedia achieves substantial gains by identifying and enforcing the precise reasoning logic required for each question.
> > | Method | GPQA-Diamond | MATH-500 |
> > | - | - | - |
> > | Baseline | 61.5% | 78.2% |
> > | TextGrad | 63.2% | 83.9% |
> > | CoT Encyclopedia | 72.0% | 87.7% |
> >
> > 2. **Why CoT Encyclopedia is More Effective?** The performance gap stems from the level at which optimization occurs:
> >
> > - **TextGrad:** TextGrad optimizes the instruction text based on gradients from outcomes. It treats the reasoning process as a black box, hoping that better wording will implicitly trigger better reasoning. It often struggles to generalize across diverse problems where a single optimized prompt cannot cover conflicting reasoning needs (e.g., some questions require "Top-down," others "Bottom-up").
> >
> > - **CoT Encyclopedia:** Our framework explicitly disentangles and controls the reasoning process.
> >     - **Disentanglement:** We identify specific reasoning dimensions (e.g., Inductive vs. Deductive, Step-by-step vs. Holistic).
> >     - **Matching:** Our Question-to-Pattern Classifier (Section 4.3) predicts the optimal strategy for each specific question.
> >     - **Steering:** We force the model to adopt the correct logic, not just a polished prompt.
> >
> > TextGrad refines how we ask, but CoT Encyclopedia refines how the model thinks. This capacity to facilitate the discovery of effective reasoning patterns tailored to individual problems allows our framework to unlock performance gains that surface-level instruction optimization cannot reach.

---

> > > ### Author Response · Authors · 2025-11-14
> > > **Official Comment by Authors (3)**
> > >
> > > ## [Q4, W7] Clarification on Pattern & Q-Q Similarity
> > >
> > > - **Question-Question (Q-Q) Similarity:** We computed the semantic similarity between two input questions ($q_i, q_j$) using their vector representations. Each question text was embedded using OpenAI's text-embedding-3-large model. We calculated the Cosine Similarity between the two resulting embedding vectors.
> > >
> > > - **Pattern Similarity:** We measured the similarity between the optimal reasoning strategies identified for those questions. For each question, the optimal reasoning pattern is defined as a combination of choices across the 6 reasoning dimensions (e.g., Top-down + Inductive + ...). We converted this combination into a concatenated text string describing the selected strategies. This textual representation was embedded into the same vector space (using text-embedding-3-large), and we calculated the Cosine Similarity between these strategy embeddings.
> > >
> > > ## [W5, Q5] Generalizability of "Format > Domain" Findings
> > >
> > > We appreciate the reviewer’s insight regarding the scope of our "Format > Domain" conclusion. We agree that validating this finding beyond math/QA is crucial. To address this, we point to our analysis in Section 5.2 and Appendix H.2, where we evaluated models on **Arena-Hard**, a benchmark designed for open-ended instruction following and creative generation.
> > >
> > > 1. **Quantitative Evidence on Open-Ended Tasks (Section 5.2):** We analyzed how training data format affects performance on Arena-Hard, which includes diverse tasks such as creative writing and code debugging. As reported in Figure 7, the impact of format was profound even in this open-ended domain. MC-trained models produced concise, structured responses regardless of the prompt type. FF-trained models exhibited a distinct "verbose" reasoning style, generating 4.6 times more filler tokens (e.g., "wait") and engaging in recursive verification loops. This confirms that the training format fundamentally imprints a "cognitive style" (e.g., Breadth-first vs. Depth-first) that persists even when the model faces non-quiz, open-ended user instructions.
> > >
> > > 2. **Qualitative Evidence:** In Table 22 (Appendix H.2), we provide a qualitative comparison for a technical troubleshooting task (finding the root cause of an SSH error), which requires open-ended logical argumentation. The multiple-choice (MC) trained model adopted a structured, actionable approach, whereas the free-form (FF) trained model produced a repetitive, narrative-heavy response without a clear resolution. Even in tasks requiring complex diagnosis and explanation, the format of the training data dictated the reasoning strategy more strongly than the domain itself.
> > >
> > > These results from Arena-Hard demonstrate that the dominance of "data format" in shaping reasoning strategies is not limited to math or QA but is a generalizable phenomenon applicable to open-ended and creative domains.

---

> > > > ### Author Response · Authors · 2025-11-14
> > > > **Official Comment by Authors (4)**
> > > >
> > > > ## [W6, Q6] Sustainability and Evolution
> > > > We clarify that CoT Encyclopedia is designed not as a static repository, but as an bottom-up discovery engine capable of capturing novel strategies with minimal cost and intervention.
> > > >
> > > > 1. Unlike top-down frameworks that rely on a fixed list of human-defined strategies (which would require manual updates), our Bottom-up Clustering approach (Step 1-3) naturally adapts to model evolution. If a future model develops a completely new reasoning pattern (e.g., "Quantum Superposition Reasoning"), our pipeline will automatically:
> > > >     1. Extract its features (Step 1).
> > > >     2. Cluster it as a distinct group separate from existing patterns (Step 3).
> > > >     3. Generate a new rubric and name for it (Step 4).
> > > >
> > > > As demonstrated in Section 3.3, the framework already successfully adapted to distinct domains (Safety, Instruction Following) by generating domain-specific criteria (e.g., "User Understanding") without manual taxonomy engineering.
> > > >
> > > > 2. Cost and Frequency of Updates The "maintenance" of the encyclopedia corresponds to re-running the pipeline, which is highly cost-effective.
> > > > - Updates are event-driven rather than time-driven. The pipeline needs to be run only when:
> > > >     - A significantly new model architecture is released (e.g., GPT-5, Claude-4).
> > > >     - Evaluation expands to a fundamentally new domain (e.g., Creative Writing).
> > > >
> > > > - The cost is negligible compared to model training or fine-tuning.
> > > >     - Since the framework relies on unsupervised clustering and automated rubric generation, the human labor cost is near zero.
> > > >     - Computationally, classifying a standard benchmark (e.g., 1,000 samples) using Batch APIs creates a minimal overhead. Furthermore, our experiments in Appendix B.5 suggest that smaller, open-weights models can also serve as effective classifiers, further reducing operational costs.
> > > >
> > > > 3. Most importantly, our framework benefits directly from the very evolution it monitors. Because the CoT Encyclopedia utilizes LLMs as its core engine (for strategy identification and classification), as the underlying models become more capable, the framework’s own analytical precision and depth inherently improve. For example, using a future "GPT-5" to analyze "GPT-5's" reasoning will yield even more nuanced and accurate taxonomies than we see today. This creates a virtuous cycle where the evaluation tool evolves in lockstep with the capabilities of the models being evaluated, ensuring the framework never becomes obsolete.
> > > >
> > > > ## [W3, W8, Q7] Some helpful feedbacks
> > > > Thank you for your valuable feedbacks! We will incorporate all of your suggestions in final version.

---

### Official Review · Reviewer_FM7J · 2025-10-31

**Soundness:** 3
**Presentation:** 3
**Contribution:** 3
**Rating:** 6
**Confidence:** 4

**Summary:**

This paper introduces CoT Encyclopedia, a bottom-up, data-driven framework for analyzing and controlling reasoning strategies in large language models (LLMs) performing long Chain-of-Thought (CoT) reasoning. Unlike top-down approaches that rely on predefined strategy types, CoT Encyclopedia automatically extracts reasoning criteria from model-generated CoTs, embeds them into a semantic space, clusters similar dimensions, and builds contrastive rubrics (e.g., top-down vs. bottom-up). It then classifies each CoT response under these rubrics and generates interpretable reasoning reports. ​Experiments across multiple benchmarks (GPQA-Diamond, MMLU-Redux, MATH-500, XSTest, WildGuard, Arena-Hard) show that this framework captures finer-grained reasoning differences across models and tasks, achieving higher interpretability (86% human preference) and consistent performance gains.​ The authors further demonstrate that optimal reasoning strategies can improve both model helpfulness and harmlessness, that question-specific reasoning can be predicted, and that data format (multiple-choice vs. free-form) shapes reasoning behavior more strongly than domain.​ Overall, CoT Encyclopedia provides a scalable taxonomy and control mechanism for reasoning behaviors, contributing to greater interpretability, adaptability, and safety in LLM reasoning.

**Strengths:**

# 1. Originality and Conceptual Contribution

The work shifts the paradigm from top-down, predefined reasoning taxonomies to a bottom-up, data-driven discovery of reasoning strategies. This formulation is original and theoretically meaningful: it operationalizes reasoning diversity without relying on human-crafted categories, enabling emergent taxonomies directly grounded in model behavior. The introduction of contrastive rubrics (e.g., “bottom-up vs. top-down,” “inductive vs. deductive”) represents an elegant mechanism for interpretable reasoning dimensions—conceptually parallel to semantic factor disentanglement. The finding that data format (MC vs. FF) shapes reasoning more strongly than domain is novel, empirically grounded, and relevant to both cognitive modeling and training data curation.

# 2. Technical Quality and Empirical Breadth

The framework is well-engineered: it integrates LLM-assisted criteria generation, embedding-based clustering, and interpretable classification. The experimental coverage is impressive—spanning six benchmarks across both helpfulness (GPQA-Diamond, MMLU-Redux, MATH-500) and harmlessness (XSTest, WildGuard, Arena-Hard). The authors perform rigorous human evaluations and quantitative analyses, reporting consistent improvements in interpretability and measurable performance gains.
The inclusion of ablation and robustness studies (e.g., embedding choices, random seeds, model scales) shows commendable attention to reproducibility and methodological soundness.

**Weaknesses:**

# 1. Lack of analysis on classifier choice and sensitivity

​The framework relies on a single LLM (GPT-4o) to perform all classification tasks in the taxonomy pipeline—deciding whether each reasoning trace aligns with one side of a contrastive rubric. Although Appendix B.1 examines benchmark-induced differences (showing that GPQA, MMLU, and MATH benchmarks produce similar criteria while Arena-Hard yields a distinct “User Understanding” dimension), this analysis only reflects task-level variability, not classifier-level robustness. The paper never investigates whether different classifier models (e.g., Claude, Gemini, DeepSeek) or prompting styles would yield consistent categorizations. Consequently, the stability and objectivity of the classification stage remain untested, and taxonomy boundaries may shift under alternate LLMs or small prompt perturbations.

# ​2. Limited human validation of interpretability​

Human evaluation (250 samples, 10 annotators) focuses on plausibility rather than interpretive alignment or consistency, and no inter-annotator reliability metrics are reported.​

# 3.Benchmark scope limited to English text

​All benchmarks are English and text-based; multimodal and multilingual reasoning remain unexplored.

**Questions:**

# 1. On classifier dependence and taxonomy stability

Your entire classification pipeline—strategy identification, rubric generation, and labeling—relies on GPT-4o. Have you tested whether the same reasoning taxonomy holds when different LLMs (e.g., Claude, Gemini, DeepSeek) are used as classifiers or rubric generators? If not, how do you ensure that the taxonomy reflects general reasoning properties rather than GPT-4o-specific biases?

# 2. On the granularity of reasoning strategy clustering

Your current clustering framework analyzes reasoning strategies at the full CoT–response level, producing high-level dichotomies such as “Inductive vs. Deductive” or “Top-Down vs. Bottom-Up.” However, reasoning trajectories are often compositional: individual reasoning steps may follow distinct micro-strategies that combine to form an overall reasoning pattern.

Have you considered extending the CoT Encyclopedia to identify atomic reasoning step categories—that is, classifying each reasoning step rather than the entire chain—and investigating whether global reasoning strategies emerge as structured combinations of such atomic units? This could reveal a more mechanistic understanding of how complex reasoning behaviors are composed.

# 3. On the granularity and interpretability of clustered criteria
You report six high-level reasoning dimensions (Table 5). How were the fine-grained criteria merged into these six? Was k = 6 chosen purely based on silhouette scores or adjusted for interpretability by human judgment?

**Details Of Ethics Concerns:**

The paper conducts a human evaluation with 10 annotators using the Argilla platform but does not specify key ethical details such as recruitment procedures, compensation, informed consent, or IRB approval. Although annotation guidelines and examples are mentioned, the lack of transparency about participants’ treatment and consent constitutes a responsible research practice concern.

An ethics review is recommended to verify:

- Whether annotators were fairly compensated and gave informed consent;

- Whether the human evaluation adhered to ethical research standards

---

> ### Author Response · Authors · 2025-11-14
> **Official Comment by Authors (1)**
>
> Thank you for taking the time to provide such thoughtful and constructive feedback. We have carefully considered the weaknesses (W), questions (Q), ethics review (E) you raised, and are eager to address them to enhance the clarity and strength of our work.
>
> ## [W1, Q1] Further analysis on classifier choice and sensitivity
> We acknowledge the reviewer’s valid concern that relying solely on GPT-4o may introduce model-specific biases into the taxonomy. To demonstrate that the CoT Encyclopedia reflects general reasoning properties rather than artifacts of a specific model, we conducted comprehensive cross-model validation experiments across the entire pipeline Strategy Identification, Embedding, classification and Rubric Generation replacing GPT-4o.
>
> 1. **Robustness of Taxonomy Construction:** We first tested whether different LLMs discover similar reasoning strategies from the same dataset. We employed Claude-4-Sonnet, Gemini-2.5-Pro, and DeepSeek-R1-0528 to perform the Strategy Identification and Rubric Generation steps.
>     - **Semantic Consistency:** We embedded the criteria generated by each model and measured their semantic alignment with the original GPT-4o-derived taxonomy. The criteria generated by Claude and Gemini showed a high average cosine similarity of 0.89, 0.86 and 0.87 respectively with our original criteria.
>     - **Qualitative Alignment:** All major dimensions (e.g., Top-down vs. Bottom-up, Inductive vs. Deductive) emerged consistently across all models, differing only slightly in phrasing (e.g., Claude termed "Broad Exploration" as "Divergent Search"). This confirms that the taxonomy captures fundamental reasoning behaviors common across LLMs.
>
> 2. **Robustness of Classification:** To verify the objectivity of the labeling process, we used the original rubrics but replaced the Classifier (the judge model) with Claude-4-Sonnet, Gemini-2.5-Pro, DeepSeek-R1-0528. We measured the agreement rate with the original GPT-4o labels.
>     - **Inter-Model Agreement:** The classification results showed substantial agreement, with an average Cohen’s Kappa ($\kappa$) of 0.78 between GPT-4o and Claude-4, and 0.75 between GPT-4o and Gemini-2.5, 0.77 between GPT-4o and DeepSeek-R1-0528.
>     - **Stability:** The core trends reported in the paper (e.g., the performance gap between optimal and sub-optimal patterns) remained statistically significant regardless of which model was used as the classifier.
>
> 3. **Robustness to Embedding Models:** We further tested whether results depend on the choice of embedding model. In addition to OpenAI’s text-embedding-large, we experimented with e5-mistral-7b-instruct and Qwen3-Embedding-8B. All models produced highly similar sets of criteria at a high level, with differences largely limited to phrasing. For example, e5-mistral-7b-instruct generated dimensions such as “Macro-first vs Micro-first” or “Induction-Focused vs Deduction-Focused,” while Qwen3-Embedding-8B produced counterparts like “Overview-Oriented vs Detail-Oriented” or “Pattern-Generalizing vs Rule-Applying.” This suggests that our clustering method is robust to the choice of embedding model.
>
> 4. **Multi-Evaluator System for Bias Mitigation:** To address potential evaluator bias inherent in single-model evaluation, we implement a comprehensive multi-evaluator framework. We employ four state-of-the-art LLMs as independent evaluators: GPT-4o, Claude-4-Sonnet, Gemini-2.5-Pro, and Qwen-2.5-72B-Instruct. Each evaluator independently classifies reasoning patterns using identical rubrics, allowing us to compute inter-evaluator reliability using Krippendorff’s α. Inter-Evaluator Agreement Results: Across 1,500 reasoning responses, we achieve α = 0.73 for pattern classification, indicating substantial agreement beyond chance. Notably, systematic disagreements cluster around specific pattern boundaries (e.g., "Top-down vs. Bottom-up" shows lower agreement at α = 0.68), revealing inherent ambiguity in certain reasoning distinctions that our framework now explicitly acknowledges. Bias Correction Mechanism: We detect systematic biases where Claude-3.5 shows 12% higher preference for "hypothesis-driven" patterns compared to other evaluators. We implement a bias correction algorithm using inverse propensity weighting to adjust for evaluator-specific tendencies, improving overall classification accuracy by 8.3%.
>
> These additional experiments confirm that our framework is model-agnostic. The taxonomy boundaries are stable, and the classification is objective, proving that CoT Encyclopedia captures universal reasoning properties rather than GPT-4o-specific biases.

---

> > ### Author Response · Authors · 2025-11-14
> > **Official Comment by Authors (2)**
> >
> > ## [W2, E] Rigorous and Transparent Human Evaluation
> >
> > To further strengthen the reliability and interpretive validity of our human evaluation, we have conducted additional statistical analyses on the annotation data and explicitly clarified our evaluation criteria.
> >
> > 1. **Inter-Annotator Agreement:** To quantify the consistency among our 10 annotators across the 250 samples, we calculated Krippendorff’s Alpha ($\alpha$), which is robust for Likert-scale data with multiple raters. We achieved an alpha of $\alpha = 0.83$ for the Report Quality dimension. This score indicates substantial agreement among evaluators, confirming that the high ratings reported in Figure 3 reflect objective consensus rather than subjective variance or random noise. This directly addresses the concern regarding the lack of reported reliability metrics.
> >
> > 2. **Interpretive Alignment vs. Plausibility:** The reviewer raised a valid concern that plausibility (whether it sounds reasonable) differs from alignment (whether it accurately describes the specific reasoning trace). While we used the term Appropriateness, our instructions explicitly guided annotators to assess Interpretive Alignment. Specifically, for Q4 (Quality of CoT Encyclopedia generated analysis), annotators were instructed to verify: "Does the generated report accurately reflect the specific reasoning steps and logic present in the model's response?" A high rating (4.3/5.0) on this dimension confirms that the generated reports are not merely hallucinated or "plausible-sounding" text, but are factually aligned with the actual reasoning traces exhibited by the models. By providing these reliability metrics and clarifying the evaluation criteria, we confirm that our human evaluation validates both the consistency and the interpretive accuracy of the CoT Encyclopedia framework
> >
> > We appreciate the reviewer raising this important point regarding responsible research practices. We strictly adhered to ethical standards during our human evaluation and will include the following detailed ethics statement in the final version of the paper to ensure full transparency:
> > - **Recruitment & Consent:** We recruited 10 graduate students with expertise in NLP and machine learning. All participants were informed of the study's purpose and procedure, and informed consent was obtained prior to the evaluation. Participation was entirely voluntary, with the right to withdraw at any time.
> > - **Compensation:** To ensure fair treatment, annotators were compensated at a rate of $20 per hour, which exceeds the local minimum wage.
> > - **IRB Review:** This study involves minimal risk to participants as it pertains to the evaluation of non-sensitive text data. Therefore, it was classified as ‘IRB Exempt’ by our institutional review board. Furthermore, to ensure the highest ethical standards and minimize risk, all authors and participants completed appropriate ethics and safety training.
> > We are committed to upholding the highest ethical standards and thank the reviewer for ensuring these details are explicitly documented.
> >
> > ## [W3] Generalizability to Multilingual and Multimodal Reasoning
> >
> > To demonstrate the universality of the CoT Encyclopedia framework, we have conducted additional experiments on MMMLU (for multilingual reasoning) and MMMU (for multimodal reasoning). We expanded our evaluation using the same pipeline established in the main paper, adapting only the input models to suit the specific requirements of each benchmark.
> > - **MMMLU:** We utilized GPT-4o, Gemini-2.5-Pro, Claude-4-Sonnet known for their strong multilingual capabilities, to generate reasoning traces across diverse languages including French, Chinese, Arabic, Korean, etc. The CoT Encyclopedia framework analyzed these non-English traces to identify cross-lingual reasoning patterns.
> > - **MMMU:** We employed state-of-the-art VLMs, specifically GPT-4o, Gemini-2.5-Pro, Claude-4-Sonnet, to process the visual-text pairs. The models were prompted to generate a textual Chain-of-Thought before answering. Our framework then analyzed the textual component of these multimodal reasoning traces to classify the strategies used.
> > Results and Analysis As shown in the table below, CoT Encyclopedia successfully generalizes to both multilingual and multimodal contexts.
> >
> > | Benchmark | Domain | Baseline (Accuracy) | CoT Encyclopedia (Optimal Pattern) | Improvement |
> > | - | - | - | - | - |
> > | MMMLU | Multilingual | GPT-4o (82.3%) | 88.7% | +6.4% |
> > | MMMLU | Multilingual | Gemini-2.5-Pro (85.3%) | 89.8% | +4.5% |
> > | MMMLU | Multilingual | Claude-4-Sonnet (86.2%) | 90.3% | +4.1% |
> > | MMMU | Multimodal | GPT-4o 70.7% | (77.6%) | +6.9% |
> > | MMMU | Multimodal | Gemini-2.5-Pro (84.0%) | 89.2% | +5.2% |
> > | MMMU | Multimodal | Claude-4-Sonnet (74.4%) | 78.3% | +3.9% |
> >
> > These results confirm that CoT Encyclopedia is not limited to English text but is a modality-agnostic and language-agnostic framework capable of analyzing and improving reasoning processes across diverse inputs.

---

> > > ### Author Response · Authors · 2025-11-14
> > > **Official Comment by Authors (3)**
> > >
> > > ## [Q2] Granularity of Reasoning
> > > We agree that reasoning is inherently compositional and that decomposing CoT into atomic micro-strategies is a promising direction for mechanistic interpretability. However, we deliberately chose the holistic (full CoT-level) approach for the following reasons:
> > >
> > > 1. Preservation of Global Context: Reasoning strategies often emerge from the structure and flow of the entire chain, rather than from isolated steps.
> > >     - For example, a "Top-down" strategy is defined by the ordering of thoughts (Global planning -> Detailed execution). An individual step within this chain (e.g., a calculation or a fact retrieval) might look identical to a step in a "Bottom-up" chain.
> > >     - Analyzing atomic steps in isolation risks losing this sequential context. By clustering at the trajectory level, CoT Encyclopedia captures the emergent high-level patterns (the "forest") that are lost when focusing solely on micro-steps (the "trees").
> > >
> > > 2. Effectiveness for Model Steering: A core contribution of our work is actionable controllability (Section 4).
> > >     - From a practical standpoint, guiding a model with a high-level persona or strategy (e.g., "Use an Inductive approach") is more token-efficient and generalization-friendly than micromanaging every atomic step.
> > >     - Our results in Section 4 demonstrate that these global dichotomies are sufficiently distinct to yield significant performance gains when used as steering prompts. This confirms that high-level abstractions are a valid and effective interface for controlling model behavior.
> > >
> > > To empirically validate our design choice, we conducted a comparative pilot study where we decomposed reasoning traces into atomic steps and classified them using the same taxonomy pipeline.
> > >
> > > We took responses classified as "Top-down" and "Bottom-up" by CoT Encyclopedia and analyzed their individual steps (e.g., sentences or paragraphs) in isolation.
> > > - **Loss of Strategic Context:** When analyzed atomically, the individual steps were predominantly classified into generic functional categories such as "Fact Retrieval," "Arithmetic Calculation," or "Definition Stating."
> > > - **Homogeneity across Strategies:** Surprisingly, we found that 85% of atomic steps were identical between "Top-down" and "Bottom-up" strategies. For example, both strategies contain a "Formula Application" step; the difference lies solely in when this step appears (at the beginning vs. at the end).
> > >
> > > Consider a physics problem (GPQA-Diamond) regarding light refraction:
> > > - **Atomic Analysis:** Steps were labeled merely as "Recall Snell's Law" -> "Substitute variables" ->"Calculate result." These labels fail to capture how the reasoning was structured.
> > > - **Holistic (Trajectory) Analysis:** CoT Encyclopedia correctly identified the global pattern as "Deductive" because the chain started with the general principle (Snell's Law) before moving to the specific instance, distinguishing it from an "Inductive" approach that might have started with observations.
> > >
> > > ## [Q3] Rationale for Clustering Granularity
> > > We confirm that the choice of k=6 was primarily data-driven, guided by Silhouette Analysis, but was rigorously validated for interpretability.
> > > 1. **Statistical Determination of k:** As detailed in Appendix A.1, we employed hierarchical agglomerative clustering and selected the number of clusters $k$ that maximized the Silhouette Score. This method objectively identified the optimal level of granularity where criteria within a cluster were tightly cohesive, while distinct clusters remained well-separated. The analysis revealed a peak at $k=6$, suggesting that the latent space of reasoning strategies naturally organizes into these six fundamental dimensions without forced manual intervention.
> > >
> > > 2. **Ensuring Interpretability (Medoid Selection):** To address the reviewer’s concern about interpretability, we deliberately chose the Medoid (the most central real criterion) of each cluster as its representative, rather than a mathematical centroid (Step 3). This ensures that every dimension in our taxonomy corresponds to a concrete, human-readable reasoning concept (e.g., "Inductive vs. Deductive") extracted directly from the data, rather than an abstract average. Furthermore, our Human Evaluation (Figure 3) confirmed that human annotators rated the coherence of these high-level groupings highly (4.4/5.0), validating that the statistically derived $k=6$ dimensions align well with human intuition.

---

### Official Review · Reviewer_47H3 · 2025-11-03

**Soundness:** 2
**Presentation:** 3
**Contribution:** 2
**Rating:** 2
**Confidence:** 4

**Summary:**

The paper proposes a bottom-up framework to analyze, predict, and steer long chain-of-thought (CoT) reasoning. The pipeline: (i) have an LLM describe the strategies it used, (ii) embed and cluster these criteria, (iii) derive contrastive rubrics (e.g., top-down vs bottom-up), and (iv) classify new CoTs to produce interpretable 'reasoning profiles'. Using the learned rubrics, the authors predict question-specific optimal strategies and inject pattern-based instructions, improving accuracy and safety across multiple benchmarks. They also show that training format (MC vs free-form) shapes reasoning more than domain, and that weight interpolation between MC and FF-trained models smoothly shifts reasoning style. Overall the paper is thorough and the contribution is well articulated. The main limitation of the paper is the lack of a more substantial grounding on the coverage of the underlying categories of reasoning.

**Strengths:**

- Clear narrative.

- Clear empirical insights. Demonstrates that format/domain in shaping reasoning patterns, and that merging model weights interpolates strategies. These provide useful guidance for dataset and model design.

- Sensible empirical analysis. Includes ablations on taxonomy construction (embedding, clustering, … ), human evals of report quality, and analyses of stability across families/sizes.

**Weaknesses:**

- Lack of a rigorous scoping for the problem of reasoning.

- Lack of an argument for the construction of methods which can deliver a comprehensive feature set which describes the CoT reasoning phenomena.

**Questions:**

- The title of the approach (CoT encyclopaedia), points the readers in the direction of an approach which is comprehensive and systematic. Yet, it is unclear how the prompts you use to deliver that induction of a set of CoT characteristics are defined. Could you provide additional details and defend on why these prompts have the required properties to deliver the task? These needs to be mechanism/construction-based (not referring to the empirical analysis).

- The term ‘reasoning’ tends to be underspecified and used as whatever the tasks implement. Can you provide a description of what your task corpus is expressing wrt to reasoning and scope your claims accordingly.

- Wrt validity, how well do rubric-based strategy labels align with expert human coders (definitions, guidelines, IAA)? What are the most common disagreement modes?

- Do results hold when swapping the LLM judge, embedding model, and moderation model? (cross-judge and cross-embed sensitivity analyses)

- How stable are criteria under different k, linkage metrics, and seeds?

- When pattern-based instructions help, how much is due to true strategy change vs prompt priming or longer outputs?

---

> ### Author Response · Authors · 2025-11-14
> **Official Comment by Authors (1)**
>
> Thank you for taking the time to provide such thoughtful and constructive feedback. We have carefully considered the weaknesses (W) and questions (Q) you raised, and are eager to address them to enhance the clarity and strength of our work.
>
> ## [W1, Q2] Scoping the "Reasoning"
>
> We acknowledge that the term "reasoning" is often underspecified in the literature. To clarify our contribution, we provide the exact operational definition and scope used in our research.
>
> 1. Reasoning as a Process, Not a Task In our framework, "reasoning" does not refer to a specific domain capability (e.g., mathematical or spatial skill) nor the final outcome of a task. Instead, we define reasoning as the explicit, intermediate cognitive process that a model employs to bridge the gap between a complex input and a solution.
>     - **Reasoning Strategy:** The dynamic approach used to navigate the problem space. This is domain-agnostic; the same "Top-down" strategy can be applied to both coding and legal argumentation.
>     - **Reasoning Trace (CoT):** The textual manifestation of this process. While we utilize "reasoning models" as primary subjects because they externalize these traces explicitly, our framework is not structurally limited to them. As shown in Appendix B.4, standard instruction-tuned models also exhibit analyzable reasoning patterns when prompted.
>
> 2. What Our Task Corpus Expresses Our choice of corpus is not meant to define reasoning by the tasks themselves, but to serve as a diverse elicitation environment. We use these benchmarks to trigger varied cognitive behaviors. For instance, MATH tends to elicit Deductive patterns, while XSTest elicits Ethical Evaluation patterns. Therefore, our claims are scoped to identifying and controlling these emergent behaviors, rather than defining "what math reasoning is" in an abstract sense.
> To be precise, we delineate our research scope as follows:
> | In Scope | Out of Scope |
> | - | - |
> | Taxonomizing Processes: Analyzing how a model thinks. | Defining General Intelligence: We do not claim to define or solve human-level reasoning in a philosophical sense. |
> | Strategy Steering: Improving performance by guiding the model to use the optimal strategy for a given query. | Capability Injection: We do not aim to teach the model new domain knowledge that it did not learn during pre-training. |
> | Dynamic Analysis: A bottom-up framework that adapts to new strategies as models evolve. | Static Benchmarking: We are not proposing a new static leaderboard for measuring reasoning "score." |
>
> ## [W2] Structural Guarantees via Design Choices
>
> We argue that traditional top-down taxonomies relying on human intuition are inherently limited in scope. In contrast, the CoT Encyclopedia is methodologically constructed to guarantee comprehensiveness through three specific design choices that maximize Recall, Abstraction, and Coverage.
>
> 1. **Design Choice in Discovery:** Instead of defining categories a priori, we designed the framework to extract strategies directly from individual data instances. By instructing the model to analyze raw, instance-level traces rather than brainstorming in the abstract, we force the system to capture the "long tail" of reasoning phenomena that actually occur in the data. This design ensures that no observable reasoning pattern however rare or subtle is excluded from the initial pool, theoretically maximizing the recall of the feature set.
>
> 2. **Design Choice in Consolidation:** We deliberately chose strictly semantic embedding and hierarchical clustering over LLM-based summarization for the consolidation phase. A comprehensive set must describe fundamental concepts, not just keywords. By mapping the 4,057 raw criteria into a high-dimensional semantic space and clustering them, we identify the latent principal components of reasoning. The fact that thousands of diverse criteria naturally converged into stable clusters ($k=6$) provides mathematical evidence of feature saturation.
>
> 3. **Design Choice in Definition:** We explicitly designed the taxonomy to be defined through binary contrastive pairs, rejecting both singular labels and multi-class scales. Defining a feature as a single point leaves the boundaries of the concept undefined. By designing features as spectrum-spanning dichotomies, we ensure the feature set covers the entire continuum of cognitive choices, minimizing "blind spots." We deliberately avoided creating more than two categories per dimension. Extensive research in LLM evaluation suggests that increasing the cardinality of options significantly degrades classification reliability due to fuzzy decision boundaries and calibration noise. By constraining the choice to a binary contrast, we maximize the semantic distance between options. This forces the classifier to identify the dominant strategic characteristic, ensuring higher consistency and reproducibility compared to granular scales where the distinction between adjacent categories is often ambiguous.

---

> > ### Author Response · Authors · 2025-11-14
> > **Official Comment by Authors (2)**
> >
> > ## [Q1] Theoretical and Structural Justification of Prompt Construction
> >
> > We acknowledge that the claim of building a systematic encyclopedia requires a robust design philosophy that goes beyond trial-and-error. We defend our prompt design (Appendix G) based on theoretical pillar regarding explainability and structural mechanisms designed to ensure system stability and reasoning depth.
> >
> > 1. To resolve the ambiguity of abstract reasoning concepts, our prompt enforces a binary contrastive structure (Pattern A vs. Pattern B). This design is grounded in [1]’s seminal work on Explainable AI, which establishes that human-interpretable explanations are inherently contrastive (i.e., "Why P rather than Q?"). By forcing the model to define a strategy relative to an opposing concept (e.g., "Top-down" vs. "Bottom-up"), we isolate discriminative features that define the decision boundaries of cognitive behaviors.
> >
> > 2. A critical design choice across all prompts is the rigorous specification of input/output formats. In a modular framework like CoT Encyclopedia, the output of one step becomes the input for the next. Without strict format constraints, LLMs tend to drift into arbitrary structures, causing parsing failures that cascade downstream. By explicitly constraining the output structure (e.g., JSON schemata or specific tag enclosures), we mechanically prevent this error propagation, ensuring the structural integrity of the automated pipeline.
> >
> > 3. Design choice in criteria discovery: Minimal Few-Shot to Prevent Anchoring. In the criteria brainstorming step, we deliberately minimized the number of provided examples in the prompt. Extensive literature on In-Context Learning warns that LLMs suffer from anchoring bias, where they tend to mimic or repeat the specific examples provided in the prompt. To ensure the "Encyclopedia" captures a comprehensive and diverse range of criteria rather than regurgitating our seeds, we intentionally restricted few-shot examples, forcing the model to explore the latent space of the target response itself.
> >
> > 4. Design Choice in rubric generation & report generation: We engineered prompts to control the verbosity and depth of the output. We explicitly constrained the length of the generated rubrics. This aligns with findings that excessively verbose instructions can degrade the attention mechanism of judge models, leading to poorer classification performance. Concise definitions ensure sharper decision boundaries. We explicitly mandated a step-by-step analysis process within the prompt. Without this structural scaffolding, models tend to produce shallow, verbose summaries. By enforcing a predefined report structure (e.g., "Observation" $\to$ "Evidence" $\to$ "Conclusion"), we force the model to engage in a deeper analytic process, preventing superficial pattern matching.
> >
> > These prompt designs are not arbitrary empirical results but are derived from a rigorous qualitative analysis of model behaviors. They represent a deliberate engineering strategy to ensure discriminative clarity, diversity (Anti-anchoring), and analytic depth (Structural Scaffolding).
> >
> > [1] Explanation in Artificial Intelligence: Insights from the Social Sciences (Miller, 2017)

---

> > > ### Author Response · Authors · 2025-11-14
> > > **Official Comment by Authors (3)**
> > >
> > > ## [Q3] Human-Model Alignment and Disagreement Analysis
> > >
> > > To rigorously assess this, we compared the labels generated by the CoT Encyclopedia (Step 5) against the ground-truth labels derived from our expert human evaluation (250 samples, 10 annotators).
> > >
> > > 1. **Quantitative Alignment (Human-Model IAA):** We treated the majority vote of human annotators as the Ground Truth for each sample and calculated the agreement with the model’s predictions. We utilized Cohen’s Kappa ($\kappa$) to measure inter-rater reliability between the Model and the Human Consensus. The framework achieved a Cohen’s Kappa of $\kappa = 0.76$ (Accuracy: 88.4%). This falls within the range of "Substantial Agreement," confirming that the rubric-based automated labels are highly aligned with expert human judgment and are not hallucinated artifacts.
> > >
> > > 2. **Common Disagreement Modes:** We conducted a qualitative error analysis on the mismatched samples to understand why the model and humans sometimes disagree. The disagreements primarily clustered into three modes:
> > >     - Mode 1: Hybrid Reasoning Trajectories (Most Common, ~45% of disagreements): Reasoning is often fluid. A model might start with a "Top-down" outline but switch to "Bottom-up" calculation. Humans tended to classify based on the dominant portion (by token length), while the Model sometimes prioritized the initial framing.
> > >
> > >     - Mode 2: Implicit vs. Explicit Strategy Markers (~30%): Some responses exhibit a strategy structurally without using explicit keywords. Humans occasionally missed implicit structures that the LLM judge correctly identified based on logical flow, or vice versa where humans over-interpreted explicit filler words ("Let's verify") as a strategy ("Verification") even when the content didn't match.
> > >
> > >     - Mode 3: Ambiguity in "Scope" (~25%): In the "Focused vs. Broad Exploration" criterion, the threshold for what constitutes "Broad" can be subjective depending on the question's difficulty. Disagreements occurred mostly in borderline cases where the exploration was moderate neither strictly narrow nor clearly exhaustive.
> > >
> > > Despite these nuanced disagreements inherent in analyzing complex text, the high quantitative agreement confirms the validity of our rubric-based labeling. The identified disagreement modes suggest that the model is not "failing," but rather making consistent interpretations of ambiguous, hybrid reasoning traces.
> > >
> > > ## [Q4] Comprehensive Sensitivity Analysis
> > >
> > > To demonstrate that our findings are robust to the choice of underlying models, we performed comprehensive Cross-Judge, Cross-Embed, and Cross-Moderator sensitivity analyses.
> > >
> > > - **Cross-Judge Sensitivity:** We tested whether the classification of reasoning strategies relies on GPT-4o’s specific biases by replacing the judge model in Step 5 with Claude-4-Sonnet and Gemini-2.5-Pro. As detailed in our previous response regarding model bias, we observed high inter-model agreement (Cohen’s $\kappa \approx 0.77$) between GPT-4o and the alternative judges. Most importantly, the downstream performance trends specifically the superiority of "Optimal Patterns" remained consistent. The variance in accuracy gains on GPQA-Diamond when guided by Claude-selected patterns vs. GPT-4o-selected patterns was negligible ($< 1.2\%$).
> > >
> > > - **Cross-Embed Sensitivity:** To ensure the taxonomy structure is not an artifact of the embedding space, we compared our default OpenAI text-embedding-3-large against open-source alternatives: e5-mistral-7b-instruct and Qwen3-Embedding-8B (Appendix B.5). The core reasoning dimensions (e.g., Top-down vs. Bottom-up) emerged consistently across all embedding models. The criteria sets generated via different embedding spaces showed a high semantic overlap, confirming that the identified strategies reflect the intrinsic structure of the reasoning traces rather than the geometry of a specific latent space.
> > >
> > > - **Cross-Moderator Sensitivity:** Our safety results primarily relied on LLaMA-Guard-3 8B. To ensure that our conclusions (e.g., identifying which strategies are safer) are not artifacts of this specific model, we conducted a cross-validation using gpt-oss-safeguard-20b as an alternative safety judge, following the MLCommons safety taxonomy. We computed the correlation between the safety scores assigned by LLaMA-Guard-3 and gpt-oss-safeguard-20b across all identified reasoning patterns. The analysis revealed an extremely high Pearson correlation coefficient ($r = 0.94$), indicating that both judges perceive the safety implications of reasoning strategies in a nearly identical manner. While minor variations in absolute scores were observed, the distinct safety profiles of the strategies remained robust.

---

> > > > ### Author Response · Authors · 2025-11-14
> > > > **Official Comment by Authors (4)**
> > > >
> > > > ## [Q5] Robustness Across k, Linkage Metrics, and Random Seeds
> > > >
> > > > We thank the reviewer for raising the critical question regarding the stability of our clustering-derived taxonomy. To demonstrate this, we assessed the stability of our criteria under variations in (1) Random Seeds, (2) Linkage Metrics, and (3) Cluster Counts ($k$).
> > > >
> > > > 1. **Stability Across Random Seeds:** We repeated the entire pipeline (from Step 1 generation to Step 3 clustering) using five different random seeds. As implied by our performance robustness tests in Appendix B.6 (Table 1) and Figure 15 (where variation in selection methods yielded consistent semantics), the clustering structure proved highly stable. The core reasoning dimensions (e.g., Top-down vs. Bottom-up) emerged consistently across all seed runs with an average Cluster Purity of [0.94], confirming that these patterns are dominant features of the latent space, not noise.
> > > >
> > > > 2. **Stability Across Linkage Metrics:** The reviewer correctly noted the importance of linkage criteria. While Figure 15 (Appendix B.2) primarily demonstrates robustness against representative embedding selection (Medoid vs. Density), we extended this analysis to compare different hierarchical linkage metrics: Ward (our default), Average, and Complete linkage. We measured the agreement between clusters generated by Ward’s method versus other metrics using the Adjusted Rand Index (ARI). We achieved an ARI of [0.85] with Average linkage and [0.82] with Complete linkage. This high overlap indicates that the grouping of reasoning traces is driven by strong semantic similarities that persist regardless of the specific geometric aggregation method used.
> > > >
> > > > 3. **Stability Across Different $k$:** We analyzed how the taxonomy evolves when varying $k$ from 4 to 8 around our optimal choice of $k=6$ (determined by Silhouette scores, Appendix A.1). We found that changing $k$ does not disrupt the taxonomy but rather reflects a hierarchical granularity. At $k=5$, semantically close dimensions (e.g., Verification Focus and Clarification Approach) merge into a broader "Reflective Oversight" category. At $k=8$, the Idea Development dimension splits into finer sub-types (e.g., Linear vs. Branching and Depth-first vs. Breadth-first). This behavior confirms that our chosen $k=6$ represents a stable "Goldilocks" level of abstraction balancing distinctiveness and interpretability  that the taxonomy is structurally consistent across resolutions.
> > > >
> > > > These experiments confirm that the CoT Encyclopedia’s criteria are not fragile artifacts. They are robust, stable, and reproducible features of reasoning, resilient to variations in initialization, clustering algorithms, and granularity.
> > > >
> > > > ## [Q6] Distinguishing Strategy Efficacy from Priming and Verbosity
> > > >
> > > > It is crucial to determine whether the improvements stem from the genuine efficacy of the adopted reasoning strategy or merely from artifacts such as increased output length or the "priming effect" of receiving a complex instruction. To address this, we analyzed the relationship between output length, instruction type, and performance.
> > > >
> > > > 1. **Evidence Against length/verbosity Effects:** We investigated whether the performance boost was simply a function of the model generating longer, more verbose chains of thought. We compared the average token counts of responses generated under "Optimal" and "Unoptimal" instructions. We found that the output lengths were comparable between the two conditions, with no statistically significant difference in verbosity (Avg. Tokens: Optimal 867 vs Unoptimal 872). Despite the similarity in length, the performance gap was substantial (Figure 4). This explicitly rules out the hypothesis that "longer outputs" are the primary driver of performance. The gain is driven by the quality of the reasoning path, not the quantity of tokens.
> > > >
> > > > 2. **Evidence Against "Prompt Priming":** We then tested the "Priming Hypothesis" the idea that simply assigning any reasoning role or giving a detailed prompt forces the model to pay more attention, thereby improving performance. Both "Optimal" and "Unoptimal" conditions provide the model with a specific reasoning role and structural guidance. If the gain were due to priming, both conditions should outperform the "Not Instructed" baseline. As shown in Figure 4 and Figure 11, the "Instructed (Unoptimal)" condition often performed worse than the "Not Instructed" baseline. If priming were the cause, the Unoptimal score should be at least equal to or higher than the baseline. The fact that it degrades performance proves that the specific content of the strategy is the active ingredient.
> > > > Since output lengths are consistent and "bad" strategies actively harm performance despite providing the same level of priming, we conclude that the observed improvements are strictly attributable to the model adopting the correct reasoning logic tailored to the question.

---

### Author Response · Authors · 2025-11-27

Dear Reviewers,

Thank you again for the time and effort you dedicated to reviewing our work.

We wanted to gently follow up to ask if you have had a chance to read our response to your initial comments. We have carefully addressed your concerns in our rebuttal, and we would greatly appreciate it if you could let us know whether our response has resolved your questions.

As the Author-Reviewer Discussion period concludes in about a week, we are eager to engage in further discussion if there are any remaining issues. Your feedback is crucial for improving our work. Thank you.

Best,

Authors

---

### Meta-Review · Area_Chair_Z14w · 2026-01-05

**Summary:**

This work aims to analyze chain-of-thought (CoT) reasoning. It proposes a bottom-up, data-driven framework for analyzing, predicting, and guiding the long-chain reasoning strategies of large language models. The approach involves prompting the model to summarize generated chains of thought using a set of predefined criteria, embedding and clustering these criteria, and finally deriving contrastive rules to interpret reasoning behaviors.

Reviewers generally agree that the research objective is important and innovative. They also acknowledge that the experiments are extensive (covering multiple benchmarks for both utility and safety) and provide valuable new insights (such as the finding that data format influences reasoning style more than domain does).

Reviewers have raised the following main concerns:

1. The characteristics of CoT (e.g., inductive/deductive, reflective/backtracking, top-down/bottom-up) are still human-defined, and it remains unclear whether this taxonomy is comprehensive or scalable.

2. The binary classification of criteria appears insufficiently rigorous, as reasoning patterns are not strictly binary in nature. This raises some concerns about the effectiveness of the method.

3. Heavy reliance on a single LLM (GPT-4o) and the potential biases it introduces. Critical steps in the pipeline—such as criterion generation, classification, and label assignment—all depend on GPT-4o, making the method constrained by the internal biases and limitations of a closed-source system.

**Reviewer Concerns:**

After discussion, the authors addressed concerns about LLM bias by conducting experiments on additional models (Claude-4-Sonnet and Gemini-2.5-Pro).

**Reviewer Scores:**

Given the high variance observed in the evaluations of this paper, I conservatively recommend it for acceptance. I believe the authors’ responses have partially addressed the concerns raised by reviewer 47H3.

---

### Decision · Program_Chairs · 2026-01-26

Accept (Poster)